# Learning Representations for Independence Testing

**Nathaniel Xu**                                    *xunathan@cs.ubc.ca*
*University of British Columbia*

**Feng Liu**                                        *feng.liu1@unimelb.edu.au*
*University of Melbourne*

**Danica J. Sutherland**                            *dsuth@cs.ubc.ca*
*University of British Columbia*
*Alberta Machine Intelligence Institute*

**Reviewed on OpenReview:** *https://openreview.net/forum?id=pDvKoXRsnW*

## Abstract

Many tools exist to detect dependence between random variables, a core question across a wide range of machine learning, statistical, and scientific endeavors. Although several statistical tests guarantee eventual detection of any dependence with enough samples, standard tests may require an exorbitant amount of samples for detecting subtle dependencies between high-dimensional random variables with complex distributions. In this work, we study two related ways to learn powerful independence tests. First, we show how to construct powerful statistical tests with finite-sample validity by using variational estimators of mutual information, such as the InfoNCE or NWJ estimators. Second, we establish a close connection between these variational mutual information-based tests and tests based on the Hilbert-Schmidt Independence Criterion (HSIC); in particular, learning a variational bound (typically parameterized by a deep network) for mutual information is closely related to learning a kernel for HSIC. Finally, we show how to, rather than selecting a representation to maximize the statistic itself, select a representation which can maximize the power of a test, in either setting; we term the former case a Neural Dependency Statistic (NDS). While HSIC power optimization has been recently considered in the literature, we correct some important misconceptions and expand to considering deep kernels. In our experiments, while all approaches can yield powerful tests with exact level control, optimized HSIC tests generally outperform the other approaches on difficult problems of detecting structured dependence.

## 1 Introduction

Independence testing, the question of using paired samples to determine whether a random variable $X$ and another $Y$ are associated with one another or if they are statistically independent, is one of the most common tasks across scientific and data-based fields. Traditional methods make strong parametric assumptions, for instance assuming that $X$ and $Y$ are jointly normal so that dependence is characterized by covariance, and/or operate only in limited settings, for instance the tabular setting of the celebrated $\chi^2$ test or Fisher's exact test. Applying these approaches to high-dimensional continuous data is difficult at best.

One characterization of independence is the Shannon mutual information (MI): this quantity is zero if two random variables are independent, and positive if they are dependent. Substantial effort has been made in estimating this quantity with a variety of estimators; see e.g. the broad list based on binning, nearest-neighbors, kernel density estimation, and so on implemented by Szabó (2014). In high-dimensional cases, however, recent work has focused on estimating variational bounds defined by deep networks (see e.g. Poole et al., 2019), which can ideally learn problem-specific structure. To our knowledge, this class of estimators has not yet been used to construct *statistical tests* of independence: in particular, if we run the estimation algorithm and

estimate a lower bound on the MI of 0.12, does that mean that the variables are (perhaps weakly) dependent, or might that be due to random noise on the samples we saw? The first contribution of this paper is to construct and evaluate tests of this form.

Mutual information, though, is notoriously difficult to estimate (Paninski, 2003). If the question we want to ask is "are $X$ and $Y$ dependent," we can also consider turning to a different characterization of dependence which may be statistically easier to estimate. The Hilbert-Schmidt Independence Criterion (HSIC), introduced by Gretton et al. (2005), measures the total cross-covariance between feature representations of $X$ and $Y$, and can be estimated from samples efficiently. The construction supports even *infinite-dimensional* features in a reproducing kernel Hilbert space (RKHS), equivalent to choosing a kernel function. With appropriate choices of kernel (see Szabó & Sriperumbudur, 2018), the HSIC is also zero if and only if $X \perp\!\!\!\perp Y$. Székely et al. (2007) and Lyons (2013) separately proposed *distance covariance* tests, which measure the covariance of pairwise distances between $X$ values with distances between $Y$ values; this can be viewed as HSIC with a particular kernel (Sejdinovic et al., 2013). Our second contribution describes how HSIC, in fact, is also a lower bound on mutual information, and that tests based on either are very closely related.

Alternatively, we can characterize an independence problem as a two-sample one. Two-sample problems are concerned with the question: given samples from $\mathbb{P}$ and $\mathbb{Q}$, is $\mathbb{P} = \mathbb{Q}$? Under this framework, when we consider samples from the joint distribution $\mathbb{P}_{xy}$ and the product of the marginals $\mathbb{P}_x \times \mathbb{P}_y$ (where $X \perp\!\!\!\perp Y$ by construction), the two-sample problem characterizes an equivalent independence problem between variables $X$ and $Y$. One way we can measure the discrepancy between $\mathbb{P}_{xy}$ and $\mathbb{P}_x \times \mathbb{P}_y$ is with the kernel-based Maximum-Mean Discrepancy (MMD) (Gretton et al., 2012a); doing so recovers HSIC.

Any reasonable choice of kernel, such as the Gaussian kernel with unit bandwidth or the distance kernel that yields distance covariance, will *eventually* (with enough samples) be able to detect any fixed dependence. In practice, however, this scheme can perform extremely poorly; if, for instance, the data varies on a very different scale than the bandwidth of the Gaussian kernel, it may take exorbitant quantities of data to achieve any reasonable test power. Thus, tests using Gaussian kernels often rely on the *median heuristic* to choose a kernel relevant to the data at hand: choosing a bandwidth based on the median pairwise distance among data points (Gretton et al., 2012a). While this is a reasonable first guess for many data types, there exist many two-sample testing problems where it can be dramatically better to instead select a bandwidth that optimizes a measure of test power (Sutherland et al., 2017). Beyond that, there are many problems where no Gaussian kernel performs well – for instance, many problems on natural image data or involving sparsity – but Gaussian kernels applied to latent representations of such data do. As an extreme example in the dependence case, consider the construction $X \sim \mathcal{N}(0,1)$, $Y_0 \sim \mathcal{N}(0,1)$, but then $Y$ is obtained by replacing the fifteenth decimal place of $Y_0$ with that of $X$. Representations based on Euclidean distance would require an absurd number of samples to detect the dependence between $X$ and $Y$, but a representation that extracts only the fifteenth decimal place will make the dependence obvious.

These considerations apply similarly to tests based on mutual information. In fact, we can view estimation of a variational MI bound as learning a representation of the data that maximizes its measure of dependency. Thus, learning representations of the data that make any dependency more explicit is central to developing more powerful independence tests. Our main algorithmic contribution involves developing a scheme to learn these optimal representations, both for HSIC tests and for a class of tests closely related to the variational MI-based tests. In both settings, this is based on maximizing an estimate of the asymptotic power of the test, the primary term of which is an estimate of the signal-to-noise ratio of the statistic: the estimator divided by its standard deviation. In both cases, we can efficiently, and differentiably, estimate this quantity; we also show via a uniform convergence argument that optimizing the power estimate leads to a representation which generalizes well. Using data splitting and permutation testing (as in e.g. Rindt et al., 2021), we obtain a test which is exactly valid and achieves high power. In experiments, we find that while both methods often work well, there are settings where the HSIC optimization scheme far outperforms the other categories of tests.

In both cases, the overall approach builds on prior work in kernel two-sample testing (Gretton et al., 2012b; Jitkrittum et al., 2016; Sutherland et al., 2017; Liu et al., 2020). We show, however, that performing the direct reduction to two-sample testing and applying these techniques, in addition to breaking the assumptions of their theoretical results, yields a notably less powerful test than our direct HSIC power optimization, due to

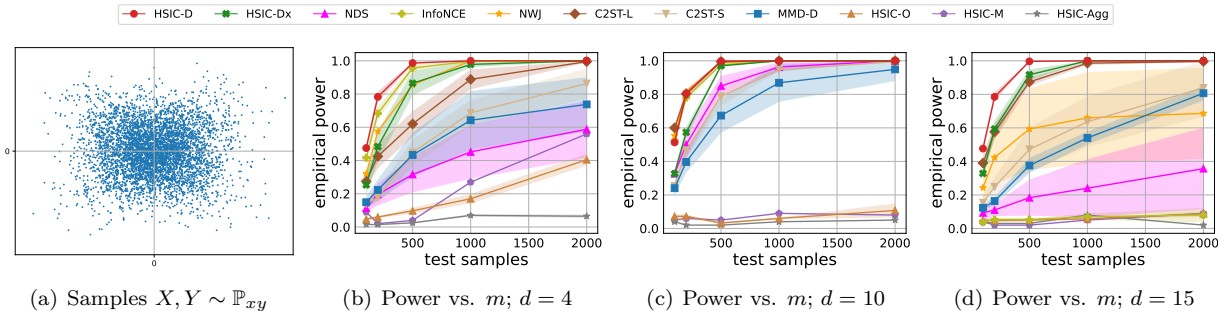

(a) Samples $X, Y \sim \mathbb{P}_{xy}$     (b) Power vs. $m$; $d = 4$     (c) Power vs. $m$; $d = 10$     (d) Power vs. $m$; $d = 15$

Figure 1: Vanilla kernel-based HSIC tests struggle on high-dimensional data. (a) A bimodal Gaussian mixture, with visible dependence between $X$ and $Y$. (b)-(d) Test power as we add an increasing number of independent noise dimensions. When $d = 4$ the median heuristic (HSIC-M) confidently detects dependence between $X$ and $Y$ with a reasonably large sample size, but struggles when $d = 10$ or $d = 15$. Our method (HSIC-D) detects dependence in high dimensions with many fewer samples.

statistical properties of the estimators. We also note that Ren et al. (2024) recently proposed a closely related scheme for the HSIC power optimization setting; we argue, however, that several of their main algorithmic suggestions are not well-supported (particularly in Appendix F.5) and identify an apparent gap in the proof of their main theorem (see Footnote 10).

## 2 Tests Based On Variational Mutual Information Bounds

**Problem 2.1** (Independence testing). *Let $Z = (X, Y) \sim \mathbb{P}_{xy}$ on a domain $\mathcal{X} \times \mathcal{Y}$, and $\mathbb{P}_x$, $\mathbb{P}_y$ be the corresponding marginal distributions of $\mathcal{X}$ and $\mathcal{Y}$. We observe $m$ paired samples $\mathbb{X} = (x_1, ..., x_m)$ and $\mathbb{Y} = (y_1, ..., y_m)$ jointly drawn from $\mathbb{P}_{xy}$. We wish to conduct a null hypothesis significance test, with null hypothesis $\mathfrak{H}_0 : \mathbb{P}_{xy} = \mathbb{P}_x \times \mathbb{P}_y$ and alternative $\mathfrak{H}_1 : \mathbb{P}_{xy} \neq \mathbb{P}_x \times \mathbb{P}_y$.*

We wish to solve this problem without making strong (parametric) assumptions about the form of $\mathbb{P}_{xy}$, $\mathbb{P}_x$, or $\mathbb{P}_y$. Most independence tests are based on estimating the "amount" of dependence between $X$ and $Y$, or equivalently the discrepancy between between $\mathbb{P}_{xy}$ and $\mathbb{P}_x \times \mathbb{P}_y$. Given a nonnegative quantity which is zero if $X \perp\!\!\!\perp Y$, we can reject $\mathfrak{H}_0$ if our estimate is large enough that we are confident the true value is positive.

The most famous such quantity is the Shannon mutual information $\mathrm{I}(X; Y)$. It can be defined as the Kullback-Liebler divergence of $\mathbb{P}_{xy}$ from $\mathbb{P}_x \times \mathbb{P}_y$, and is zero if and only if $X \perp\!\!\!\perp Y$; equivalently, it is the amount by which knowledge of $Y$ decreases the entropy of $X$. While many estimators exist based roughly on various forms of density estimation (see e.g. Szabó, 2014), as discussed above these can fail to detect subtle or structured forms of dependence with reasonable numbers of samples. Variational estimators of mutual information give an opportunity for problem-specific representations. As an example, consider the following lower bound (van den Oord et al., 2018; Poole et al., 2019), called InfoNCE for its connection to noise-contrastive estimation (Gutmann & Hyvärinen, 2012):

$$\mathrm{I}_{\mathrm{NCE}}^{f,K}(X; Y) = \underset{(x_i, y_i) \sim \mathbb{P}_{xy}^K}{\mathbb{E}} \left[ \hat{\mathrm{I}}_{\mathrm{NCE}}^{f,K} \right] \leq \mathrm{I}(X; Y) \qquad \text{where} \quad \hat{\mathrm{I}}_{\mathrm{NCE}}^{f,K} = \frac{1}{K} \sum_{i=1}^{K} \log \frac{e^{f(x_i, y_i)}}{\frac{1}{K} \sum_{j=1}^{K} e^{f(x_i, y_j)}}.$$

Here $K$ is a batch size, which in our setting will typically be the total number of available points. Each $f : \mathcal{X} \times \mathcal{Y} \to \mathbb{R}$ (and each $K$) leads to a different lower bound; the largest $\mathrm{I}_{\mathrm{NCE}}^{f,K}$ is the tightest bound. In practice, users typically parameterize $f$ as a deep network and maximize $\hat{\mathrm{I}}_{\mathrm{NCE}}^{f,K}$ on minibatches from a training set, providing an opportunity for $f$ to learn useful feature adapted to the problem at hand. There are a variety of bounds of this general type; Poole et al. (2019) give a unified accounting of many. Different bounds yield different bias-variance tradeoffs, but run up against various limitations on the possibility of estimation (Song & Ermon, 2020; McAllester & Stratos, 2020).

For independence testing, the key question is whether the true value $\mathrm{I}(X; Y) > 0$. This is guaranteed if a lower bound, such as $\mathrm{I}_{\mathrm{NCE}}^{f,K}$ for some particular $f$ and $K$, is positive *at the population level*, i.e. with the true

expectation. To estimate this, by far the easiest approach is data splitting: choose $f$ to maximize $\hat{\mathrm{I}}_{\mathrm{NCE}}^{f,K}$ on (minibatches from) a training set, then evaluate $\hat{\mathrm{I}}_{\mathrm{NCE}}^{f,K}$ on the heldout test set.

How large should $\hat{\mathrm{I}}_{\mathrm{NCE}}^{f,K}$ be in order to be confident that $\mathrm{I}_{\mathrm{NCE}}^{f,K} > 0$? That is, what does the distribution of $\hat{\mathrm{I}}_{\mathrm{NCE}}^{f,K}$ look like when $\mathrm{I}(X;Y) = 0$, and hence $\mathrm{I}_{\mathrm{NCE}}^{f,K} \leq 0$? For a given $f$, we can answer this question with *permutation testing*, which estimates values under the null hypothesis ($\mathbb{P}_{xy} = \mathbb{P}_x \times \mathbb{P}_y$) by randomly shuffling the test data, breaking dependence. To construct a test with probability of false rejection at most $\alpha$, we can compute the empirical $1 - \alpha$ quantile from this permuted set, as long as we include the original paired data in this shuffling (Hemerik & Goeman, 2018, Theorem 2). We reject the null hypothesis if this quantile is smaller than the test statistic

$$\hat{\mathrm{I}}_{\mathrm{NCE}}^{f,K} = \frac{1}{K} \sum_{i=1}^{K} f(x_i, y_i) - \frac{1}{K} \sum_{i=1}^{K} \log \left( \frac{1}{K} \sum_{j=1}^{K} e^{f(x_i, y_j)} \right). \tag{1}$$

Written in this form, notice that the second term of $\hat{\mathrm{I}}_{\mathrm{NCE}}$ is permutation-invariant: changing each $y_j$ with $y_{\sigma_j}$ for some permutation $\sigma$ changes the value of the first term, but does not change the value of the second. Put another way, the test statistic and each of its permuted versions are $\frac{1}{K} \sum_{i=1}^{K} f(x_i, y_i)$ shifted by the same constant. Thus, although this second term plays a vital role in selecting the critic function $f$, at test time the only thing that matters is whether the mean value of $f(x, y)$ is higher for the true pairings than for random pairs. We call this term the *neural dependency statistic* (NDS) and denote it for a batch of $K$ paired samples $\mathbb{X} = (x_1, ..., x_K)$ and $\mathbb{Y} = (y_1, ..., y_K)$ by

$$\hat{T}(\mathbb{X}, \mathbb{Y}) = \frac{1}{K} \sum_{i=1}^{K} f(x_i, y_i). \tag{2}$$

The same is true for the NWJ (Nguyen et al., 2010), DV (Donsker & Varadhan, 1983), $\mathrm{I}_{\mathrm{JS}}$, and $\mathrm{I}_\alpha$ lower bounds discussed by Poole et al. (2019), as well as the MINE estimator (Belghazi et al., 2018); that is, at test time only the NDS statistic (2) matters.

## 2.1 Asymptotic test power

Typically, a mutual information lower bound is considered better if the population value of the bound is larger: that makes it a tighter bound. This viewpoint, however, neglects the issue of statistical estimation of that bound, e.g. the difference between $\mathrm{I}_{\mathrm{NCE}}^{f,K}$ and $\hat{\mathrm{I}}_{\mathrm{NCE}}^{f,K}$. The best statistical test, among tests with appropriate Type I (false rejection) control, is the one with highest *test power*: the probability of correctly rejecting the null hypothesis $\mathfrak{H}_0$ when $X \not\perp Y$. Taking this into account, we examine the behavior of a permutation test based on (1) or the many other mutual information bounds based only on the NDS (2). Each bound will choose a different $f$ during training, but at test time only the NDS matters.

Let $T_{\mathfrak{H}_0} = \mathbb{E}_{\mathbb{P}_x \times \mathbb{P}_y} f(x, y)$ and $T_{\mathfrak{H}_1} = \mathbb{E}_{\mathbb{P}_{xy}} f(x, y)$ be the population level statistics under the null and alternative distributions. Assuming a fixed critic $f$ is chosen independently from test samples $\mathbb{X}$ and $\mathbb{Y}$ (e.g. because of sample splitting), and that $0 < \tau_{\mathfrak{H}_1}^2 = \mathrm{Var}_{\mathbb{P}_{xy}} f(x, y) < \infty$, the central limit theorem implies that $\frac{1}{\tau_{\mathfrak{H}_1}} \sqrt{m}(\hat{T} - T_{\mathfrak{H}_1}) \xrightarrow{\mathrm{d}} \mathcal{N}(0, 1)$. Then, using $\Phi$ for the standard normal cdf, the rejection threshold $r_m$ satisfies

$$\alpha = \Pr_{\mathfrak{H}_0} \left( \sqrt{m} \hat{T} > r_m \right) = \Pr_{\mathfrak{H}_0} \left( \sqrt{m} \frac{\hat{T} - T_{\mathfrak{H}_0}}{\tau_{\mathfrak{H}_0}} > \frac{r_m - \sqrt{m} T_{\mathfrak{H}_0}}{\tau_{\mathfrak{H}_0}} \right) = 1 - \Phi \left( \frac{r_m - \sqrt{m} T_{\mathfrak{H}_0}}{\tau_{\mathfrak{H}_0}} \right) + o(1).$$

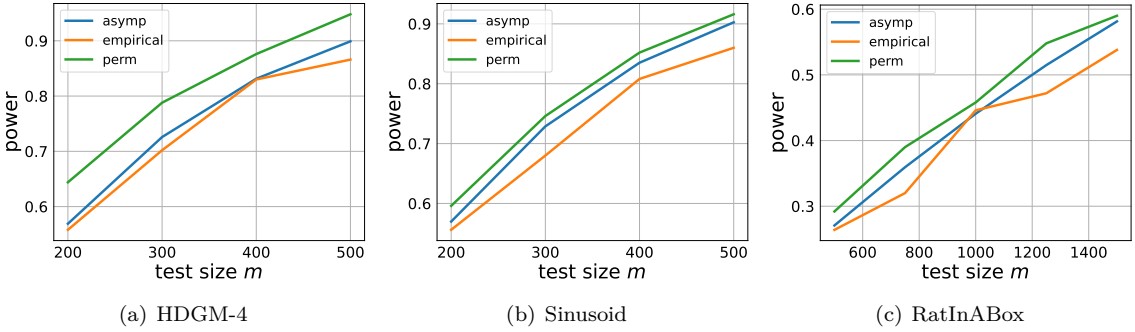

(a) HDGM-4        (b) Sinusoid        (c) RatInABox

Figure 2: Power of NDS tests with a particular critic $f$ (learned by the approach of Section 5) on three datasets described in Section 6. For each problem, the blue line (asymp) is power described by the asymptotic formula (3) with the $o(1)$ error terms set to zero, where we use $20\,000$ samples to estimate the population-level statistics $T_{\mathfrak{H}_1}, T_{\mathfrak{H}_0}$ and their standard deviations $\tau_{\mathfrak{H}_1}, \tau_{\mathfrak{H}_0}$. The orange line (empirical) computes the power based on the simulated null distribution, i.e. the threshold estimated via the empirical CDF of $\widehat{T}_0$ with test size $m$. The green line (perm) is the permutation test power. All three lines roughly agree with each other.

As $\Phi$ is Lipschitz, this implies $r_m = \sqrt{m} T_{\mathfrak{H}_0} + \tau_{\mathfrak{H}_0}\big(\Phi^{-1}(1 - \alpha) + o(1)\big)$. Using the asymptotic normality of $\hat{T}$ under $\mathfrak{H}_1$, we then have that

$$
\begin{aligned}
\Pr_{\mathfrak{H}_1}\left(\sqrt{m}\hat{T} > r_m\right) &= \Pr_{\mathfrak{H}_1}\left(\sqrt{m}\frac{\hat{T} - T_{\mathfrak{H}_1}}{\tau_{\mathfrak{H}_1}} > \frac{r_m - \sqrt{m} T_{\mathfrak{H}_1}}{\tau_{\mathfrak{H}_1}}\right) \\
&= 1 - \Phi\left(\frac{\sqrt{m} T_{\mathfrak{H}_0} + \tau_{\mathfrak{H}_0}\big(\Phi^{-1}(1 - \alpha) + o(1)\big) - \sqrt{m} T_{\mathfrak{H}_1}}{\tau_{\mathfrak{H}_1}}\right) + o(1) \\
&= \Phi\left(\sqrt{m}\frac{T_{\mathfrak{H}_1} - T_{\mathfrak{H}_0}}{\tau_{\mathfrak{H}_1}} - \frac{\tau_{\mathfrak{H}_1}}{\tau_{\mathfrak{H}_0}}\big(\Phi^{-1}(1 - \alpha) + o(1)\big)\right) + o(1).
\end{aligned}
\tag{3}
$$

To confirm that these asymptotics give a reasonable approximation in finite sample sizes, we can check that the asymptotic test power (3) for a given critic function $f$ roughly agrees with the empirical test power using a non-asymptotic rejection threshold estimated by obtaining fresh samples from the null distribution, i.e. the empirical quantile of $\hat{T}$ under the null. (This would not be available in practical situations, but is in synthetic problems where we can obtain arbitrary numbers of fresh samples.) Figure 2 indeed confirms that (3) gives a good estimate of the empirical power even at moderate sample sizes. (Intriguingly, the power of a permutation test is consistently slightly higher than the test based on the correct non-asymptotic threshold; we will discuss this issue in Section 7.)

This argument shows that, as long as $T_{\mathfrak{H}_1} > T_{\mathfrak{H}_0}$ and $\hat{T}$ has finite positive variance under both $\mathbb{P}_{xy}$ and $\mathbb{P}_x \times \mathbb{P}_y$, a test based on $\hat{T}$ will almost surely eventually reject any fixed alternative: the test is consistent. How quickly in $m$ it reaches high power, however, depends on the expression (3), which in particular for large $m$ is dominated by the signal-to-noise ratio (SNR) given by $(T_{\mathfrak{H}_1} - T_{\mathfrak{H}_0})/\tau_{\mathfrak{H}_1}$. When we choose a test based on maximizing the value of a mutual information lower bound, we maximize a criterion, e.g., (1), which does not directly correspond to the test power. We will discuss instead maximizing the asymptotic power in Section 5.

## 3 Tests with the Hilbert-Schmidt Independence Criterion (HSIC)

The Hilbert-Schmidt Independence Criterion (HSIC, Gretton et al., 2005) is also zero if and only if $X$ and $Y$ are independent when an appropriate kernel is chosen (Szabó & Sriperumbudur (2018) give precise conditions). Unlike the mutual information, HSIC is easy to estimate from samples; for a fixed bounded kernel, the typical estimators concentrate to the population value with deviation $\mathcal{O}_p(1/\sqrt{m})$.

To define HSIC, we first briefly review positive-definite kernels. A (real-valued) *kernel* is a function $k : \mathcal{X} \times \mathcal{X} \to \mathbb{R}$ that can be expressed as the inner product between feature maps $\phi : \mathcal{X} \to \mathcal{F}$, $k(x, x') = \langle \phi(x), \phi(x') \rangle_{\mathcal{F}}$, where $\mathcal{F}$ is any Hilbert space. An important special case is $\mathcal{F} = \mathbb{R}^p$ and $k(x, x') = \phi(x) \cdot \phi(x')$, where $\phi(x)$ extracts $p$-dimensional features of $x$. For every kernel function, there exists a unique *reproducing kernel Hilbert space* (RKHS), which consists of functions $f : \mathcal{X} \to \mathbb{R}$. The key *reproducing property* of an RKHS $\mathcal{F}$ states that for any function $f \in \mathcal{F}$ and any point $x \in \mathcal{X}$, we have $\langle f, \phi(x) \rangle_{\mathcal{F}} = f(x)$.

Suppose we have a kernel $k$ on $\mathcal{X}$ with RKHS $\mathcal{F}$ and feature map $\phi$, as well as another kernel $l$ on $\mathcal{Y}$ with RKHS $\mathcal{G}$ and feature map $\psi$. Let $\otimes$ denote the outer product.[1] The *cross-covariance operator* is

$$C_{xy} = \mathbb{E}_{xy} \Big[ \big( \phi(x) - \mathbb{E}_x \phi(x) \big) \otimes \big( \psi(y) - \mathbb{E}_y \psi(y) \big) \Big];$$

for kernels with finite-dimensional feature maps, this is exactly the standard (cross-)covariance matrix between the features of $X$ and those of $Y$. Under mild integrability conditions on the kernel and the distributions,[2] the reproducing property shows that $\langle f, C_{xy} g \rangle = \mathrm{Cov}(f(X), g(Y))$ for all $f \in \mathcal{F}$, $g \in \mathcal{G}$. One definition of independence is whether there exist any correlated "test functions" $f$ and $g$. Thus, for rich enough choices of kernel – using universal $k$ and $l$ suffices, but is not necessary (Szabó & Sriperumbudur, 2018) – we have that $X \perp\!\!\!\perp Y$ if and only if the operator $C_{xy} = 0$. We can thus check whether the operator is zero, and hence whether $X \perp\!\!\!\perp Y$, by checking the squared Hilbert-Schmidt norm of $C_{xy}$, $\mathrm{HSIC}(X, Y) = \|C_{xy}\|_{\mathrm{HS}}^2$. With finite-dimensional features, this is the squared Frobenius norm of the feature cross-covariance matrix.

Another way to interpret HSIC is as a distance between $\mathbb{P}_{xy}$ and $\mathbb{P}_x \times \mathbb{P}_y$, similarly to how the mutual information is the KL divergence between those same distributions.

**Proposition 3.1** (Gretton et al., 2012a, Theorem 25). *Let $k$ and $l$ be kernels on $\mathcal{X}$ and $\mathcal{Y}$, and define a kernel on $\mathcal{X} \times \mathcal{Y}$ by $h\big((x, y), (x', y')\big) = k(x, x') l(y, y')$ with RKHS $\mathcal{H}$. Then*

$$\sqrt{\mathrm{HSIC}_{k,l}(X, Y)} = \mathrm{MMD}_h(\mathbb{P}_{xy}, \mathbb{P}_x \times \mathbb{P}_y) = \sup_{\substack{f \in \mathcal{H} \\ \|f\|_{\mathcal{H}} \leq 1}} \mathbb{E}_{(X,Y) \sim \mathbb{P}_{xy}}[f(X, Y)] - \mathbb{E}_{\substack{X \sim \mathbb{P}_x \\ Y' \sim \mathbb{P}_y}}[f(X, Y')]$$

$$= \sqrt{ \mathbb{E}_{\substack{(X,Y),(X',Y') \sim \mathbb{P}_{xy} \\ Y'', Y''' \sim \mathbb{P}_y}} \Big[ k(X, X') l(Y, Y') - 2k(X, X') l(Y, Y'') + k(X, X') l(Y'', Y''') \Big] }.$$

Taking the last form and rearranging to save repeated computation yields two similar, popular estimators of HSIC. "The biased estimator" is $\mathrm{HSIC}(\hat{P}_{xy}, \hat{P}_x \times \hat{P}_y)$ for empirical distributions $\hat{P}$:

$$\widehat{\mathrm{HSIC}}_{\mathrm{b}}(\mathbb{X}, \mathbb{Y}) = \frac{1}{m^2} \langle \mathbf{K}, \mathbf{HLH} \rangle_F \qquad \text{for } \langle A, B \rangle_F = \sum_{ij} A_{ij} B_{ij}, \tag{4}$$

where here $\mathbf{K}$ is the $m \times m$ matrix with entries $k_{ij} = k(x_i, x_j)$, $\mathbf{L}$ similarly has entries $l_{ij} = l(y_i, y_j)$, and $\mathbf{H}$ is the "centering matrix" $\mathbf{I}_m - \frac{1}{m} \mathbf{1}_m \mathbf{1}_m^\top$ (so the estimator can be easily implemented without matrix multiplication). This estimator has $\mathcal{O}(1/m)$ bias, but is consistent.

The other common estimator, "the unbiased estimator," is a $U$-statistic (Song et al., 2012):

$$\widehat{\mathrm{HSIC}}_{\mathrm{u}}(\mathbb{X}, \mathbb{Y}) = \frac{1}{m(m-3)} \left[ \langle \tilde{\boldsymbol{K}}, \tilde{\boldsymbol{L}} \rangle_F + \frac{\mathbf{1}_m^\top \tilde{\boldsymbol{K}} \mathbf{1}_m \mathbf{1}_m^\top \tilde{\boldsymbol{L}} \mathbf{1}_m}{(m-1)(m-2)} - \frac{2 \mathbf{1}_m^\top \tilde{\boldsymbol{K}} \tilde{\boldsymbol{L}} \mathbf{1}_m}{m-2} \right], \tag{5}$$

where $\tilde{\boldsymbol{K}}$ and $\tilde{\boldsymbol{L}}$ are $m \times m$ matrices whose diagonal entries are zero but whose off-diagonal entries agree with those of $\mathbf{K}$ or $\mathbf{L}$. It is unbiased, $\mathbb{E}\, \widehat{\mathrm{HSIC}}_{\mathrm{u}}(\mathbb{X}, \mathbb{Y}) = \mathrm{HSIC}(X, Y)$; it is also consistent, and can be computed in the same $\mathcal{O}(m^2)$ time as $\widehat{\mathrm{HSIC}}_{\mathrm{b}}$, without matrix multiplication. Either statistic can be used with a permutation test to construct an independence test with finite-sample validity, in the same way as described for mutual information bounds. The following subsection describes the behavior of such tests in the large-sample limit.

---

[1]The Euclidean outer product $ab^\top$ is a matrix with $[ab^\top]b' = a[b^\top b']$; in Hilbert spaces, $f \otimes g : \mathcal{G} \to \mathcal{F}$ has $[f \otimes g]g' = f \langle g, g' \rangle_{\mathcal{G}}$.

[2]It suffices that $\mathbb{E}[\sqrt{k(x,x) l(y,y)}] < \infty$; this is guaranteed regardless of the distribution when $k$, $l$ are bounded.

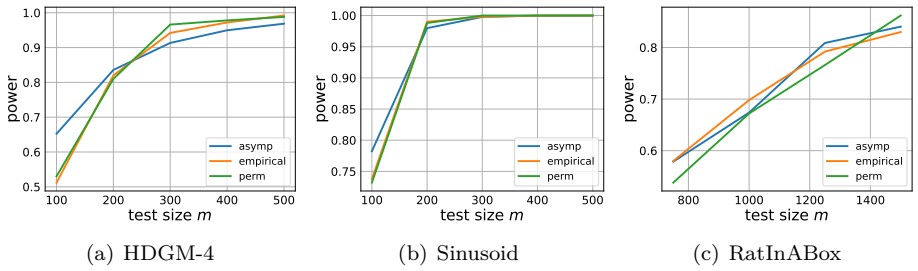

(a) HDGM-4      (b) Sinusoid      (c) RatInABox

Figure 3: Power of HSIC tests based on kernels $k$ and $l$ learned with the method of Section 5, evaluated on problems described in Section 6. For each problem, the blue line (asymp) is power described by the asymptotic formula (6), where we use 10 000 samples to estimate the population level statistics HSIC, its deviation $\sigma_{\mathfrak{H}_1}$, and the threshold $\Psi^{-1}(1-\alpha)$ via simulation. The orange line (empirical) computes the power based on the simulated null distribution, i.e. the threshold estimated via the empirical CDF of $\widehat{\text{HSIC}}_u$ with test size $m$. The green line (perm) is the permutation test power. All three lines roughly agree.

### 3.1 Asymptotic test power

The unbiased HSIC estimator is asymptotically normal with $\sqrt{m}(\widehat{\text{HSIC}}_{\text{u}} - \text{HSIC}) \overset{\text{d}}{\to} \mathcal{N}(0, \sigma^2_{\mathfrak{H}_1})$ (Proposition A.1, in Appendix A). It is possible, however, to have $\sigma^2_{\mathfrak{H}_1} = 0$, which makes that result less useful; this occurs in particular whenever HSIC $= 0$, and hence is always true under $\mathfrak{H}_0$. In that case, it is more informative to say instead that $m\,\widehat{\text{HSIC}}_{\text{u}}$ converges in distribution (which is not the case when $\sigma^2_{\mathfrak{H}_1} > 0$). The distribution to which it converges is a mixture of shifted chi-squareds with complex dependence on $\mathbb{P}_x$, $\mathbb{P}_y$, $k$, and $l$ (Proposition A.2). Thus, if we consider a test statistic $m\,\widehat{\text{HSIC}}_{\text{u}}$, when HSIC $= 0$ this statistic has mean zero and standard deviation $\Theta(1)$. When HSIC $> 0$, though, the statistic has mean and standard deviation $\Theta(\sqrt{m}) \to \infty$. Thus, as $m \to \infty$, eventually the test will reject if HSIC $> 0$ (Rindt et al., 2021).

Recall that the optimal test is one with highest test power among those with appropriate level control. As for NDS-based tests, we can describe this power asymptotically. Let $\Psi$ be the cdf of the null distribution of $m\,\widehat{\text{HSIC}}_{\text{u}}$, which depends on $k$, $l$, $\mathbb{P}_x$, and $\mathbb{P}_y$. It follows that the rejection threshold $r_m$ satisfies

$$\alpha = \Pr_{\mathfrak{H}_0}(\widehat{\text{HSIC}}_{\text{u}} > r_m) = 1 - \Psi(m r_m) + o(1) \qquad \text{so} \qquad r_m = \frac{1}{m}\big(\Psi^{-1}(1-\alpha) + o(1)\big).$$

Using Proposition A.1, when $\sigma_{\mathfrak{H}_1} > 0$ the asymptotic test power then satisfies

$$\Pr_{\mathfrak{H}_1}\left(\widehat{\text{HSIC}}_{\text{u}} > r_m\right) = \Pr_{\mathfrak{H}_1}\left(\sqrt{m}\frac{\widehat{\text{HSIC}}_{\text{u}} - \text{HSIC}}{\sigma_{\mathfrak{H}_1}} > \sqrt{m}\frac{r_m - \text{HSIC}}{\sigma_{\mathfrak{H}_1}}\right)$$

$$= 1 - \Phi\left(\sqrt{m}\frac{\frac{1}{m}\big(\Psi^{-1}(1-\alpha) + o(1)\big) - \text{HSIC}}{\sigma_{\mathfrak{H}_1}}\right) + o(1)$$

$$= \Phi\left(\frac{\sqrt{m}\,\text{HSIC}}{\sigma_{\mathfrak{H}_1}} - \frac{\Psi^{-1}(1-\alpha) + o(1)}{\sqrt{m}\,\sigma_{\mathfrak{H}_1}}\right) + o(1). \tag{6}$$

Figure 3 verifies that this expression lines up well with the empirical test power even at moderate test sizes. Additionally, (6) also tells us that, as long as HSIC $> 0$ and $\widehat{\text{HSIC}}_{\text{u}}$ has finite positive variance, an HSIC-based test will eventually reject with probability of one. The rate at which it increases in power is described by the gap between the signal-to-noise ratio $\text{HSIC}/\sigma_{\mathfrak{H}_1}$ and a threshold-dependent term $\Psi^{-1}(1-\alpha)/\sigma_{\mathfrak{H}_1}$, with the former dominating the latter. This gap increases at a faster rate than the one described for NDS-based tests (3); the ratio of the two terms is $\Theta(m)$ for HSIC, while it is only $\Theta(\sqrt{m})$ for NDS. (Appendix B gives another motivation for the power being dominated by this signal-to-noise ratio, without using the central limit theorem.)

## 3.2 MMD-based independence test

Given that more attention has been recently paid to learning two-sample tests than dependence tests, we can consider reformulating the independence problem into a two-sample one: are $\mathbb{P}_{xy}$ and $\mathbb{P}_x \times \mathbb{P}_y$ the same distribution? A natural measure of difference between distributions $\mathbb{P}_{xy}$ and $\mathbb{P}_x \times \mathbb{P}_y$ is $\mathrm{MMD}^2(\mathbb{P}_{xy}, \mathbb{P}_x \times \mathbb{P}_y)$ with some kernel $h$, which becomes $\mathrm{HSIC}(X, Y)$ if $h$ is a product kernel. For sufficiently powerful $h$, as characterized by Szabó & Sriperumbudur 2018, $\mathrm{MMD}(\mathbb{P}_{xy}, \mathbb{P}_x \times \mathbb{P}_y) = 0$ if and only if $X \perp\!\!\!\perp Y$. Thus it makes sense to use any consistent MMD estimator as the test statistic; we use the biased U-statistic estimator described in Gretton et al. (2012a) for simplicity.

Next we consider a testing procedure. The standard independence problem observes paired samples $\mathbb{Z} = \big((x_1, y_1), \ldots, (x_m, y_m)\big)$ drawn i.i.d. from the joint distribution $\mathbb{P}_{xy}$, and so to simulate samples from $\mathbb{P}_x \times \mathbb{P}_y$ we first shuffle the $Y$ samples by some permutation $\sigma$ on $\{1, ..., m\}$. We define this action of permuting $\mathbb{Z}$ with $\sigma$ as $\sigma\mathbb{Z} = \big((x_1, y_{\sigma(1)}), \ldots, (x_m, y_{\sigma(m)})\big)$. These samples are then used to compute $\widehat{\mathrm{MMD}}_b^2(\mathbb{Z}, \sigma\mathbb{Z})$. The following proposition shows that, with enough samples, this statistic is a consistent estimator of the true MMD value.

**Proposition 3.2.** *Suppose $h$ satisfies $\sup_{x \in \mathcal{X}, y \in \mathcal{Y}} h((x, y), (x, y)) \leq \nu^2$. The estimator $\widehat{\mathrm{MMD}}_b^2(\mathbb{Z}, \sigma\mathbb{Z})$, where $\mathbb{Z}$ is independent of a uniformly random $\sigma$, satisfies with probability at least $1 - \delta$ that*

$$|\widehat{\mathrm{MMD}}_b^2(\mathbb{Z}, \sigma\mathbb{Z}) - \mathrm{MMD}^2(\mathbb{P}_{xy}, \mathbb{P}_x \times \mathbb{P}_y)| \leq \frac{17\nu^2}{\sqrt{m}} \left[ \sqrt{2 \log \frac{8}{\delta}} + \frac{1}{\sqrt{m}} \right].$$

This result is proved in Appendix F.4.

Viewing this permuted MMD estimator as a function of the original paired samples, i.e. $f(\mathbb{Z}) := \widehat{\mathrm{MMD}}_b^2(\mathbb{Z}, \sigma\mathbb{Z})$, we then perform an independence permutation test by shuffling only the $Y$ samples in $\mathbb{Z}$. Under the null hypothesis the distribution of $\mathbb{Z}$ is unchanged by shuffling the $Y$ samples, and so an independence permutation test on $f(\mathbb{Z})$ is guaranteed by Theorem 2 of Hemerik & Goeman (2018) to have correct level control, provided the original sample order is included in the permutations.[3]

It's also worth noting that, since the MMD estimator is a two-sample statistic, the number of samples of $\mathbb{P}_{xy}$ and $\mathbb{P}_x \times \mathbb{P}_y$ can be different. As such, we can also consider using multiple permutations $\boldsymbol{\sigma} = \{\sigma_1, ..., \sigma_p\}$ rather than just one, and then simulate the null samples as $\boldsymbol{\sigma}\mathbb{Z} = \bigcup \sigma_i \mathbb{Z}$. It turns out that if we consider all rotation permutations, $\widehat{\mathrm{MMD}}_b^2(\mathbb{Z}, \boldsymbol{\sigma}\mathbb{Z})$ is exactly the biased HSIC estimator $\widehat{\mathrm{HSIC}}_b(\mathbb{X}, \mathbb{Y})$. We elaborate on this connection in Appendix F.4 and show that using more permutations improves estimation quality. The HSIC estimator, which effectively uses all permutations, is the minimum variance estimator for this permuted MMD statistic (Proposition F.2).

# 4 Connecting HSIC and Mutual Information Tests

## 4.1 HSIC as a lower bound on MI

Suppose the kernel $h$ on $(x, y)$ pairs of Proposition 3.1 is bounded: $\sup_{x \in \mathcal{X}, y \in \mathcal{Y}} h((x, y), (x, y)) \leq \nu^2$ for $\nu \geq 0$. Then

$$|f(x, y)| = |\langle f, \phi_h(x, y) \rangle_{\mathcal{H}}| \leq \|f\|_{\mathcal{H}} \|\phi_h(x, y)\|_{\mathcal{H}} \leq \nu \|f\|_{\mathcal{H}}$$

---

[3]One might wonder whether we can instead perform a two-sample permutation test. Consider the test statistic as a function of the pooled samples $f([\mathbb{Z}, \sigma\mathbb{Z}]) = \widehat{\mathrm{MMD}}_b^2(\mathbb{Z}, \sigma\mathbb{Z})$. The two-sample permutation test involves shuffling the order of $\mathbb{Z} \cup \sigma\mathbb{Z}$, however, exchangeability of $\mathbb{Z} \cup \sigma\mathbb{Z}$ under the null is not guaranteed as variables in the first split are deterministically tied to variables in the second; thus a two-sample permutation test may not be valid. A workaround would be to independently split the data into $\mathbb{Z}_1$ and $\mathbb{Z}_2$ and then construct the pooled sample as $\mathbb{Z}_1 \cup \sigma\mathbb{Z}_2$, but this process is more involved and potentially lower-power.

by the reproducing property and Cauchy-Schwarz, so $\|f\|_{\mathcal{H}} \geq \frac{1}{\nu} \sup_{x,y} |f(x,y)| = \frac{1}{\nu} \|f\|_{\infty}$. This implies that $\{f : \|f\|_{\mathcal{H}} \leq 1\} \subseteq \{f : \|f\|_{\infty} \leq \nu\}$, and so by Proposition 3.1,

$$\sqrt{\mathrm{HSIC}(X,Y)} \leq \sup_{f:\|f\|_{\infty} \leq \nu} \mathop{\mathbb{E}}_{(X,Y) \sim \mathbb{P}_{xy}} [f(X,Y)] - \mathop{\mathbb{E}}_{\substack{X \sim \mathbb{P}_x \\ Y' \sim \mathbb{P}_y}} [f(X,Y')] = 2\nu \, \mathrm{TV}(\mathbb{P}_{xy}, \mathbb{P}_x \times \mathbb{P}_y), \tag{7}$$

where TV is the total variation distance between distributions (Sriperumbudur et al., 2012).[4] Applying standard bounds relating the total variation to the KL divergence, we obtain the following.

**Proposition 4.1.** *In the setting of Proposition 3.1, suppose* $\sup_{x \in \mathcal{X}, y \in \mathcal{Y}} h((x,y),(x,y)) \leq \nu^2$. *Then*

$$\frac{1}{2\nu^2} \mathrm{HSIC}(X,Y) \leq \mathrm{I}(X;Y) \qquad and \qquad -\log\left(1 - \frac{1}{4\nu^2} \mathrm{HSIC}(X,Y)\right) \leq \mathrm{I}(X;Y).$$

*Proof.* The first bound applies Pinsker's inequality, which relates total variation to KL, to (7). The second instead applies the bound of Bretagnolle & Huber (1978) (also see Canonne, 2022). □

The second bound is tighter for large values of $\mathrm{I}(X;Y)$, but both are monotonic in $\mathrm{HSIC}(X,Y)/\nu^2$. We could thus consider, as in Section 2, choosing kernels $k$, $l$ to maximize a lower bound on $\mathrm{I}(X;Y)$, by maximizing $\mathrm{HSIC}(X,Y)/\nu^2$. Indeed, maximizing HSIC has been used by previous applications in many areas (e.g. Blaschko & Gretton, 2009; Song et al., 2012; Li et al., 2021; Dong et al., 2023).

### 4.2 Kernel-based tests and variational MI tests

Now consider a test based on $\mathrm{MMD}_f(\mathbb{P}_{xy}, \mathbb{P}_x \times \mathbb{P}_y)$ using a kernel of the form $h((x,y),(x',y')) = f(x,y)f(x',y')$ for some real-valued function $f$. If $f(x,y) = f_1(x)f_2(y)$, this is an HSIC test with kernels $k(x,x') = f_1(x)f_1(x')$ and $l(y,y') = f_2(y)f_2(y')$. Because $\phi_h(x,y) = f(x,y) \in \mathbb{R}$ is a valid feature map, every function in $\mathcal{H}$ is of the form $\alpha f = [(x,y) \mapsto \alpha f(x,y)]$ with $\|\alpha f\|_{\mathcal{H}} = |\alpha|$. By Proposition 3.1,

$$\mathrm{MMD}_f(\mathbb{P}_{xy}, \mathbb{P}_x \times \mathbb{P}_y)^2 = \left(\sup_{|\alpha| \leq 1} \alpha \left(\mathop{\mathbb{E}}_{\mathbb{P}_{xy}} f(X,Y) - \mathop{\mathbb{E}}_{\mathbb{P}_x \times \mathbb{P}_y} f(X,Y')\right)\right)^2 = \left(\mathop{\mathbb{E}}_{\mathbb{P}_{xy}} f(X,Y) - \mathop{\mathbb{E}}_{\mathbb{P}_x \times \mathbb{P}_y} f(X,Y')\right)^2.$$

The plug-in estimator would yield the test statistic

$$\widehat{\mathrm{MMD}}_{\mathrm{b}}^2(\mathbb{X}, \mathbb{Y}) = \left(\frac{1}{m} \sum_{i=1}^{m} f(x_i, y_i) - \frac{1}{m^2} \sum_{i=1}^{m} \sum_{j=1}^{m} f(x_i, y_j)\right)^2, \tag{8}$$

which when $f(x,y) = f_1(x)f_2(y)$, corresponds exactly to $\widehat{\mathrm{HSIC}}_{\mathrm{b}}$.

Comparing (8) to (1) with $K = m$, we can see that the main term $\hat{T} = \frac{1}{m} \sum_i f(x_i, y_i)$ is identical, which we called the NDS (2). The other term is permutation-invariant; it is the mean of $\hat{T}$ over all possible permutations, $\bar{T}$. Thus, a permutation test based on (8) asks how far the value of $\hat{T}$ for the true data is from $\bar{T}$, while a test based on any variational MI estimator based on (2) is equivalent to asking how much the value of $\hat{T}$ exceeds $\bar{T}$. The only difference is that (8) gives a two-sided test, while (1) is a one-sided test.[5]

Our usual test uses $\widehat{\mathrm{HSIC}}_{\mathrm{u}}$ of (5) instead of $\widehat{\mathrm{HSIC}}_{\mathrm{b}}$, but the difference in estimators is typically small. Thus, if we use deep kernels of the form $k(x,x') = f(x)f(x')$ and $l(y,y') = g(y)g(y')$, obtaining a test quite closely related to a witness two-sample test (Kübler et al., 2022) used for independence, the HSIC test is nearly equivalent to the NCE test with a separable critic function $(x,y) \mapsto f(x)g(y)$. This relationship is roughly analogous to the relationship between classifier two-sample tests and MMD tests observed by Liu et al. (2020): while each test chooses a critic/kernel in a different way, at test time they are essentially equivalent.

---

[4] Sriperumbudur et al. (2012) define the TV as twice the more common definition, which we use here.
[5] Li et al. (2021, Section 3.1) also found a relationship between HSIC and $\mathrm{I}_{\mathrm{NCE}}$ for categorical $Y$.

# 5 Learning Representations for Independence Testing

Choosing a test based on maximizing the value of its statistic does not directly correspond to the test power. Instead, we would perhaps be better served by maximizing the power directly. Recall the asymptotic power expressions for NDS and HSIC-based tests given by (3) and (6), respectively:

$$\Pr_{\mathfrak{H}_1}(\hat{T} > r_m) \approx \Phi\left(\frac{\sqrt{m}(T - T_{\mathfrak{H}_0})}{\tau_{\mathfrak{H}_1}} - \frac{\tau_{\mathfrak{H}_0}}{\tau_{\mathfrak{H}_1}}\Phi^{-1}(1-\alpha)\right) \text{ and } \Pr_{\mathfrak{H}_1}(\widehat{\text{HSIC}}_{\text{u}} > r_m) \approx \Phi\left(\frac{\sqrt{m}\,\text{HSIC}}{\sigma_{\mathfrak{H}_1}} - \frac{\Psi^{-1}(1-\alpha)}{\sqrt{m}\sigma_{\mathfrak{H}_1}}\right).$$

In both cases, as the sample size $m$ grows, the power is dominated by the signal-to-noise ratio

$$\text{SNR}[T] = (T - T_{\mathfrak{H}_0})/\tau_{\mathfrak{H}_1} \qquad \text{and} \qquad \text{SNR}[\text{HSIC}] = \text{HSIC}\,/\sigma_{\mathfrak{H}_1}. \tag{9}$$

Maximizing the SNR maximizes the limiting power of the test as $m \to \infty$. (Also see Appendix B.) Thus, maximizing estimates of the SNR (based on a finite number of samples $n$) will roughly maximize the power of a test at an arbitrarily large sample size ($m \to \infty$); also see discussion by Sutherland & Deka (2019); Deka & Sutherland (2023).

Alternatively, we can also choose to include the threshold-dependent term. This may be helpful, particularly in estimating the power at points where the number of samples is not overwhelming: for tests with a 50% probability of rejection, the two terms in the power expression are necessarily the same size. For NDS, the gap between the two terms only grows with $\sqrt{m}$, and additionally estimating $\tau_{\mathfrak{H}_1}$ is no problem when already estimating $T_{\mathfrak{H}_1}$, thus we propose to include the threshold term in choosing an NDS critic. For HSIC, however, the threshold-dependent term is both relatively less important as $m$ grows and much more difficult to estimate.[6] Therefore, we propose to optimize the SNR with the threshold term for NDS, and without it for HSIC. Our population level objectives, as functions of random variables $X$ and $Y$, are then

$$J_{\text{NDS}}(X, Y; f) = \frac{T(X, Y; f) - T_{\mathfrak{H}_0}(X, Y; f)}{\tau_{\mathfrak{H}_1}(X, Y; f)} - \frac{\tau_{\mathfrak{H}_0}(X, Y; f)}{\sqrt{m}\,\tau_{\mathfrak{H}_1}(X, Y; f)}\Phi^{-1}(1-\alpha) \tag{10}$$

and

$$J_{\text{HSIC}}(X, Y; k, l) = \frac{\text{HSIC}(X, Y; k, l)}{\sigma_{\mathfrak{H}_1}(X, Y; k, l)}. \tag{11}$$

Then, if we have a batch of paired observations $\mathbb{X} = (X_1, ..., X_m)$ and $\mathbb{Y} = (Y_1, ..., Y_m)$ sampled jointly from $\mathbb{P}_{XY}$, we can approximate $J_{\text{NDS}}$ and $J_{\text{HSIC}}$ by the estimators $\hat{J}^\lambda_{\text{NDS}}$ and $\hat{J}^\lambda_{\text{HSIC}}$, which we define below.

**NDS objective.** We define $\hat{J}^\lambda_{\text{NDS}}$ as the plug-in estimator of $J_{\text{NDS}}$ with variance regularization $\lambda > 0$. We estimate $T_{\mathfrak{H}_1} = \mathbb{E}_{\mathbb{P}_{XY}}[f(X, Y)]$ by its sample mean $\hat{T}(\mathbb{X}, \mathbb{Y}) = \frac{1}{m}\sum_{i=1}^m f(X_i, Y_i)$, and $T_{\mathfrak{H}_0} = \mathbb{E}_{\mathbb{P}_X \times \mathbb{P}_Y} f(X, Y)$ by the V-statistic $\hat{T}_{\mathfrak{H}_0}(\mathbb{X}, \mathbb{Y}) = \frac{1}{m^2}\sum_{i=1}^m \sum_{j=1}^m f(X_i, Y_j)$. The variance terms $\tau^2_{\mathfrak{H}_0}$ and $\tau^2_{\mathfrak{H}_1}$ are estimated with a regularized version of the sample variance,

$$\hat{\tau}^2_{\mathfrak{H}_0, \lambda}(\mathbb{X}, \mathbb{Y}; f) = \frac{1}{m^2}\sum_{i=1}^m \sum_{j=1}^m \left(f(X_i, Y_j) - \frac{1}{m^2}\sum_{i'=1}^m \sum_{j'=1}^m f(X_{j'}, Y_{j'})\right)^2 + \lambda$$

$$\hat{\tau}^2_{\mathfrak{H}_1, \lambda}(\mathbb{X}, \mathbb{Y}; f) = \frac{1}{m}\sum_{i=1}^m \left(f(X_i, Y_i) - \frac{1}{m}\sum_{j=1}^m f(X_j, Y_j)\right)^2 + \lambda,$$

where $\lambda > 0$. Putting it all together, our complete NDS objective is

$$\hat{J}^\lambda_{\text{NDS}}(\mathbb{X}, \mathbb{Y}; f) = \frac{\hat{T}(\mathbb{X}, \mathbb{Y}; f) - \hat{T}_0(\mathbb{X}, \mathbb{Y}; f)}{\hat{\tau}_{\mathfrak{H}_1, \lambda}(\mathbb{X}, \mathbb{Y}; f)} - \frac{\hat{\tau}_{\mathfrak{H}_0}(\mathbb{X}, \mathbb{Y}; f)}{\sqrt{m}\,\hat{\tau}_{\mathfrak{H}_1}(\mathbb{X}, \mathbb{Y}; f)}\Phi^{-1}(1-\alpha). \tag{12}$$

---

[6] Ren et al. (2024) proposed using a moment-matched gamma approximation to the threshold, and claimed that not including this threshold can lead to catastrophically wrong kernel choice. We argue in Appendix F.5, however, that their argument is unjustified. We also find experimentally in Section 6 that including it can choose worse kernels in practice. More common contemporary approaches to estimating the threshold use permutation testing or eigendecomposition; both involve substantial computational overhead, and the latter is particularly expensive to differentiate while the former can be quite difficult to usefully estimate on the same data as the HSIC estimate, as observed for MMD by Deka & Sutherland (2023) and discussed further in Section 7.

**HSIC objective.** $\hat{J}_{\text{HSIC}}^\lambda$ is defined in a similar manner. We estimate HSIC by the unbiased estimator $\widehat{\text{HSIC}}_u(\mathbb{X}, \mathbb{Y})$ given in (5), and we approximate the variance term $\sigma^2_{\mathfrak{H}_1,\lambda}$ according to

$$\hat{\sigma}^2_{\mathfrak{H}_1,\lambda}(\mathbb{X}, \mathbb{Y}; k, l) = 16(R - \widehat{\text{HSIC}}_{\text{u}}^2) + \lambda$$

which is just a regularized version of the variance estimate stated in Proposition A.1. Following Song et al. (2012), this can be computed more efficiently with $R = \frac{((n-4)!)^2}{4n((n-1)!)^2}\|\boldsymbol{h}\|^2$, where the vector $\boldsymbol{h}$ is

$$\begin{aligned}
\boldsymbol{h} = {} & (n-2)^2\left(\tilde{\boldsymbol{K}} \circ \tilde{\boldsymbol{L}}\right)\boldsymbol{1} - n(\tilde{\boldsymbol{K}}\boldsymbol{1}) \circ (\tilde{\boldsymbol{L}}\boldsymbol{1}) + (\boldsymbol{1}^\top\tilde{\boldsymbol{L}}\boldsymbol{1})\tilde{\boldsymbol{K}}\boldsymbol{1} + (\boldsymbol{1}^\top\tilde{\boldsymbol{K}}\boldsymbol{1})\tilde{\boldsymbol{L}}\boldsymbol{1} - (\boldsymbol{1}^\top\tilde{\boldsymbol{K}}\tilde{\boldsymbol{L}}\boldsymbol{1})\boldsymbol{1} \\
& + (n-2)\left((\boldsymbol{1}^\top(\tilde{\boldsymbol{K}} \circ \tilde{\boldsymbol{L}})\boldsymbol{1})\boldsymbol{1} - \tilde{\boldsymbol{K}}\tilde{\boldsymbol{L}}\boldsymbol{1} - \tilde{\boldsymbol{L}}\tilde{\boldsymbol{K}}\boldsymbol{1}\right),
\end{aligned}$$

with $\circ$ denoting elementwise multiplication on matrices, and $\boldsymbol{1} = (1, \ldots, 1) \in \mathbb{R}^n$. Thus, the complete HSIC objective is

$$\hat{J}_{\text{HSIC}}^\lambda(\mathbb{X}, \mathbb{Y}; k, l) = \frac{\widehat{\text{HSIC}}_{\text{u}}(\mathbb{X}, \mathbb{Y}; k, l)}{\hat{\sigma}_{\mathfrak{H}_1,\lambda}(\mathbb{X}, \mathbb{Y}; k, l)}. \tag{13}$$

Maximizing (12) or (13) then gives us a structured approach to select the critic or kernel yielding the strongest asymptotic test. Typically, we run some variant of a gradient-based optimization algorithm with respect to the parameters of the critic or kernel class, and do hyperparameter selection and early stopping based on the same objective on a validation set.

**Critic & kernel architecture.** We consider function families parameterized by deep networks, as this allows us to learn representations of the data that more efficiently capture dependency. For critics, this parameterization is straightforward and is typically some problem-specific architecture like multilayer perceptrons or convolutional networks. For kernels, we first incorporate a featurizer that acts on individual inputs $x$ and $y$, and is then fed to some standard kernel function. When our feature mapping is parameterized by deep networks, we get the class of deep kernels (Wilson et al., 2016), which have been successfully used in two-sample testing (Sutherland et al., 2017; Liu et al., 2020; 2021) and many other settings (e.g. Li et al., 2017; Arbel et al., 2018; Jean et al., 2018; Li et al., 2021; Gao et al., 2021). We use the following deep kernels for $X$ and $Y$:

$$\begin{aligned}
k_\omega(x, x') &= (1 - \epsilon_X)\,\kappa_X(f_\omega(x), f_\omega(x')) + \epsilon_X\,q_X(x, x') \\
l_\gamma(y, y') &= (1 - \epsilon_Y)\,\kappa_Y(g_\gamma(y), g_\gamma(y')) + \epsilon_Y\,q_Y(y, y').
\end{aligned}$$

Here $f_\omega$ and $g_\gamma$ are deep networks with parameters in $\omega$, $\gamma$, which extract relevant features from $\mathcal{X}$ or $\mathcal{Y}$ to a feature space $\mathbb{R}^D$. These features are then used inside a Gaussian kernel $\kappa$ on the space $\mathbb{R}^D$, to compute the baseline similarity between data points. We then take a convex combination of that kernel with a Gaussian kernel $q$ on the input space; the weight of this component is determined by a parameter $\epsilon \in (0, 1)$.[7] The lengthscale of $\kappa$ and $q$ as well as the mixture parameter $\epsilon$ are included in the overall parameters, $\omega$ or $\gamma$, and learned during the optimization process.

**Overall representation learning algorithm.** The overall procedure, written based on minibatch gradient ascent for simplicity, is shown in Algorithm 1. In practice, we use AdamW (Loshchilov & Hutter, 2019), and draw minibatches in epochs; experimental details are given in Appendix E.1. The randomized $p$-value is potentially conservative in the case of ties; breaking ties randomly can give a slight improvement when ties are present (Hemerik & Goeman, 2018).

**Time complexity.** Each training iteration is dominated by computing the objectives $\hat{J}_{\text{NDS}}^\lambda$ and $\hat{J}_{\text{HSIC}}^\lambda$. Suppose $K$ is the minibatch size. For NDS tests, if $E_Z$ is the cost of evaluating critic $f_\theta(x, y)$, then one training step costs $\mathcal{O}(K^2 E_Z)$. For HSIC tests, if $E_X$ and $E_Y$ are the costs of computing embeddings $f_\omega(x)$ and $f_\gamma(y)$, and $L$ the cost of computing $k_\omega(x, x')$ and $l_\gamma(y, y')$ given the embeddings, then each training

---

[7]Using $\epsilon > 0$ provides a "backup" to the deep kernel, perhaps giving some signal early in optimization when the deep kernel features are not yet useful, and guaranteeing that the overall kernel is characteristic.

---

**Algorithm 1** Independence testing with learned representations

---

**Input:** paired samples $S_Z = (S_X, S_Y)$; split the data into $S_Z^{\text{tr}} \cup S_Z^{\text{te}}$ with $(S_X^{\text{tr}}, S_Y^{\text{tr}}) \leftarrow S_Z^{\text{tr}}$; $(S_X^{\text{te}}, S_Y^{\text{te}}) \leftarrow S_Z^{\text{te}}$;
      model parameters $\theta \leftarrow \theta_0$ and test statistic $\mathcal{T}(X, Y; \theta)$; various hyperparameters like $\lambda \leftarrow 10^{-8}$, etc.

  # *Phase 1: train the parameters $\theta$ on $S_Z^{\text{tr}}$*
  **for** $T = 1, 2, \ldots, T_{\max}$ **do**
    $(\mathbb{X}, \mathbb{Y}) \leftarrow$ minibatch from $S_Z^{tr} = (S_X^{tr}, S_Y^{tr})$;
    $\hat{J}^\lambda(\mathbb{X}, \mathbb{Y}; \theta) \leftarrow$ compute SNR estimate;           # *as in Equation* (12) *or Equation* (13)
    $\theta \leftarrow \theta + \eta \nabla_\theta \hat{J}^\lambda(\mathbb{X}, \mathbb{Y}; \theta)$                             # *gradient ascent step*
  # *Phase 2: permutation test on $S_Z^{\text{te}}$ with learned representations*
  $perm_1 \leftarrow \mathcal{T}(S_X^{\text{te}}, S_Y^{\text{te}}; \theta)$                            # *evaluate test statistic*
  **for** $i = 2, \ldots, n_{perm}$ **do**
    $perm_i \leftarrow \mathcal{T}(S_X^{\text{te}}, \text{shuffle}(S_Y^{\text{te}}); \theta)$              # *no need to shuffle both sets*
  **Output:** $k_\omega$, $l_\gamma$, $perm_1$, $p$-value $\frac{1}{n_{perm}} \sum_{i=1}^{n_{perm}} \mathbb{1}(perm_i \geq perm_1)$

---

iteration costs $\mathcal{O}\left(KE_X + KE_Y + K^2L\right)$. Typically, for practical values of $K$, $E_X + E_Y \gg KL$, so this cost is "almost" linear in practice.[8]

**Theoretical analysis.** While neither $\hat{J}_{\text{NDS}}^\lambda$ nor $\hat{J}_{\text{HSIC}}^\lambda$ are biased estimates of the relevant population quantities,[9] the following Theorems 5.1 and 5.2 show that they are both uniformly bounded in probability.

**Theorem 5.1** (Uniform convergence of $\hat{J}_{\text{NDS}}^\lambda$). *Let $\{f_\theta : \theta \in \Theta\}$ be a critic family with parameter space $\Theta$ satisfying Assumptions (A) to (C) in Appendix C.2 which define a critic bound $B$, dimension $D$, and smoothness $L$. Suppose $\tau_\theta^2 \geq s^2$ for some positive $s$ under both $\mathfrak{H}_0$ and $\mathfrak{H}_1$, and $\lambda = \Theta(n^{-1/3})$. Then*

$$\sup_{\theta \in \Theta} \left| \hat{J}_{NDS}^\lambda(\theta) - J_{NDS}(\theta) \right| = \tilde{\mathcal{O}}_P\left( \frac{1}{s^2 n^{1/3}} \left[ \frac{B}{s} + B^2 L + B^3 \sqrt{D} \right] \right).$$

**Theorem 5.2** (Uniform convergence of $\hat{J}_{\text{HSIC}}^\lambda$). *Let $\{k_\omega : \omega \in \Omega\}$ and $\{l_\gamma : \gamma \in \Gamma\}$ be kernel families with parameter spaces $\Omega$ and $\Gamma$ satisfying Assumptions (A') to (C') in Appendix D.2 which define kernel dimensions $D_\Omega$, $D_\Gamma$ and smoothnesses $L_k$, $L_l$. Suppose $\sigma_{\omega,\gamma}^2 \geq s^2 > 0$ under $\mathfrak{H}_1$ and $\lambda = \Theta(n^{-1/3})$. Then*

$$\sup_{\omega \in \Omega, \gamma \in \Gamma} \left| \hat{J}_{HSIC}^\lambda(\omega, \gamma) - J_{HSIC}(\omega, \gamma) \right| = \tilde{\mathcal{O}}_p\left( \frac{1}{s^2 n^{1/3}} \left[ \frac{1}{s} + L_k + L_l + \sqrt{D_\Omega} + \sqrt{D_\Gamma} \right] \right).$$

Appendices C and D state and prove non-asymptotic versions of the above results based on covering numbers; the assumptions and proof techniques are similar to those of Liu et al. (2020)[10], but is slightly more involved for a few reasons: 1) Uniform convergence of HSIC and its variance estimators now include two kernels, $k$ and $l$, instead of one, and a somewhat different estimator form. 2) For NDS we also need to show convergence of a null-to-alternative variance ratio, a term which is not present in the MMD or HSIC objectives.

Following Theorem C.13 (a standard result from van der Vaart, 1998), successfully maximizing these estimates will thus also maximize the population quantity, and consequently the asymptotic test power, for sufficiently large training set sizes $n$.

# 6 Experiments

The repository https://github.com/xunathan96/deephsic/ contains implementations of our methods and code for our experiments.

---

[8]Equation (13) could use block estimators (Zaremba et al., 2014) or incomplete $U$-statistics (Blom, 1976) to reduce $\mathcal{O}(K^2L)$ to $\mathcal{O}(K^\beta L)$ for any $\beta \leq 2$, at the cost of increased variance (see Ramdas et al., 2015).

[9]Indeed, no unbiased estimator is likely to exist; see Appendix A of Deka & Sutherland 2023.

[10]Ren et al. (2024) stated a similar result to Theorem 5.2, but with a seemingly incorrect proof: in their Appendix D.3, the term $|r^{(n)} - r|$ in their (63) should be bounded uniformly over kernel parameters, but they appeal to a result about convergence of distribution functions for a single distribution. For the uniform result over kernel parameters that their theorem statement claims, they would need to additionally use a cover or other approach (as is done for the HSIC and the variance estimators), but none is attempted.

**Baselines.** We compare our HSIC-based (HSIC-D/Dx/O) and MI-based (NDS/InfoNCE/NWJ) tests with various alternative methods. All tests are performed using permutation testing.

- HSIC-D: HSIC using deep kernels on each space $\mathcal{X}$ and $\mathcal{Y}$; simultaneously trained via Section 5.
- HSIC-Dx: HSIC using a tied deep kernel, i.e. $k_\omega = l_\gamma$, and trained via Section 5.
- HSIC-O: HSIC using Gaussian kernels, with each bandwidth parameter optimized via Section 5.
- NDS: The neural dependency statistic (2) trained via Section 5.
- InfoNCE (van den Oord et al., 2018; Poole et al., 2019): the statistic $\hat{I}_{\text{NCE}}$ as in (1).
- NWJ (Nguyen et al., 2010; Poole et al., 2019): another mutual information bound statistic $\hat{I}_{\text{NWJ}}$.
- HSIC-M: HSIC using a Gaussian kernel, with bandwidth selected via the median heuristic.
- HSIC-Agg (Albert et al., 2021; Schrab et al., 2023): aggregating Gaussian kernels of various bandwidths. We use the complete U-statistic with default settings: $B1 = 500$, $B2 = 500$, $B3 = 50$, and uniform weights.
- MMD-D: The method of Liu et al. (2020) applied to $\mathbb{P}_{xy}$ vs $\mathbb{P}_x \times \mathbb{P}_y$,[11] with a Gaussian kernel on $\mathcal{X} \times \mathcal{Y}$.
- C2ST-S (Lopez-Paz & Oquab, 2017) / C2ST-L (Cheng & Cloninger, 2022): Sign/logit-based classifier two-sample test for $\mathbb{P}_{xy}$ vs $\mathbb{P}_x \times \mathbb{P}_y$, with samples set up the same way as for MMD-D.

**Datasets.** We consider four informative datasets, where the true answers are known.

● **High-dimensional Gaussian mixture**. The distribution HDGM-$d$ has $d$ total dimensions (divided between $X$ and $Y$), but has dependence between only two of them:

$$[X_1, Y_{\lfloor d/2 \rfloor}, \ldots, X_{\lceil d/2 \rceil}, Y_1] \sim \sum_{i=1}^{2} \frac{1}{2} \mathcal{N} \left( \mathbf{0}_d, \begin{bmatrix} 1 & 0.5(-1)^i & \mathbf{0}_{d-2}^T \\ 0.5(-1)^i & 1 & \mathbf{0}_{d-2}^T \\ \mathbf{0}_{d-2} & \mathbf{0}_{d-2} & I_{d-2} \end{bmatrix} \right),$$

where the odd dimensions are taken to be from $\mathbb{P}_x$ and even dimensions to be from $\mathbb{P}_y$. Moreover, for $d \geq 4$ the dependent variables $X_1$ and $Y_{\lfloor d/2 \rfloor}$ are at different dimensions. We perform independence tests at dimensions 4, 8, 10, 20, 30, 40, and 50. This distribution is similar to one used by Liu et al. (2020).

● **Sinusoid** (Sejdinovic et al., 2012). We sample from sinusoidally dependent data with distribution $\mathbb{P}_{xy} \propto 1 + \sin(\ell x) \sin(\ell y)$ on support $\mathcal{X} \times \mathcal{Y} = [-\pi, \pi]^2$. Higher frequencies $\ell$ produce subtler departures from the uniform distribution, resulting in a harder independence problem; we use $\ell = 4$. A visualization of this density is given in Figure 8.

● **RatInABox** (George et al., 2024). RatInABox simulates hippocampal cells of a rat in motion. In particular, we test for dependence between firing rates of grid cells and the rat's head direction. Grid cells respond near points in a grid covering the environment surface, and should be subtly connected to head direction because of the geometry of the "box" (Figure 7). We consider 8 grid cells, and simulate motion for 100 000 seconds, taking a measurement every 5 seconds as our dataset.

● **Wine Quality** (Paulo et al., 2009). Details physicochemical properties (e.g., sugar, pH, chlorides) of different types of red and white wines and their perceived quality as an integer value from 1 to 10. We test for dependency between residual sugar levels and quality.

**Power vs. test size.** We first compare how well methods identify dependency with a large-size training set, by comparing the rate at which the learned tests achieve perfect power (1.0) as the test size $m$ increases. Training and validation sizes for each dataset are given in Appendix E.1, and results are visualized in Figure 4 (with comprehensive results in Figure 9). Overall, HSIC-D outperforms baselines, and is able to reach perfect power at smaller test sizes $m$. We note that HSIC-O performs reasonably well for simpler problems like Sinusoid, but struggles on harder problems like RatInABox. This suggests that no Gaussian kernel on the input space is well-suited for the task. On the other hand, kernels applied to optimal representations of the data (HSIC-D) are the most powerful. Surprisingly, optimizing the power of NDS tests is significantly weaker than directly maximizing InfoNCE or NWJ; we elaborate on this in the following section.

---

[11] Specifically, we compare $\mathbb{Z} \sim \mathbb{P}_{xy}$ against a single shuffle of the samples $\sigma \mathbb{Z}$, as detailed in Section 3.2. We use this approach over the data-splitting approach since it both minimizes memory (compared to using a large number of permutations) and performs slightly better in practice, as seen in Figure 13.

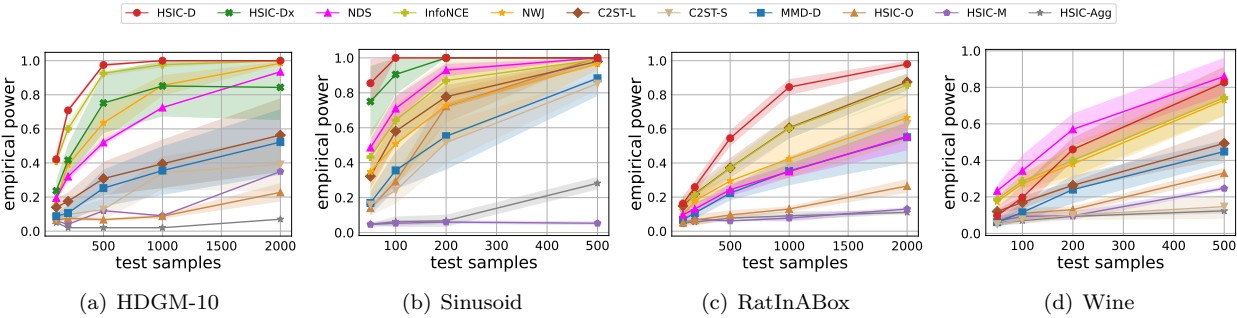

(a) HDGM-10    (b) Sinusoid    (c) RatInABox    (d) Wine

Figure 4: Empirical power vs sample size $m$ for different datasets, when trained with a large training set. The average test power is computed over 5 training runs, where the empirical power is determined over 100 permutation tests. The shaded region covers one standard error from the mean.

**Power vs. dataset size.** A drawback to kernel selection via optimization is that we must hold out a split of the data for training. In contrast, HSIC-M and HSIC-Agg are able to utilize the entirety of the dataset for their test. To examine this trade-off, we consider consistent data splitting at datasets of varying sizes. For the HDGM and Sinusoid problems we use a 7:2:1 train-val-test split, and for RatInABox and Wine we use a 3:2 train-test split. Conversely, HSIC-M and HSIC-Agg use the *entire* dataset for testing. Results are shown in Figure 5. Non-splitting methods are able to take advantage of the additional test samples and outperform some data-splitting methods at smaller dataset sizes; for large dataset sizes the reverse is true. The validation set in this experiment is used only for early stopping of the training process, and not for hyperparameter selection.

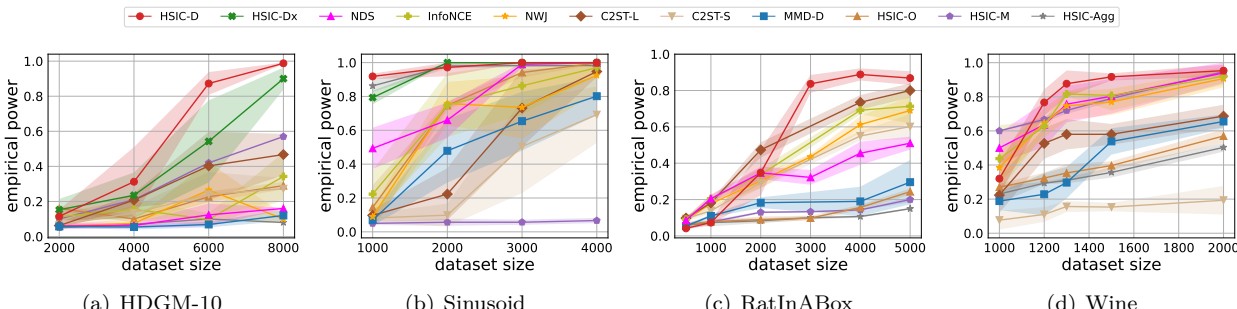

(a) HDGM-10    (b) Sinusoid    (c) RatInABox    (d) Wine

Figure 5: Empirical power vs dataset size. HDGM and Sinusoid uses a consistent 7:2:1 train-val-test split across all dataset sizes, while RatInABox and Wine maintain a 3:2 train-test split. HISC-Agg and HSIC-M do not split the data. The shaded region covers one standard deviation over 5 training runs.

**SNR with vs. without threshold.** Our HSIC objective $\hat{J}^\lambda_{\mathrm{HSIC}}$ disregards the threshold-dependent term, whereas the NDS objective $\hat{J}^\lambda_{\mathrm{NDS}}$ does not. The impact this term has during training, however, is still unclear. We compare the strength of tests maximized with and without this term in Figure 15. It is important to note that the threshold estimate for HSIC is challenging (see also Footnote 6); we use a method based on the moment-matched Gamma approximation, but with improved gradient estimation rather than the finite differences estimate they used (Appendix F.6). This estimate, in either form, introduces substantial additional code complexity and its own bias and variance that could each hurt overall performance. In general, we find that including the threshold for HSIC has no strong effect on simpler problems, but degrades the test power for harder problems. On the other hand, NDS with the threshold yields stronger tests. This is not too surprising as, unlike for HSIC, the null distribution here follows the CLT and is easily approximated given enough samples.

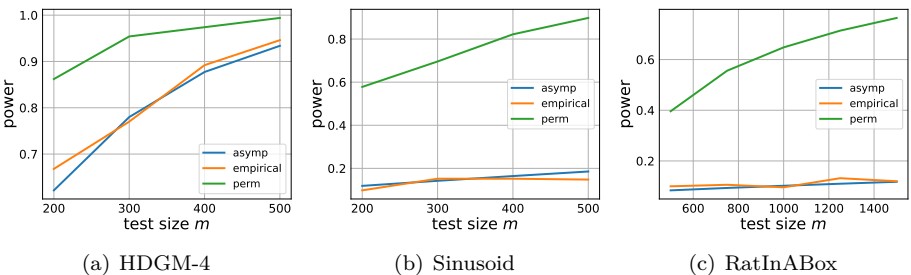

(a) HDGM-4      (b) Sinusoid      (c) RatInABox

Figure 6: Power of NDS tests derived from a learned InfoNCE critic $f$. For each problem, the blue line (asymp) is power described by the asymptotic formula (3), where we use a very large sample size ($n = 20\,000$) to estimate the population level statistics $T, T_0$ and their deviations $\sigma_{\mathfrak{H}_1}, \sigma_{\mathfrak{H}_0}$. The orange line (empirical) computes the power based on the simulated null distribution, i.e. the threshold estimated via the empirical CDF of $\hat{T}_0$ with test size $m$. The green line (perm) is the permutation test power.

## 7 Discussion

**Independence tests with MMD.** We find that MMD-D underperforms HSIC-D in almost all settings, despite the equivalence between MMD and HSIC specified in Proposition 3.1. This equivalence, while true at the population level, is not the case when considering finite sample estimates. These estimators exhibit different statistical properties, with some being more conducive to learning than others. In particular, we notice that MMD estimators based on a single permutation of the samples (as described in Section 3.2) exhibit larger variance compared to HSIC ones. We prove this for the biased estimators in Appendix F.4, and observe the same phenomenon when training MMD-D and HSIC-D.

**Permutation tests sometimes disagree with asymptotics.** We hypothesized that maximizing an estimate of the asymptotic power would produce stronger tests than maximizing the test statistic directly. Although this seems to be true for HSIC (Appendix F.3), it is surprisingly not the case for MI-based statistics; The previous power versus test size experiments show that directly maximizing InfoNCE or NWJ outperforms maximizing the NDS asymptotic power, at least when using a permutation test. Why is this?

Figure 6 examines several power estimates for an NDS test based on an InfoNCE-learned critic. Notice that the asymptotic formula (blue) agrees well with the simulated test power (orange), indicating that the asymptotics describe the test well. However, the permutation power (green) is significantly greater than what the asymptotics suggest. Recalling that the simulated test power is in fact using a near-exact threshold based on the empirical quantile of the test statistic under the null, this should be roughly the most powerful valid test with a *sample-independent* threshold. Permutation tests, however, use a *sample-dependent* threshold, which in this case apparently can yield a much more powerful test than the best sample-independent threshold. We explore this discrepancy further in Appendix F.7.

Thus, while maximizing the power of the NDS test maximizes the asymptotic power of a test with a data-independent threshold, it does *not* necessarily maximize the power of a permutation test. InfoNCE, and other variational MI estimators, seem to frequently choose critics whose sample-independent threshold tests are worse than those obtained by NDS maximization, but whose *permutation* tests are actually much more powerful. Exploring this connection more deeply is a very intriguing area for future work.

We did not observe this discrepancy for HSIC; maximizing the sample-independent asymptotics for HSIC seems to yield permutation tests (roughly matching their asymptotics) which still outperform InfoNCE or similar NDS permutation tests. Overall, HSIC-D seems to be the most reliable method of those considered here.

## 8  Conclusion and Future Work

Independence testing aims to see if two paired random variables are statistically independent. We explored two families of tests to address this problem. The first are tests based on variational mutual information estimators, which, to the best of our knowledge, we are the first to construct. The second are tests based on maximizing the asymptotic power, which was explored for the two-sample problem but not for independence testing. Our findings show that learning representations of the data via our proposed methods lead to powerful tests, with HSIC-based tests generally outperforming MI-based ones. Future work may look to extend this learning scheme to conditional independence testing and apply this to causal discovery. Meanwhile, it will also be valuable to investigate approaches for mitigating, or even removing, the data-splitting procedure – as done by Biggs et al. (2023); Kübler et al. (2020) – while maintaining the ability to learn rich data representations.

### Acknowledgments

The authors would like to thank Roman Pogodin for productive discussions. This work was enabled in part by support provided by the Natural Sciences and Engineering Research Council of Canada (with grant numbers RGPIN-2021-02974 and ALLRP-57708-2022), the Canada CIFAR AI Chairs program, Calcul Québec, the BC DRI Group, and the Digital Research Alliance of Canada. This work was also supported by the Australian Research Council (ARC) with grant numbers DE240101089 and DP230101540.

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

# A  Asymptotics of HSIC Estimators

**Proposition A.1** (Song et al., 2012, Theorem 5). *Under the alternative hypothesis $\mathfrak{H}_1 : \mathbb{P}_{xy} \neq \mathbb{P}_x \times \mathbb{P}_y$, the unbiased estimator of HSIC is asymptotically normal: with $m$ samples,*

$$\sqrt{m}(\widehat{\mathrm{HSIC}}_{\mathrm{u}} - \mathrm{HSIC}) \xrightarrow{d} \mathcal{N}(0, \sigma_{\mathfrak{H}_1}^2). \tag{14}$$

*The asymptotic variance of $\sqrt{m}\,\widehat{\mathrm{HSIC}}_{\mathrm{u}}$, $\sigma_{\mathfrak{H}_1}^2$, can be consistently estimated from $n$ samples[12] as $16\left(R - \mathrm{HSIC}^2\right)$, where $R = \frac{1}{n} \sum_{i=1}^{n} \left( \frac{(n-4)!}{(n-1)!} \sum_{(j,q,r)\in \mathbf{i}_3^n / \{i\}} h(i,j,q,r) \right)^2$. Here $\mathbf{i}_n^\ell \setminus \{i\}$ denotes the set of all $\ell$-tuples drawn without replacement from the set $\{1,\ldots,n\} \setminus \{i\}$, and $h(i,j,q,r) = \frac{1}{24} \sum_{(s,t,u,v)}^{(i,j,q,r)} k_{st}(l_{st} + l_{uv} - 2l_{su})$, where the sum ranges over all $4! = 24$ ways to assign the distinct indices $\{i,j,q,r\}$ to the four variables $(s,t,u,v)$ without replacement.*

The behavior of the estimator is different under the null hypothesis $(X \perp\!\!\!\perp Y)$; in this regime $m\,\widehat{\mathrm{HSIC}}_{\mathrm{u}}$ (rather than $\sqrt{m}\widehat{\mathrm{HSIC}}_{\mathrm{u}}$) converges in distribution to something with complex dependence on $\mathbb{P}_x$, $\mathbb{P}_y$, $k$, and $l$.

**Proposition A.2** (Gretton et al., 2007, Theorem 2). *Under the null hypothesis $\mathfrak{H}_0 : \mathbb{P}_{xy} = \mathbb{P}_x \times \mathbb{P}_y$, the U-statistic estimator of $\mathrm{HSIC}$ is degenerate. In this case $m\widehat{\mathrm{HSIC}}_{\mathrm{u}}$ converges in distribution as*

$$m\widehat{\mathrm{HSIC}}_{\mathrm{u}} \xrightarrow{d} \sum_{\ell=1}^{\infty} \lambda_\ell(z_\ell^2 - 1),$$

*where $z_\ell \stackrel{iid}{\sim} \mathcal{N}(0,1)$, and $\lambda_\ell$ are the solutions to the eigenvalue problem*

$$\lambda_\ell \psi_\ell(z_j) = \int h_{ijqr} \psi_\ell(z_i) \, \mathrm{d}F_{i,q,r},$$

*where $h_{ijqr} := h(i,j,q,r)$ is defined in Proposition A.1, and $F_{i,q,r}$ denotes the probability measure with respect to variables $z_i$, $z_q$, and $z_r$.*

# B  Bernstein-based Justification for the SNR

For both the NDS and HSIC tests, our justifications of the SNR criterion for test power, (3) and (6), relied on the asymptotic central limit theorem. A Berry-Esseen theorem would provide uniform bounds on the $o(1)$ terms that appear in these derivations, but one might expect for HSIC in particular that these might not be especially tight in some cases: since the null distribution is asymptotically a mixture of chi-squared variables (Proposition A.2), when the dependence under the current kernel is weak, the distribution may be further from normal.

We can also justify the same criterion, although less precisely, with a non-asymptotic argument. While we present it here for HSIC, it also applies in the same way to NDS. The Bernstein-type bound of Maurer (2017, Theorem 6) gives that, for kernels bounded as in Assumption (C'),

$$\Pr\left(|\widehat{\mathrm{HSIC}} - \mathrm{HSIC}| > \varepsilon\right) \leq 2\exp\left(\frac{-n\varepsilon^2}{2\sigma_{\mathfrak{H}_1}^2 + \frac{2304\nu_k\nu_l}{n-4} + \frac{256}{3}\varepsilon}\right) \leq \delta.$$

Solving for $\varepsilon$, we obtain

$$\frac{-m\varepsilon^2}{2\sigma_{\mathfrak{H}_1}^2 + \frac{2304\nu_k\nu_l}{m-4} + \frac{256}{3}\varepsilon} \leq \log\frac{\delta}{2}$$

$$\varepsilon^2 \geq \frac{1}{m}\log\frac{2}{\delta}\left(2\sigma_{\mathfrak{H}_1}^2 + \frac{2304\nu_k\nu_l}{m-4} + \frac{256}{3}\varepsilon\right)$$

$$\varepsilon^2 - \frac{256}{3m}\log\frac{2}{\delta}\varepsilon - \left(\frac{2}{m}\sigma_{\mathfrak{H}_1}^2 \log\frac{2}{\delta} + \frac{2304\nu_k\nu_l}{m-4}\right) \geq 0.$$

---

[12]Typically $m = n$, but we might want to use a few ($n$) samples to roughly estimate the power of an $m$-sample test with $m \gg n$, as done in a different context by Sutherland & Deka (2019); Deka & Sutherland (2023).

This can be written $\varepsilon^2 - B\varepsilon - C^2 \geq 0$ for positive $B, C$; restricting to positive $\varepsilon$, this occurs when $\varepsilon \geq \frac{1}{2}B + \frac{1}{2}\sqrt{B^2 + 4C^2}$, and hence also occurs when $\varepsilon$ is larger than the (slightly larger but simpler) bound $B + C$. That is, with probability at least $1 - \delta$, it holds that

$$
\begin{aligned}
|\widehat{\mathrm{HSIC}} - \mathrm{HSIC}| &\leq \frac{256}{3m} \log \frac{2}{\delta} + \sqrt{\frac{2}{m}\sigma_{\mathfrak{H}_1}^2 \log \frac{2}{\delta} + \frac{2304\nu_k\nu_l}{m-4}} \\
&\leq \frac{256}{3m} \log \frac{2}{\delta} + \sigma_{\mathfrak{H}_1}\sqrt{\frac{2}{m}\log\frac{2}{\delta}} + \sqrt{\frac{2304\nu_k\nu_l}{m-4}}.
\end{aligned}
$$

Recall from the discussion preceding (6) that the rejection threshold $r_m$ is $\frac{1}{m}\left(\Psi^{-1}(1-\alpha) + o(1)\right)$ (where $\Psi$ will vary with the kernel choice). Now, a sufficient condition for the test rejecting is that both $\widehat{\mathrm{HSIC}} \geq \mathrm{HSIC} - \varepsilon$ and $\mathrm{HSIC} - \varepsilon > r_m$. Plugging in the deviation from above, we obtain power at least $1 - \delta$ as long as

$$
\begin{aligned}
\mathrm{HSIC} &> r_m + \frac{256}{3m}\log\frac{2}{\delta} + \sigma_{\mathfrak{H}_1}\sqrt{\frac{2}{m}\log\frac{2}{\delta}} + \sqrt{\frac{2304\nu_k\nu_l}{m-4}} \\
&= \sigma_{\mathfrak{H}_1}\sqrt{\frac{2}{m}\log\frac{2}{\delta}} + \frac{1}{m}\left(\Psi^{-1}(1-\alpha) + \mathcal{O}(1)\right) + 48\sqrt{\frac{\nu_k\nu_l}{m-4}},
\end{aligned}
$$

absorbing $\frac{256}{3m}\log\frac{2}{\delta} = \mathcal{O}(1/m)$. Equivalently, the power condition holds if

$$
\frac{\mathrm{HSIC}}{\sigma_{\mathfrak{H}_1}} - \frac{48}{\sqrt{m-4}}\frac{\sqrt{\nu_k\nu_l}}{\sigma_{\mathfrak{H}_1}} > \sqrt{\frac{2}{m}\log\frac{2}{\delta}} + \frac{1}{m}\frac{\Psi^{-1}(1-\alpha)}{\sigma_{\mathfrak{H}_1}} + \mathcal{O}\left(\frac{1}{m\sigma_{\mathfrak{H}_1}}\right).
$$

Now, we know that scaling $k$ or $l$ should not change the difficulty of the problem: indeed, it will scale HSIC, $\sigma_{\mathfrak{H}_1}$, $\Psi^{-1}(1-\alpha)$, and $\sqrt{\nu_k\nu_l}$ all equally. Thus, the only term in this inequality which is potentially sensitive to scaling is the final $\mathcal{O}$ term. From this point, then, while other changes in the kernel could potentially affect $\Psi^{-1}(1-\alpha)/\sigma_{\mathfrak{H}_1}$ in meaningful ways, for reasonably large $m$ and $\sigma_{\mathfrak{H}_1}$ not too drastically small, the right-hand side is only weakly dependent on the kernel choice. It is thus reasonable to think that we can maximize the left-hand side of the inequality, and ignore the right-hand side, to roughly maximize power. It seems highly unlikely, however, for the coefficient $48/\sqrt{m-4}$ to be exactly the right trade-off between HSIC and the kernel bound; this comes out of the details of the (conservative and somewhat loose) concentration inequalities to reach this point. To maximize our chance at having high power, then, it is very reasonable to maximize the SNR.

## C  Uniform Convergence: NDS

### C.1  Preliminaries

We start by defining the notation used in the following proofs. Let $f_\theta$ be the critic function with parameters $\theta$, and $\{X_i, Y_i\}_{i=1..n}$ be samples drawn from the joint distribution, with $\mathbb{X} = \{X_i\}_{i=1..n}$ and $\mathbb{Y} = \{Y_i\}_{i=1..n}$.

We define the population level statistics based on critic $f_\theta$ as

$$
T_{1,\theta} := \underset{\mathbb{P}_{XY}}{\mathbb{E}}\left[f_\theta(X,Y)\right] \qquad \text{and} \qquad T_{0,\theta} := \underset{\mathbb{P}_X \times \mathbb{P}_Y}{\mathbb{E}}\left[f_\theta(X,Y)\right],
$$

with their corresponding finite sample estimates

$$
\hat{T}_{1,\theta} := \frac{1}{n}\sum_{i=1}^{n} f_\theta(X_i, Y_i) \qquad \text{and} \qquad \hat{T}_{0,\theta} = \frac{1}{n^2}\sum_{i=1}^{n}\sum_{j=1}^{n} f_\theta(X_i, Y_j).
$$

We note that $\hat{T}_1$ is just the NDS (2), and that $\hat{T}_0$ is the permutation invariant term in the numerator of Equation (12). For brevity, we often omit the subscripts $0, 1$ and/or the parameters $\theta$ when it is uninformative or clear from the context.

We respectively denote the alternative and null variances by

$$\sigma_\theta^2 := \operatorname*{Var}_{\mathbb{P}_{XY}}[f_\theta(X,Y)] \qquad \text{and} \qquad \xi_\theta^2 := \operatorname*{Var}_{\mathbb{P}_X \times \mathbb{P}_Y}[f_\theta(X,Y)],$$

with corresponding finite sample estimates

$$\hat{\sigma}_\theta^2 := \frac{1}{n}\sum_{i=1}^n \left( f_\theta(X_i, Y_i) - \frac{1}{n^2}\sum_{j=1}^n f_\theta(X_j, Y_j) \right)^2.$$

and

$$\hat{\xi}_\theta^2 := \frac{1}{n^2}\sum_{i=1}^n\sum_{j=1}^n \left( f_\theta(X_i, Y_j) - \frac{1}{n^2}\sum_{i'=1}^n\sum_{j'=1}^n f_\theta(X_{i'}, Y_{j'}) \right)^2$$

We choose this notation, rather than $\tau_{\mathfrak{H}_1}^2$ and $\tau_{\mathfrak{H}_0}^2$ used in the main body, for distinguishability. As before, we often omit the subscripts when it is clear from the context.

## C.2   Assumptions

Our main results assume the following

(A) The set of critic parameters $\Theta$ lies in a Banach space of dimension $D$, where each parameter in the space is bounded by $R$:

$$\Theta \subseteq \{\theta \,:\, \|\theta\| \le R\}.$$

(B) The critic $f$ is uniformly Lipschitz on $\mathcal{X} \times \mathcal{Y}$ with respect to the parameter space $\Theta$:

$$|f_\theta(x,y) - f_{\theta'}(x,y)| \le L\|\theta - \theta'\| \qquad \forall x \in \mathcal{X}, y \in \mathcal{Y}, \theta \in \Theta.$$

(C) The critic is uniformly bounded:

$$\sup_{\theta\in\Theta} \sup_{\substack{x\in\mathcal{X}\\y\in\mathcal{Y}}} |f_\theta(x,y)| \le B;$$

that such a $B$ exists is implied by the previous two assumptions.

## C.3   Lemmas

**Lemma C.1** (Concentration of $\hat{T}_1$). *For any critic $f$ satisfying Assumption (C), with probability at least $1 - \delta$ we have*

$$|\hat{T}_1 - T_1| \le B\sqrt{\frac{2}{n}\log\frac{2}{\delta}}.$$

*Proof.* The result follows from McDiarmid's inequality. First, we show that $\hat{T}_1$ satisfies bounded differences. Let $\hat{T}_1^{(k)}$ denote the $\hat{T}_1$ statistic but with sample $X_k, Y_k$ replaced by $x', y'$. For any $k \in [n]$, we have

$$\sup_{\substack{x',y'\\\mathbb{X},\mathbb{Y}}} |\hat{T}_1^{(k)} - \hat{T}_1| = \sup_{\substack{x',y'\\X_k,Y_k}} \left| \frac{1}{n}\left(f(x',y') - f(X_k,Y_k)\right) \right| \le \frac{2B}{n},$$

so that with probability $1 - \delta$,

$$|\hat{T}_1 - \mathbb{E}\,\hat{T}_1| \le \sqrt{\frac{1}{2}\sum_{i=1}^n \left(\frac{2B}{n}\right)^2 \cdot \log\frac{2}{\delta}} = B\sqrt{\frac{2}{n}\log\frac{2}{\delta}}.$$

$\square$

**Lemma C.2** (Concentration of $\hat{T}_0$). *For any critic $f$ satisfying Assumption (C), with probability at least $1 - \delta$ we have*

$$|\hat{T}_0 - \mathbb{E}\,\hat{T}_0| < 2B\sqrt{\frac{2}{n}\log\frac{2}{\delta}}.$$

*Proof.* The result follows from McDiarmid's inequality. First, we show that $\hat{T}_0$ satisfies bounded differences. Let $\hat{T}_0^{(k)}$ denote the $\hat{T}_0$ statistic but with sample $X_k, Y_k$ replaced by $x', y'$. For any $k \in [n]$, we have

$$\sup_{\substack{x',y' \\ \mathbb{X},\mathbb{Y}}} |\hat{T}_0^{(k)} - \hat{T}_0| = \sup_{\substack{x',y' \\ \mathbb{X},\mathbb{Y}}} \frac{1}{n^2}\left|\sum_{\substack{i=k \\ j\neq k}}\Big(f(x',Y_j) - f(X_k,Y_j)\Big)\right.$$

$$\left. + \sum_{\substack{i\neq k \\ j=k}}\Big(f(X_i,y') - f(X_i,Y_k)\Big) + \sum_{i=j=k}\Big(f(x',y') - f(X_k,Y_k)\Big)\right|$$

$$\leq \frac{1}{n^2}\Big(2B(n-1) + 2B(n-1) + 2B\Big) = \frac{2B(2n-1)}{n^2},$$

so that with probability $1 - \delta$,

$$|\hat{T}_0 - \mathbb{E}\,\hat{T}_0| \leq 2B\sqrt{\left(\frac{2}{n} - \frac{2}{n^2} + \frac{1}{2n^3}\right)\log\frac{2}{\delta}} < 2B\sqrt{\frac{2}{n}\log\frac{2}{\delta}}$$

for all integers $n > 0$. $\qquad\square$

**Lemma C.3** (Bias of $\hat{T}_0$). *For any critic $f$ satisfying Assumption (C), the bias of $\hat{T}_0$ is bounded by*

$$|\mathbb{E}\,\hat{T}_0 - T_0| \leq \frac{2B}{n}.$$

*Proof.* Notice that

$$\mathbb{E}\,\hat{T}_0 = \mathbb{E}\left[\frac{1}{n^2}\sum_{i\neq j}f(X_i,Y_j) + \frac{1}{n^2}\sum_{i=j}f(X_i,Y_i)\right] = \frac{n-1}{n}T_0 + \frac{1}{n}T_1,$$

so that

$$|\mathbb{E}\,\hat{T}_0 - T_0| = \frac{1}{n}|T_1 - T_0| \leq \frac{2B}{n}.$$

$\qquad\square$

**Lemma C.4** (Concentration of $\hat{\sigma}^2$). *For any critic $f$ satisfying Assumption (C), with probability at least $1 - \delta$ we have*

$$|\hat{\sigma}^2 - \mathbb{E}\,\hat{\sigma}^2| \leq 5B^2\sqrt{\frac{1}{2n}\log\frac{2}{\delta}}.$$

*Proof.* The result follows from McDiarmid's inequality. First, we show that $\hat{\sigma}^2$ satisfies bounded differences. Let $\hat{\mu}_2 := \frac{1}{n}\sum_{i=1}^n f(X_i,Y_i)^2$ be the 2nd moment estimate so that $\hat{\sigma}^2 = \hat{\mu}_2 - \hat{T}_1^2$. Let $\hat{\sigma}_{(k)}^2, \hat{\mu}_{2,(k)}, \hat{T}_{1,(k)}$ denote the corresponding statistics $\hat{\sigma}^2, \hat{\mu}_2, \hat{T}_1$ but with sample $X_k, Y_k$ replaced by $x', y'$. For any $k \in [n]$, we have

$$\sup_{\substack{x',y' \\ \mathbb{X},\mathbb{Y}}} |\hat{\sigma}_{(k)}^2 - \hat{\sigma}^2| \leq \sup_{\substack{x',y' \\ \mathbb{X},\mathbb{Y}}} |\hat{\mu}_{2,(k)} - \hat{\mu}_2| + \sup_{\substack{x',y' \\ \mathbb{X},\mathbb{Y}}} |\hat{T}_{1,(k)}^2 - \hat{T}_1^2|$$

$$= \sup_{\substack{x',y' \\ X_k,Y_k}} \frac{1}{n}|f(x',y')^2 - f(X_k,Y_k)^2| + \sup_{\substack{x',y' \\ \mathbb{X},\mathbb{Y}}} |\hat{T}_{1,(k)} - \hat{T}_1||\hat{T}_{1,(k)} + \hat{T}_1|$$

$$\leq \frac{B^2}{n} + \frac{4B^2}{n} = \frac{5B^2}{n},$$

where we use the finite differences computed in the proof of Lemma C.1 to bound $|\hat{T}_{1,(k)} - \hat{T}_1|$. Thus, with probability at least $1 - \delta$,

$$|\hat{\sigma}^2 - \mathbb{E}\,\hat{\sigma}^2| \le 5B^2 \sqrt{\frac{1}{2n} \log \frac{2}{\delta}}.$$

$\square$

**Lemma C.5** (Bias of $\hat{\sigma}^2$). *For any critic $f$ satisfying Assumption (C), the bias of $\hat{\sigma}^2$ is bounded by*

$$|\mathbb{E}\,\hat{\sigma}^2 - \sigma^2| \le \frac{B^2}{n}$$

*Proof.* Let $\hat{\mu}_2 = \frac{1}{n} \sum_{i=1}^{n} f(X_i, Y_i)^2$ and $\mu_2 = \mathbb{E}[\hat{\mu}_2] = \mathbb{E}_{\mathbb{P}_{XY}} f(X,Y)^2$. It follows that

$$
\mathbb{E}\,\hat{\sigma}^2 = \mathbb{E}[\hat{\mu}_2 - \hat{T}_1^2] = \mu_2 - \mathbb{E}\left[\left(\frac{1}{n}\sum_{i=1}^{n} f(X_i, Y_i)\right)^2\right] = \mu_2 - \frac{1}{n^2}\mathbb{E}\left[\sum_i \sum_j f(X_i, Y_i) f(X_j, Y_j)\right]
$$

$$
= \mu_2 - \frac{1}{n^2}\mathbb{E}\left[\sum_{i=j} f(X_i, Y_i)^2 + \sum_{i \ne j} f(X_i, Y_i) f(X_j, Y_j)\right]
$$

$$
= \mu_2 - \frac{1}{n^2}\left(n\mu_2 + n(n-1)T_1^2\right) = \frac{n-1}{n}\sigma^2,
$$

which yields

$$|\mathbb{E}\,\hat{\sigma}^2 - \sigma^2| = \frac{1}{n}\sigma^2 \le \frac{B^2}{n}.$$

$\square$

**Lemma C.6** (Concentration of $\hat{\xi}^2$). *For any critic $f$ satisfying Assumption (C), with probability at least $1 - \delta$ we have*

$$\left|\hat{\xi}^2 - \mathbb{E}\,\hat{\xi}^2\right| < 10B^2 \sqrt{\frac{1}{2n} \log \frac{2}{\delta}}.$$

*Proof.* The result follows from McDiarmid's inequality. We first show that $\hat{\xi}^2$ satisfies bounded differences. Let $\hat{\nu}_2 := \frac{1}{n^2} \sum_{i,j \in [n]} f(X_i, Y_j)^2$ be the 2nd moment estimate so that $\hat{\xi}^2 = \hat{\nu}_2 - \hat{T}_0^2$. Let $\hat{\xi}^2_{(\ell)}, \hat{\nu}_{2,(\ell)}, \hat{T}_{0,(\ell)}$ denote the corresponding statistics $\hat{\xi}^2, \hat{\nu}_2, \hat{T}_0$ but with sample $X_\ell, Y_\ell$ replaced by $x', y'$. For any $\ell \in [n]$ we have

$$
\sup_{\substack{x', y' \\ \mathbb{X}, \mathbb{Y}}} |\hat{\xi}^2_{(\ell)} - \hat{\xi}| \le \sup_{\substack{x', y' \\ \mathbb{X}, \mathbb{Y}}} |\hat{\nu}_{2,(\ell)} - \hat{\nu}_2| + \sup_{\substack{x', y' \\ \mathbb{X}, \mathbb{Y}}} |\hat{T}^2_{0,(\ell)} - \hat{T}_0^2|
$$

$$
= \sup_{\substack{x', y' \\ \mathbb{X}, \mathbb{Y}}} \frac{1}{n^2} \left|\sum_{\substack{i=\ell \\ \text{or} \\ j=\ell}} f(X_i, Y_j)^2\right| + \sup_{\substack{x', y' \\ \mathbb{X}, \mathbb{Y}}} |\hat{T}_{0,(\ell)} - \hat{T}_0||\hat{T}_{0,(\ell)} + \hat{T}_0|
$$

$$
\le \frac{B^2(2n-1)}{n^2} + \frac{2B(2n-1)}{n^2}2B \le \frac{10B^2}{n} - \frac{5B^2}{n^2}.
$$

It follows that with probability at least $1 - \delta$,

$$\left|\hat{\xi}^2 - \mathbb{E}\,\hat{\xi}^2\right| \le \left(10B^2 - \frac{5B^2}{n}\right)\sqrt{\frac{1}{2n} \log \frac{2}{\delta}}.$$

$\square$

**Lemma C.7** (Bias of $\hat{\xi}^2$). *For any critic $f$ satisfying Assumption (C), the bias of $\hat{\xi}^2$ is bounded by*

$$\left|\mathbb{E}[\hat{\xi}^2] - \xi^2\right| < \frac{7B^2}{n}$$

*Proof.* Let $\nu_2 = \mathbb{E}_{\mathbb{P}_X \times \mathbb{P}_Y}[f(X,Y)^2]$ and $\mu_2 = \mathbb{E}_{\mathbb{P}_{XY}}[f(X,Y)^2]$ be the second moment under the null and alternative distributions, and let $\hat{\nu}_2 = \frac{1}{n^2}\sum_{i,j} f(X_i,Y_j)^2$ and $\hat{\mu}_2 = \frac{1}{n}\sum_i f(X_i,Y_i)^2$ be their estimates. First, we can write the bias as

$$\left|\mathbb{E}[\hat{\xi}^2] - \xi^2\right| = \left|\mathbb{E}[\hat{\nu}_2 - \hat{T}_0^2] - (\nu_2 - T_0^2)\right| \leq |\mathbb{E}\,\hat{\nu}_2 - \nu_2| + \left|\mathbb{E}[\hat{T}_0^2] - T_0^2\right|.$$

Now since

$$\mathbb{E}\,\hat{\nu}_2 = \frac{1}{n^2}\,\mathbb{E}\left[\sum_{i=j} f(X_i,Y_i)^2 + \sum_{i \neq j} f(X_i,Y_j)^2\right] = \frac{1}{n^2}\left(n\mu_2 + n(n-1)\nu_2\right) = \frac{\mu_2}{n} + \frac{n-1}{n}\nu_2,$$

we have that $|\mathbb{E}\,\hat{\nu}_2 - \nu_2| = \frac{1}{n}|\mu_2 - \nu_2| \leq B^2/n$. For the remaining term, note that

$$\mathbb{E}[\hat{T}_0^2] = \mathbb{E}\left[\left(\frac{1}{n^2}\sum_{i,j} f(X_i,Y_j)\right)^2\right] = \frac{1}{n^4}\,\mathbb{E}\left[\sum_{i,j,q,r} f(X_i,Y_j)f(X_q,Y_r)\right]$$

$$= \frac{1}{n^4}\,\mathbb{E}\left[\sum_{(i,j,q,r)\in\mathbf{i}_4^n} f(X_i,Y_j)f(X_q,Y_r) + \sum_{(i,j,q,r)\notin\mathbf{i}_4^n} f(X_i,Y_j)f(X_q,Y_r)\right]$$

$$\leq \frac{1}{n^4}\left[(n)_4 T_0^2 + \left(n^4 - (n)_4\right)B^2\right] = \left(1 - \frac{6}{n} + \frac{11}{n^2} - \frac{6}{n^3}\right)T_0^2 + \left(\frac{6}{n} - \frac{11}{n^2} + \frac{6}{n^3}\right)B^2,$$

so that $|\mathbb{E}[\hat{T}_0^2] - T_0^2| = |(\frac{6}{n} - \frac{11}{n^2} + \frac{6}{n^3})(B^2 - T_0^2)| \leq (\frac{6}{n} - \frac{11}{n^2} + \frac{6}{n^3})B^2$. Therefore, putting it all together we get

$$\left|\mathbb{E}[\hat{\xi}^2] - \xi^2\right| \leq \frac{7B^2}{n} - \frac{11B^2}{n^2} + \frac{6B^2}{n^3} < \frac{7B^2}{n}.$$

$\square$

## C.4 Propositions

**Proposition C.8** (Uniform convergence of $\hat{T}_1$)**.** *For any critic family $\{f_\theta : \theta \in \Theta\}$ and parameter space $\Theta$ satisfying Assumptions (A) to (C), we have with probability at least $1 - \delta$ that*

$$\sup_{\theta\in\Theta} |\hat{T}_{1,\theta} - T_{1,\theta}| \leq \frac{2L}{\sqrt{n}} + \frac{B}{\sqrt{n}}\sqrt{2\log\frac{2}{\delta} + 2D\log(4R\sqrt{n})} = \tilde{\mathcal{O}}(n^{-1/2}).$$

*Proof.* We employ a covering-number argument on the parameter space. Let $\mathcal{U} = \{\theta_i\}_{i=1..N} \subseteq \Theta$ be a $\rho$-cover of $\Theta$; i.e., any point in $\Theta$ is in the closed $\rho$-neighborhood of some point in $\mathcal{U}$. Assumption (A) ensures the minimal cover exists with at most $N = (4R/\rho)^D$ points (Cucker & Smale, 2002, Proposition 5). Now, for any $\theta \in \Theta$, let $\theta'$ be the center in $\mathcal{U}$ closest to $\theta$, and then decompose the bound into

$$\sup_{\theta\in\Theta}|\hat{T}_\theta - T_\theta| \leq \underbrace{\sup_{\theta\in\Theta}|\hat{T}_\theta - \hat{T}_{\theta'}|}_{(a)} + \underbrace{\max_{\theta'\in\mathcal{U}}|\hat{T}_{\theta'} - T_{\theta'}|}_{(b)} + \underbrace{\sup_{\theta\in\Theta}|T_{\theta'} - T_\theta|}_{(c)}.$$

We first handle terms (a) and (c). From Assumption (B), we have

$$\sup_{\theta\in\Theta}|T_{\theta'} - T_\theta| \leq \sup_{\theta\in\Theta}\mathbb{E}\,|f_{\theta'}(X,Y) - f_\theta(X,Y)| \leq \sup_{\theta\in\Theta} L\|\theta' - \theta\| \leq L\rho.$$

This same bound applies for (a) by replacing $\mathbb{E}$ with its empirical expectation $\hat{\mathbb{E}}$.

Next, we find a high probability bound on (b). Recalling the result of Lemma C.1, and then taking the union bound over all $N$ points in $\mathcal{U}$, we get that with probability at least $1 - \delta$

$$\max_{\theta\in\mathcal{U}}|\hat{T}_\theta - T_\theta| \leq B\sqrt{\frac{2}{n}\left(\log\frac{2}{\delta} + D\log\frac{4R}{\rho}\right)}.$$

Finally, putting it all together, we get with probability at least $1 - \delta$ that

$$\sup_{\theta \in \Theta} |\hat{T}_\theta - T_\theta| \le 2L\rho + B\sqrt{\frac{2}{n}\left(\log\frac{2}{\delta} + D\log\frac{4R}{\rho}\right)},$$

which, for $\rho = 1/\sqrt{n}$, gives the desired result. $\qquad\square$

**Proposition C.9** (Uniform convergence of $\hat{T}_0$). *For any critic family $\{f_\theta : \theta \in \Theta\}$ and parameter space $\Theta$ satisfying Assumptions (A) to (C), we have with probability at least $1 - \delta$ that*

$$\sup_{\theta \in \Theta} |\hat{T}_{0,\theta} - T_{0,\theta}| \le \frac{2L}{\sqrt{n}} + \frac{2B}{n} + \frac{2B}{\sqrt{n}}\sqrt{2\log\frac{2}{\delta} + 2D\log(4R\sqrt{n})} = \tilde{\mathcal{O}}(n^{-1/2}).$$

*Proof.* We employ a covering-number argument on the parameter space. Let $\mathcal{U} = \{\theta_i\}_{i=1..N} \subseteq \Theta$ be a $\rho$-cover of $\Theta$; i.e., any point in $\Theta$ is in the closed $\rho$-neighborhood of some point in $\mathcal{U}$. Assumption (A) ensures the minimal cover exists with at most $N = (4R/\rho)^D$ points (Cucker & Smale, 2002, Proposition 5). Now, for any $\theta \in \Theta$, let $\theta'$ be the center in $\mathcal{U}$ closest to $\theta$, and then decompose the bound into

$$\sup_{\theta \in \Theta} |\hat{T}_\theta - T_\theta| \le \underbrace{\sup_{\theta \in \Theta} |\hat{T}_\theta - \hat{T}_{\theta'}|}_{(a)} + \underbrace{\max_{\theta' \in \mathcal{U}} |\hat{T}_{\theta'} - \mathbb{E}\,\hat{T}_{\theta'}|}_{(b)} + \underbrace{\max_{\theta' \in \mathcal{U}} |\mathbb{E}\,\hat{T}_{\theta'} - T_{\theta'}|}_{(c)} + \underbrace{\sup_{\theta \in \Theta} |T_{\theta'} - T_\theta|}_{(d)}.$$

We first handle terms (a) and (d). From Assumption (B), we have

$$\sup_{\theta \in \Theta} |T_{\theta'} - T_\theta| \le \sup_{\theta \in \Theta} \mathbb{E}\,|f_{\theta'}(X, \tilde{Y}) - f_\theta(X, \tilde{Y})| \le \sup_{\theta \in \Theta} L\|\theta' - \theta\| \le L\rho,$$

which is the same bound when considering $\hat{T}_{0,\theta}$ instead of $T_{0,\theta}$.

Next, we find a high probability bound on (b). Recalling the result of Lemma C.2, and then taking the union bound over all $N$ points in $\mathcal{U}$, we get that with probability at least $1 - \delta$

$$\max_{\theta \in \mathcal{U}} |\hat{T}_\theta - \mathbb{E}\,\hat{T}_\theta| \le 2B\sqrt{\frac{2}{n}\left(\log\frac{2}{\delta} + D\log\frac{4R}{\rho}\right)}.$$

Finally, combining everything and using Lemma C.3 to bound (c), we get that with probability at least $1 - \delta$

$$\sup_{\theta \in \Theta} |\hat{T}_\theta - T_\theta| \le 2L\rho + \frac{2B}{n} + 2B\sqrt{\frac{2}{n}\left(\log\frac{2}{\delta} + D\log\frac{4R}{\rho}\right)},$$

which, for $\rho = 1/\sqrt{n}$, gives the desired result. $\qquad\square$

**Proposition C.10** (Uniform convergence of $\hat{\sigma}^2$). *For any critic family $\{f_\theta : \theta \in \Theta\}$ and parameter space $\Theta$ satisfying Assumptions (A) to (C), we have with probability at least $1 - \delta$ that*

$$\sup_{\theta \in \Theta} |\hat{\sigma}_\theta^2 - \sigma_\theta^2| \le \frac{8BL}{\sqrt{n}} + \frac{B^2}{n} + \frac{5B^2}{2\sqrt{n}}\sqrt{2\log\frac{2}{\delta} + 2D\log(4R\sqrt{n})} = \tilde{\mathcal{O}}(n^{-1/2}).$$

*Proof.* We employ a covering-number argument on the parameter space. Let $\mathcal{U} = \{\theta_i\}_{i=1..N} \subseteq \Theta$ be a $\rho$-cover of $\Theta$; i.e., any point in $\Theta$ is in the closed $\rho$-neighborhood of some point in $\mathcal{U}$. Assumption (A) ensures the minimal cover exists with at most $N = (4R/\rho)^D$ points (Cucker & Smale, 2002, Proposition 5). Now, for any $\theta \in \Theta$, let $\theta'$ be the center in $\mathcal{U}$ closest to $\theta$, and then decompose the bound into

$$\sup_{\theta \in \Theta} |\hat{\sigma}_\theta^2 - \sigma_\theta^2| \le \underbrace{\sup_{\theta \in \Theta} |\hat{\sigma}_\theta^2 - \hat{\sigma}_{\theta'}^2|}_{(a)} + \underbrace{\max_{\theta' \in \mathcal{U}} |\hat{\sigma}_{\theta'}^2 - \mathbb{E}\,\hat{\sigma}_{\theta'}^2|}_{(b)} + \underbrace{\max_{\theta' \in \mathcal{U}} |\mathbb{E}\,\hat{\sigma}_{\theta'}^2 - \sigma_{\theta'}^2|}_{(c)} + \underbrace{\sup_{\theta \in \Theta} |\sigma_{\theta'}^2 - \sigma_\theta^2|}_{(d)}.$$

We first handle terms (a) and (d). From Assumption (B), we have

$$
\begin{aligned}
\sup_{\theta \in \Theta} |\sigma_{\theta'}^2 - \sigma_\theta^2| &\leq \sup_{\theta \in \Theta} \left| \mathbb{E}[f_{\theta'}(X,Y)^2 - f_\theta(X,Y)^2] \right| + \sup_{\theta \in \Theta} \left| (\mathbb{E} f_{\theta'}(X,Y))^2 - (\mathbb{E} f_\theta(X,Y))^2 \right| \\
&\leq \sup_{\theta \in \Theta} \mathbb{E} \left| f_{\theta'}(X,Y) - f_\theta(X,Y) \right| \left| f_{\theta'}(X,Y) + f_\theta(X,Y) \right| \\
&\quad + \sup_{\theta \in \Theta} \mathbb{E} \left| f_{\theta'}(X,Y) - f_\theta(X,Y) \right| \mathbb{E} \left| f_{\theta'}(\tilde{X},\tilde{Y}) + f_\theta(\tilde{X},\tilde{Y}) \right| \\
&\leq \sup_{\theta \in \Theta} L\|\theta' - \theta\| \cdot 2B + \sup_{\theta \in \Theta} L\|\theta' - \theta\| \cdot 2B = 4BL\rho,
\end{aligned}
$$

which yields the same bound for (a) when replacing $\mathbb{E}$ with its empirical expectation $\hat{\mathbb{E}}$.

Next, we find a high probability bound on (b). Recalling the result of Lemma C.4, and then taking the union bound over all $N$ points in $\mathcal{U}$, we get that with probability at least $1 - \delta$

$$
\max_{\theta \in \mathcal{U}} |\hat{\sigma}_\theta^2 - \mathbb{E}\,\hat{\sigma}_\theta^2| \leq 5B^2 \sqrt{\frac{1}{2n}\left( \log \frac{2}{\delta} + D \log \frac{4R}{\rho} \right)}.
$$

Finally, combining (a), (b), (d), and using Lemma C.5 to bound (c), we get that with probability at least $1 - \delta$

$$
\sup_{\theta \in \Theta} |\hat{\sigma}_\theta^2 - \sigma_\theta^2| \leq 8BL\rho + \frac{B^2}{n} + 5B^2 \sqrt{\frac{1}{2n}\left( \log \frac{2}{\delta} + D \log \frac{4R}{\rho} \right)},
$$

which, for $\rho = 1/\sqrt{n}$, yields the desired result. $\qquad\square$

**Proposition C.11** (Uniform convergence of $\hat{\xi}^2$). *For any critic family $\{f_\theta : \theta \in \Theta\}$ and parameter space $\Theta$ satisfying Assumptions (A) to (C), we have with probability at least $1 - \delta$ that*

$$
\sup_{\theta \in \Theta} |\hat{\xi}_\theta^2 - \xi_\theta^2| < \frac{8BL}{\sqrt{n}} + \frac{7B^2}{n} + \frac{10B^2}{2\sqrt{n}}\sqrt{2\log\frac{2}{\delta} + 2D\log(4R\sqrt{n})} = \tilde{\mathcal{O}}(n^{-1/2}).
$$

*Proof.* We employ a covering-number argument on the parameter space. Let $\mathcal{U} = \{\theta_i\}_{i=1..N} \subseteq \Theta$ be a $\rho$-cover of $\Theta$; i.e., any point in $\Theta$ is in the closed $\rho$-neighborhood of some point in $\mathcal{U}$. Assumption (A) ensures the minimal cover exists with at most $N = (4R/\rho)^D$ points (Cucker & Smale, 2002, Proposition 5). Now, for any $\theta \in \Theta$, let $\theta'$ be the center in $\mathcal{U}$ closest to $\theta$, and then decompose the bound into

$$
\sup_{\theta \in \Theta} |\hat{\xi}_\theta^2 - \xi_\theta^2| \leq \underbrace{\sup_{\theta \in \Theta} |\hat{\xi}_\theta^2 - \hat{\xi}_{\theta'}^2|}_{(a)} + \underbrace{\max_{\theta' \in \mathcal{U}} |\hat{\xi}_{\theta'}^2 - \mathbb{E}\,\hat{\xi}_{\theta'}^2|}_{(b)} + \underbrace{\max_{\theta' \in \mathcal{U}} |\mathbb{E}\,\hat{\xi}_{\theta'}^2 - \xi_{\theta'}^2|}_{(c)} + \underbrace{\sup_{\theta \in \Theta} |\xi_{\theta'}^2 - \xi_\theta^2|}_{(d)}.
$$

First, we bound terms (a) and (d) using Assumption (B) and the covering-number argument. Let $T_\theta = \mathbb{E}[f_\theta(X,Y)]$ and $\nu_\theta = \mathbb{E}[f_\theta(X,Y)^2]$ be the first and second moments under the null distribution, so that $\xi_\theta^2 = \nu_\theta - T_\theta^2$. It follows that

$$
\begin{aligned}
\sup_{\theta \in \Theta} |\xi_{\theta'}^2 - \xi_\theta^2| &\leq \sup_{\theta \in \Theta} |\nu_{\theta'} - \nu_\theta| + \sup_{\theta \in \Theta} |T_{\theta'}^2 - T_\theta^2| \\
&\leq \sup_{\theta \in \Theta} \mathbb{E}\,|f_{\theta'}(X,Y) - f_\theta(X,Y)||f_{\theta'}(X,Y) + f_\theta(X,Y)| + \sup_{\theta \in \Theta} |T_{\theta'} - T_\theta||T_{\theta'} + T_\theta| \\
&= 4BL \sup_{\theta \in \Theta} \|\theta' - \theta\| = 4BL\rho.
\end{aligned}
$$

We obtain the same bound for (a) by replacing $\mathbb{E}$ with its V-statistic estimator.

Next, we bound (b) using Lemma C.6 and a uniform bound over all $N$ points in the $\rho$-cover. We get that with probability at least $1 - \delta$

$$
\max_{\theta \in \mathcal{U}} |\hat{\xi}_\theta^2 - \mathbb{E}\,\hat{\xi}_\theta^2| < 10B^2 \sqrt{\frac{1}{2n}\left( \log \frac{2}{\delta} + D \log \frac{4R}{\rho} \right)}.
$$

Finally, combining (a), (b), (d), and using Lemma C.7 to bound (c), we get that with probability at least $1 - \delta$

$$\sup_{\theta \in \Theta} |\hat{\xi}_\theta^2 - \xi_\theta^2| < 8BL\rho + \frac{7B^2}{n} + 10B^2 \sqrt{\frac{1}{2n}\left(\log\frac{2}{\delta} + D\log\frac{4R}{\rho}\right)},$$

which, for $\rho = 1/\sqrt{n}$, yields the desired result. $\qquad\square$

### C.5 Main Results

**Theorem C.12** (Uniform convergence of $\hat{J}_{\text{NDS}}^\lambda$)*. Let $\{f_\theta : \theta \in \Theta\}$ be a critic family with parameter space $\Theta$ satisfying Assumptions (A) to (C). Suppose $\sigma_\theta^2 \geq s^2$ and $\xi_\theta^2 \geq s^2$ for some positive $s$, and $\lambda = n^{-1/3}$. Then, with probability at least $1 - \delta$,*

$$\sup_{\theta \in \Theta} |\hat{J}_{NDS}^\lambda - J_{NDS}| < \frac{2B}{s^3 n^{1/3}} + \frac{16B^2 L}{s^2 n^{1/3}} + \frac{5B^3}{s^2 n^{1/3}}\sqrt{2\log\frac{12}{\delta} + 2D\log(4R\sqrt{n})} + \mathrm{o}(n^{-1/3}),$$

*and thus*

$$\sup_{\theta \in \Theta} |\hat{J}_{NDS}^\lambda - J_{NDS}| = \tilde{\mathcal{O}}_P\left(\frac{1}{s^2 n^{1/3}}\left[\frac{B}{s} + B^2 L + B^3 \sqrt{D}\right]\right).$$

*Proof.* Let $z_\alpha$ denote the standard normal $\alpha$-quantile. We start by making the following decomposition:

$$\sup_{\theta \in \Theta} |\hat{J}_{\text{NDS}}^\lambda - J_{\text{NDS}}| = \sup_{\theta \in \Theta}\left| \frac{\hat{T}_{1,\theta} - \hat{T}_{0,\theta}}{\hat{\sigma}_{\theta,\lambda}} - \frac{T_{1,\theta} - T_{0,\theta}}{\sigma_\theta} + \frac{\xi_\theta}{\sqrt{n}\,\sigma_\theta}z_{1-\alpha} - \frac{\hat{\xi}_{\theta,\lambda}}{\sqrt{n}\,\hat{\sigma}_{\theta,\lambda}}z_{1-\alpha} \right|$$

$$\leq \underbrace{\sup_{\theta \in \Theta}\left|\frac{\hat{T}_{1,\theta}}{\hat{\sigma}_{\theta,\lambda}} - \frac{T_{1,\theta}}{\sigma_\theta}\right|}_{(a)} + \underbrace{\sup_{\theta \in \Theta}\left|\frac{\hat{T}_{0,\theta}}{\hat{\sigma}_\theta} - \frac{T_{0,\theta}}{\sigma_\theta}\right|}_{(b)} + \frac{z_{1-\alpha}}{\sqrt{n}}\underbrace{\sup_{\theta \in \Theta}\left|\frac{\hat{\xi}_{\theta,\lambda}}{\hat{\sigma}_{\theta,\lambda}} - \frac{\xi_\theta}{\sigma_\theta}\right|}_{(c)}.$$

Define $\sigma_{\theta,\lambda}^2 = \sigma_\theta^2 + \lambda$ and notice that $\hat{\sigma}_\theta^2 \geq 0$ since it is the biased variance estimator; this implies that $\sigma_{\theta,\lambda}^2 = \sigma_\theta^2 + \lambda \geq s^2 + \lambda$ and $\hat{\sigma}_{\theta,\lambda}^2 = \hat{\sigma}_\theta^2 + \lambda \geq \lambda$.

Now, we rewrite (a) (with subscripts $\theta$ omitted) as

$$\sup_{\theta \in \Theta}\left|\frac{\hat{T}_1}{\hat{\sigma}_\lambda} - \frac{T_1}{\sigma}\right| \leq \sup_{\theta \in \Theta}\left|\frac{\hat{T}_1}{\hat{\sigma}_\lambda} - \frac{\hat{T}_1}{\sigma_\lambda}\right| + \sup_{\theta \in \Theta}\left|\frac{\hat{T}_1}{\sigma_\lambda} - \frac{\hat{T}_1}{\sigma}\right| + \sup_{\theta \in \Theta}\left|\frac{\hat{T}_1}{\sigma} - \frac{T_1}{\sigma}\right|$$

$$= \sup_{\theta \in \Theta}|\hat{T}_1|\left|\frac{\hat{\sigma}_\lambda^2 - \sigma_\lambda^2}{\hat{\sigma}_\lambda\sigma_\lambda(\hat{\sigma}_\lambda + \sigma_\lambda)}\right| + \sup_{\theta \in \Theta}|\hat{T}_1|\left|\frac{\sigma_\lambda^2 - \sigma^2}{\sigma_\lambda\sigma(\sigma_\lambda + \sigma)}\right| + \sup_{\theta \in \Theta}\frac{1}{\sigma}|\hat{T}_1 - T_1|$$

$$\leq \frac{B}{\lambda\sqrt{s^2 + \lambda} + \sqrt{\lambda}(s^2 + \lambda)}\sup_{\theta \in \Theta}|\hat{\sigma}^2 - \sigma^2| + \frac{B\lambda}{s(s^2 + \lambda) + s^2\sqrt{s^2 + \lambda}} + \frac{1}{s}\sup_{\theta \in \Theta}|\hat{T}_1 - T_1|$$

$$< \frac{B}{\sqrt{\lambda}s^2}\sup_{\theta \in \Theta}|\hat{\sigma}^2 - \sigma^2| + \frac{B\lambda}{s^3} + \frac{1}{s}\sup_{\theta \in \Theta}|\hat{T}_1 - T_1|,$$

where the first term in the last line uses the slowest growing upper bound as $\lambda \to 0$.

Employ Propositions C.8 and C.10 as uniform bounds on $|\hat{T}_1 - T_1|$ and $|\hat{\sigma}^2 - \sigma^2|$ respectively, then take the union bound to give us, with probability at least $1 - \delta$,

$$
\begin{aligned}
\sup_{\theta \in \Theta} \left| \frac{\hat{T}_1}{\hat{\sigma}_\lambda} - \frac{T_1}{\sigma} \right| &< \frac{8B^2 L}{s^2 \sqrt{n\lambda}} + \frac{B^3}{s^2 n \sqrt{\lambda}} + \frac{5B^3}{2s^2 \sqrt{n\lambda}} \sqrt{2 \log \frac{4}{\delta} + 2D \log(4R\sqrt{n})} \\
&\quad + \frac{B\lambda}{s^3} \\
&\quad + \frac{2L}{s\sqrt{n}} + \frac{B}{s\sqrt{n}} \sqrt{2 \log \frac{4}{\delta} + 2D \log(4R\sqrt{n})} \\
&= \frac{B\lambda}{s^3} + \frac{8B^2 L}{s^2 \sqrt{n\lambda}} + \frac{2L}{s\sqrt{n}} + \frac{B^3}{s^2 n \sqrt{\lambda}} + \left( \frac{5B^3}{2s^2 \sqrt{n\lambda}} + \frac{B}{s\sqrt{n}} \right) \sqrt{2 \log \frac{4}{\delta} + 2D \log(4R\sqrt{n})}.
\end{aligned}
$$

The best-case (fastest to decay) asymptotic bound follows when $\lambda = n^{-1/3}$, which yields with probability at least $1 - \delta$,

$$
\sup_{\theta \in \Theta} \left| \frac{\hat{T}_1}{\hat{\sigma}_\lambda} - \frac{T_1}{\sigma} \right| < \frac{B}{s^3 n^{1/3}} + \frac{8B^2 L}{s^2 n^{1/3}} + \frac{5B^3}{2s^2 n^{1/3}} \sqrt{2 \log \frac{4}{\delta} + 2D \log(4R\sqrt{n})} + o(n^{-1/3}).
$$

Term (b) can be handled in a similar manner, except we use Proposition C.9 as a uniform bound on $|\hat{T}_0 - T_0|$. It turns out this gives us the same asymptotic bound as (a).

For term (c), we first define $\xi_{\theta,\lambda} = \xi_\theta^2 + \lambda$. Since $\hat{\xi}_\theta^2$ is the biased estimator we have $\hat{\xi}_\theta^2 \geq 0$. Also, $\xi_{\theta,\lambda}^2 = \xi_\theta^2 + \lambda \geq s^2 + \lambda$ and $\hat{\xi}_{\theta,\lambda} = \hat{\xi}_\theta^2 + \lambda \geq \lambda$ so that

$$
\begin{aligned}
\sup_{\theta \in \Theta} \left| \frac{\hat{\xi}_\lambda}{\hat{\sigma}_\lambda} - \frac{\xi}{\sigma} \right| &\leq \sup_{\theta \in \Theta} \left| \frac{\hat{\xi}_\lambda}{\hat{\sigma}_\lambda} - \frac{\hat{\xi}}{\hat{\sigma}_\lambda} \right| + \sup_{\theta \in \Theta} \left| \frac{\hat{\xi}}{\hat{\sigma}_\lambda} - \frac{\hat{\xi}}{\sigma_\lambda} \right| + \sup_{\theta \in \Theta} \left| \frac{\hat{\xi}}{\sigma_\lambda} - \frac{\hat{\xi}}{\sigma} \right| + \sup_{\theta \in \Theta} \left| \frac{\hat{\xi}}{\sigma} - \frac{\xi}{\sigma} \right| \\
&= \sup_{\theta \in \Theta} \left| \frac{\hat{\xi}_\lambda^2 - \hat{\xi}^2}{\hat{\sigma}_\lambda (\hat{\xi}_\lambda + \hat{\xi})} \right| + \sup_{\theta \in \Theta} |\hat{\xi}| \left| \frac{\hat{\sigma}_\lambda^2 - \sigma_\lambda^2}{\hat{\sigma}_\lambda \sigma_\lambda (\hat{\sigma}_\lambda + \sigma_\lambda)} \right| + \sup_{\theta \in \Theta} |\hat{\xi}| \left| \frac{\sigma_\lambda^2 - \sigma^2}{\sigma_\lambda \sigma (\sigma_\lambda + \sigma)} \right| + \sup_{\theta \in \Theta} \left| \frac{\hat{\xi}^2 - \xi^2}{\sigma (\hat{\xi} + \xi)} \right| \\
&\leq 1 + \frac{B}{\lambda \sqrt{s^2 + \lambda} + \sqrt{\lambda}(s^2 + \lambda)} \sup_{\theta \in \Theta} |\hat{\sigma}^2 - \sigma^2| + \frac{B\lambda}{s(s^2 + \lambda)} + \frac{1}{s^2} \sup_{\theta \in \Theta} \left| \hat{\xi}^2 - \xi^2 \right| \\
&< 1 + \frac{B}{\sqrt{\lambda} s^2} \sup_{\theta \in \Theta} |\hat{\sigma}^2 - \sigma^2| + \frac{B\lambda}{s^3} + \frac{1}{s^2} \sup_{\theta \in \Theta} \left| \hat{\xi}^2 - \xi^2 \right|.
\end{aligned}
$$

Propositions C.10 and C.11 are uniform bounds on $|\hat{\sigma}^2 - \sigma^2|$ and $|\hat{\xi}^2 - \xi^2|$. Then, taking the union bound gives us, with probability at least $1 - \delta$,

$$
\begin{aligned}
\sup_{\theta \in \Theta} \left| \frac{\hat{\xi}_\lambda}{\hat{\sigma}_\lambda} - \frac{\xi}{\sigma} \right| &< 1 + \frac{B\lambda}{s^3} + \left[ \frac{B}{s^2 \sqrt{\lambda}} + \frac{1}{s^2} \right] \frac{8BL}{\sqrt{n}} + \left[ \frac{B}{s^2 \sqrt{\lambda}} + \frac{7}{s^2} \right] \frac{B^2}{n} \\
&\quad + \left[ \frac{B}{s^2 \sqrt{\lambda}} + \frac{2}{s^2} \right] \frac{5B^2}{2\sqrt{n}} \sqrt{2 \log \frac{4}{\delta} + 2D \log(4R\sqrt{n})}.
\end{aligned}
$$

When $\lambda = n^{-1/3}$, we have $1/\sqrt{n\lambda} = n^{-1/3}$ and $1/(n\sqrt{\lambda}) = n^{-5/6}$ so that the overall order of this bound is $O(1)$. Then, multiplying by the factor $z_{1-\alpha}/\sqrt{n}$ in front of (c) gives us order $O(n^{-1/2})$.

Finally, taking the union bound of (a), (b), and (c) gives us, with probability at least $1 - \delta$,

$$
\sup_{\theta \in \Theta} \left| \hat{J}_{\text{NDS}}^\lambda - J_{\text{NDS}} \right| < \frac{2B}{s^3 n^{1/3}} + \frac{16B^2 L}{s^2 n^{1/3}} + \frac{5B^3}{s^2 n^{1/3}} \sqrt{2 \log \frac{12}{\delta} + 2D \log(4R\sqrt{n})} + o(n^{-1/3}).
$$

$\square$

**Theorem C.13** (Consistency [van der Vaart, 1998, Theorem 5.7]). *Let $\hat{J}_n$ be the finite sample estimate of $J$ with $n$ samples, as a function of the parameters $\theta$, and suppose that $J_n$ satisfies uniform convergence*

$$\sup_{\theta \in \Theta} |\hat{J}_n(\theta) - J(\theta)| \xrightarrow{P} 0.$$

*If $\theta^*$ is a well-separated maximum of $J$, then any sequence of estimators $\hat{\theta}_n$ approximately maximizing $\hat{J}_n$, i.e.*

$$\hat{J}_n(\theta^*) - \hat{J}_n(\hat{\theta}_n) \leq o_P(1),$$

*is a consistent estimator, i.e. $\hat{\theta}_n \xrightarrow{p} \theta^*$.*

## D  Uniform Convergence: HSIC

### D.1  Preliminaries

We start by defining the notation used in the proofs. Let kernels $k_\omega$ and $l_\gamma$ be parameterized by some $\omega \in \Omega$ and $\gamma \in \Gamma$, with $\Theta \subseteq \Omega \times \Gamma$ as the joint parameter space. The samples $(X_i, Y_i) \sim \mathbb{P}_{xy}$ are drawn i.i.d. from the joint distribution. We denote the $n \times n$ gram matrices of $k_\omega$ and $l_\gamma$ by $K^{(\omega)}$ and $L^{(\gamma)}$ respectively. We will often omit the kernel parameters $\omega$ and $\gamma$ when it is clear from the context.

Let $\eta$ be the HSIC test statistic and $\hat{\eta}$ to be its U-statistic estimator given by

$$\hat{\eta} = \frac{1}{(n)_4} \sum_{(i,j,q,r) \in \mathbf{i}_4^n} H_{ijqr},$$

where $(n)_k = n!/(n-k)!$ is the Pochhammer symbol and $\mathbf{i}_4^n$ is all possible 4-tuples drawn without replacement from 1 to n. $H$ is the kernel gram matrix of the U-statistic defined by

$$H_{ijqr} = \frac{1}{4!} \sum_{(a,b,c,d)}^{(i,j,q,r)} K_{ab} \left( L_{ab} + L_{cd} - 2L_{ac} \right),$$

where sum represents all 4! combinations of tuples $(a, b, c, d)$ that can be selected without replacement from $(i, j, q, r)$.

### D.2  Assumptions

Our main uniform convergence results require the following assumptions.

(A') The set of kernel parameters $\Omega$ lies in a Banach space of dimension $D_\Omega$, and the set of kernel parameters $\Gamma$ lies in a Banach space of dimension $D_\Gamma$. Furthermore, each parameter space is bounded by $R_\Omega$ and $R_\Gamma$ respectively, i.e.,

$$\Omega \subseteq \{\omega \mid \|\omega\| \leq R_\Omega\},$$
$$\Gamma \subseteq \{\gamma \mid \|\gamma\| \leq R_\Gamma\}.$$

(B') Both kernels $k$ and $l$ are Lipschitz with respect to the parameter space: for all $x, x' \in \mathcal{X}$ and $\omega, \omega' \in \Omega$

$$|k_\omega(x, x') - k_{\omega'}(x, x')| \leq L_k \|\omega - \omega'\|,$$

and for all $y, y' \in \mathcal{Y}$ and $\gamma, \gamma' \in \Gamma$

$$|l_\gamma(y, y') - l_{\gamma'}(y, y')| \leq L_l \|\gamma - \gamma'\|.$$

(C') The kernels $k_\omega$ and $l_v$ are uniformly bounded:

$$\sup_{\omega \in \Omega} \sup_{x \in \mathcal{X}} k_\omega(x, x) \leq \nu_k,$$
$$\sup_{\gamma \in \Gamma} \sup_{x \in \mathcal{Y}} l_\gamma(y, y) \leq \nu_l.$$

For the kernels we use in practice, $\nu_k = \nu_l = 1$.

### D.3 Lemmas

**Lemma D.1** (Concentration of $\hat{\sigma}^2_{\omega,\gamma}$). *For any kernels $k$ and $l$ satisfying Assumption (A'), with probability at least $1 - \delta$ we have*

$$|\hat{\sigma}^2 - \mathbb{E}\,\hat{\sigma}^2| \leq 6144\nu_k^2\nu_l^2\sqrt{\frac{2}{n}\log\frac{2}{\delta}}.$$

*Proof.* We apply McDiarmid's inequality to $\hat{\sigma}^2$. First, we show that the variance estimator satisfies bounded differences. For convenience, we denote $(i,j,q,r) \in \mathbf{i}_4^n$ simply as $(i,j,q,r)$, and $(i,j,q)\backslash k$ to be the set of 3-tuples drawn without replacement from $\mathbf{i}_3^n$ that exclude the number $k$. Recall that

$$\hat{\sigma}^2 = 16\left(\frac{1}{(n)_4(n-1)_3}\sum_{\substack{(i,j,q,r)\\(b,c,d)\backslash i}} H_{ijqr}H_{ibcd} - \hat{\eta}^2\right).$$

Let $F$ denote the kernel tensor $H$ but with sample $(X_\ell, Y_\ell)$ replaced by $(X'_\ell, Y'_\ell)$ so that $F$ agrees with $H$ except at indices $\ell$, and let $\hat{\eta}'$ and $\hat{\sigma}'^2$ denote the HSIC and its variance estimators according to this updated sample set. The deviation is then

$$|\hat{\sigma}^2 - \hat{\sigma}'^2| \leq \frac{16}{(n)_4(n-1)_3}\sum_{\substack{(i,j,q,r)\\(b,c,d)\backslash i}} |H_{ijqr}H_{ibcd} - F_{ijqr}F_{ibcd}| + 16|\hat{\eta}^2 - \hat{\eta}'^2|.$$

We bound the first term by noticing that $\Delta := H_{ijqr}H_{ibcd} - F_{ijqr}F_{ibcd}$ is zero when none of the indices $\{i,j,q,r,b,c,d\}$ is $\ell$. Let $S := \{(i,j,q,r,b,c,d) : (i,j,q,r) \in \mathbf{i}_4^n, (b,c,d) \in \mathbf{i}_3^n\backslash\{i\}, \ell \in \{i,j,q,r,b,c,d\}\}$ be the set of indices where $\Delta$ may be non-zero. By Assumption (A') we know that $|\Delta| \leq 32\nu_k^2\nu_l^2$. Thus, we can bound the first term by

$$\frac{16}{(n)_4(n-1)_3}\sum_S |\Delta| = \frac{512\nu_k^2\nu_l^2}{(n)_4(n-1)_3}|S|$$

$$= \frac{512\nu_k^2\nu_l^2}{(n)_4(n-1)_3}\left[\underbrace{4(n-1)_3^2}_{\ell\in\{i,j,q,r\}} + \underbrace{3(n-1)(n-2)_2(n-1)_3}_{\ell\in\{b,c,d\}} - \underbrace{9(n-1)_3(n-2)_2}_{\ell\in\{j,q,r\}\text{ and }\ell\in\{b,c,d\}}\right]$$

$$= 512\nu_k^2\nu_l^2\left(\frac{16}{n} - \frac{9}{n-1}\right)$$

$$\leq \frac{8192\nu_k^2\nu_l^2}{n} \quad (\forall n > 1).$$

We can bound the second term using $|\hat{\eta}| \leq 4\nu_k\nu_l$ and the bounded difference result (15) from Proposition D.3:

$$16|\hat{\eta}^2 - \hat{\eta}'^2| = 16|\hat{\eta} + \hat{\eta}'||\hat{\eta} - \hat{\eta}'| \leq 16 \cdot 8\nu_k\nu_l \cdot \frac{32\nu_k\nu_l}{n} = \frac{4096\nu_k^2\nu_l^2}{n}.$$

Combining these two terms, the maximal bounded difference for $\hat{\sigma}^2$ is

$$|\hat{\sigma}^2 - \hat{\sigma}'^2| \leq \frac{12288\nu_k^2\nu_l^2}{n}.$$

Finally, applying McDiarmid's inequality gives us, with probability at least $1 - \delta$,

$$|\hat{\sigma}^2 - \mathbb{E}\,\hat{\sigma}^2| \leq 6144\nu_k^2\nu_l^2\sqrt{\frac{2}{n}\log\frac{2}{\delta}}. \qquad \square$$

**Lemma D.2** (Bias of $\hat{\sigma}^2_{\omega,\gamma}$). *For any kernels $k$ and $l$ satisfying Assumption (A'), the bias is bounded by*

$$|\mathbb{E}\,\hat{\sigma}^2 - \sigma^2| \leq \frac{4608\nu_k^2\nu_l^2}{n}.$$

*Proof.* The expectation of the variance estimator is

$$\mathbb{E}\,\hat\sigma^2 = 16 \left( \frac{1}{(n)_4(n-1)_3} \sum_{\substack{(i,j,q,r) \\ (b,c,d)\setminus i}} \mathbb{E}[H_{ijqr}H_{ibcd}] - \frac{1}{(n)_4^2} \sum_{\substack{(i,j,q,r) \\ (a,b,c,d)}} \mathbb{E}[H_{ijqr}H_{abcd}] \right).$$

First, we can break down the left-hand sum into only terms of $\mathbb{E}[H_{1234}H_{1567}]$ by considering the cases where $\{i,j,q,r,b,c,d\}$ are unique. Let $S = \{(i,j,q,r,b,c,d) : (i,j,q,r) \in \mathbf{i}_4^n, (b,c,d) \in \mathbf{i}_3^n\setminus\{i\}\}$ be the set of all possible indices of our left-hand sum. It follows that

$$\sum_S \mathbb{E}[H_{ijqr}H_{ibcd}] = \sum_{(i,j,q,r,b,c,d)\in\mathbf{i}_7^n} \mathbb{E}[H_{ijqr}H_{ibcd}] + \sum_{S\setminus\mathbf{i}_7^n} \mathbb{E}[H_{ijqr}H_{ibcd}].$$

If all indices are unique, then the expectation $\mathbb{E}[H_{ijqr}H_{ibcd}]$ is equivalent to $\mathbb{E}[H_{1234}H_{1567}]$; otherwise, we can bound the expectation by $16\nu_k^2\nu_l^2$ via Assumption (A'). Thus, the bound on the left-hand sum is

$$\sum_{\substack{(i,j,q,r) \\ (b,c,d)\setminus i}} \mathbb{E}[H_{ijqr}H_{ibcd}] \le (n)_7\,\mathbb{E}[H_{1234}H_{1567}] + \Big((n)_4(n-1)_3 - (n)_7\Big)16\nu_k^2\nu_l^2.$$

Similarly, we can break down the right-hand sum into only terms of $\mathbb{E}[H_{1234}H_{5678}]$. Let $R = \{(i,j,q,r,a,b,c,d) : (i,j,q,r) \in \mathbf{i}_4^n, (a,b,c,d) \in \mathbf{i}_4^n\}$ be the possible indices of our right-hand sum. We have that

$$\sum_{\substack{(i,j,q,r) \\ (a,b,c,d)}} \mathbb{E}[H_{ijqr}H_{abcd}] = \sum_{(i,j,q,r,a,b,c,d)\in\mathbf{i}_8^n} \mathbb{E}[H_{ijqr}H_{abcd}] + \sum_{R\setminus\mathbf{i}_8^n} \mathbb{E}[H_{ijqr}H_{abcd}]$$

$$\le (n)_8\,\mathbb{E}[H_{1234}H_{5678}] + \Big((n)_4^2 - (n)_8\Big)16\nu_k^2\nu_l^2.$$

Now, using these two results and Assumption (A'), we can compute a bound on the desired bias of $\hat\sigma^2$:

$$|\,\mathbb{E}\,\hat\sigma^2 - \sigma^2| = 16 \left| \frac{1}{(n)_4(n-1)_3} \sum_{\substack{(i,j,q,r) \\ (b,c,d)\setminus i}} \mathbb{E}[H_{ijqr}H_{ibcd}] - \mathbb{E}[H_{1234}H_{1567}] - \frac{1}{(n)_4^2} \sum_{\substack{(i,j,q,r) \\ (a,b,c,d)}} \mathbb{E}[H_{ijqr}H_{abcd}] + \mathbb{E}[H_{1234}H_{5678}] \right|$$

$$\le 16 \left| \left(1 - \frac{(n)_7}{(n)_4(n-1)_3}\right) \left(16\nu_k^2\nu_l^2 - \underbrace{\mathbb{E}[H_{1234}H_{1567}]}_{-16\nu_k^2\nu_l^2 \le \cdot \le 16\nu_k^2\nu_l^2}\right) + \left(1 - \frac{(n)_8}{(n)_4^2}\right) \left(\underbrace{\mathbb{E}[H_{1234}H_{5678}]}_{0\le\cdot\le16\nu_k^2\nu_l^2} - 16\nu_k^2\nu_l^2\right) \right|.$$

$$\le 16 \left(1 - \frac{(n)_7}{(n)_4(n-1)_3}\right) 32\nu_k^2\nu_l^2 < 512\nu_k^2\nu_l^2 \cdot \frac{9}{n} \quad (\forall n \ge 4)$$

$$= \frac{4608\nu_k^2\nu_l^2}{n}. \qquad \square$$

## D.4 Propositions

**Proposition D.3** (Uniform convergence of $\widehat{\mathrm{HSIC}}_u$). *Under Assumptions (A') to (C'), we have that with probability at least $1 - \delta$,*

$$\sup_{\theta\in\Theta} |\hat\eta_\theta - \eta_\theta| \le \frac{8\nu_k\nu_l}{\sqrt{n}} \left( \frac{L_k}{\nu_k} + \frac{L_l}{\nu_l} + 2\sqrt{2\log\frac{2}{\delta} + 2D_\Omega\log(4R_\Omega\sqrt{n}) + 2D_\Gamma\log(4R_\Gamma\sqrt{n})} \right).$$

*Proof.* We use $\epsilon$-net arguments on both spaces $\Omega$ and $\Gamma$. Let $\{\omega_i\}_{i=1}^{T_\Omega}$ be arbitrarily placed centers with radius $\rho_\Omega$ such that any point $\omega \in \Omega$ satisfies $\min\|\omega - \omega_i\| \le \rho_\Omega$. Similarly, let $\{\gamma_i\}_{i=1}^{T_\Gamma}$ be centers with

radius $\rho_\Gamma$ satisfying $\min \|\gamma - \gamma_i\| \le \rho_\Gamma$ for any $\gamma \in \Gamma$. Assumption (A') ensures this is possible with at most $T_\Omega = (4R_\Omega/\rho_\Omega)^{D_\Omega}$ and $T_\Gamma = (4R_\Gamma/\rho_\Gamma)^{D_\Gamma}$ points respectively (Cucker & Smale, 2002, Proposition 5).

We can decompose the convergence bound into simpler components and tackle each component individually

$$\sup_{\theta \in \Theta} |\hat{\eta}_\theta - \eta_\theta| \le \sup_\theta |\hat{\eta}_\theta - \hat{\eta}_{\theta'}| + \max_{\substack{\omega' \in \{\omega_1,...,\omega_{T_\Omega}\} \\ \gamma' \in \{\gamma_1,...,\gamma_{T_\Gamma}\}}} |\hat{\eta}_{\theta'} - \eta_{\theta'}| + \sup_\theta |\eta_{\theta'} - \eta_\theta|.$$

First, let us analyze $|\eta_\theta - \eta_{\theta'}|$ for any $\theta, \theta' \in \Theta$. Recall that $\eta = \mathbb{E}[H_{1234}]$ where $H_{1234} = \frac{1}{4!} \sum_{(a,b,c,d)}^{(1,2,3,4)} K_{ab}(L_{ab} + L_{cd} - 2L_{ac})$. We have that

$$|H_{1234}^{(\theta)} - H_{1234}^{(\theta')}| \le \frac{1}{4!} \sum_{(abcd)}^{(1234)} \left| K_{ab}^{(\omega)}(L_{ab}^{(\gamma)} + L_{cd}^{(\gamma)} - 2L_{ac}^{(\gamma)}) - K_{ab}^{(\omega')}(L_{ab}^{(\gamma')} + L_{cd}^{(\gamma')} - 2L_{ac}^{(\gamma')}) \right|$$

$$\le \frac{1}{4!} \sum_{(abcd)}^{(1234)} \left( \left| K_{ab}^{(\omega)}L_{ab}^{(\gamma)} - K_{ab}^{(\omega')}L_{ab}^{(\gamma')} \right| + \left| K_{ab}^{(\omega)}L_{cd}^{(\gamma)} - K_{ab}^{(\omega')}L_{cd}^{(\gamma')} \right| + 2\left| K_{ab}^{(\omega')}L_{ac}^{(\gamma')} - K_{ab}^{(\omega)}L_{ac}^{(\gamma)} \right| \right).$$

From Assumption (A') we know that $|K_{ab}| \le \nu_k$ and $|L_{ab}| \le \nu_l$, and via Assumption (B') we notice that

$$\left| K_{ab}^{(\omega)}L_{ab}^{(\gamma)} - K_{ab}^{(\omega')}L_{ab}^{(\gamma')} \right| = \left| K_{ab}^{(\omega)}L_{ab}^{(\gamma)} - K_{ab}^{(\omega)}L_{ab}^{(\gamma')} + K_{ab}^{(\omega)}L_{ab}^{(\gamma')} - K_{ab}^{(\omega')}L_{ab}^{(\gamma')} \right|$$

$$\le \left| K_{ab}^{(\omega)} \right| \left| L_{ab}^{(\gamma)} - L_{ab}^{(\gamma')} \right| + \left| L_{ab}^{(\gamma')} \right| \left| K_{ab}^{(\omega)} - K_{ab}^{(\omega')} \right|$$

$$\le \nu_k L_l \|v - v'\| + \nu_l L_k \|\omega - \omega'\|$$

$$\le \nu_k L_l \rho_\Gamma + \nu_l L_k \rho_\Omega.$$

This expression is true for all three components of $|H_{1234}^{(\theta)} - H_{1234}^{(\theta')}|$ and so it follows that

$$|\eta_\theta - \eta_{\theta'}| = \left| \mathbb{E}[H_{1234}^{(\theta)}] - \mathbb{E}[H_{1234}^{(\theta')}] \right| \le \mathbb{E} \left| H_{1234}^{(\theta)} - H_{1234}^{(\theta')} \right| \le 4\nu_k L_l \rho_\Gamma + 4\nu_l L_k \rho_\Omega,$$

$$|\hat{\eta}_\theta - \hat{\eta}_{\theta'}| = \left| \frac{1}{(n)_4} \sum_{(i,j,q,r) \in \mathbf{i}_4^n} H_{ijqr}^{(\theta)} - H_{ijqr}^{(\theta')} \right| \le \frac{1}{(n)_4} \sum_{(i,j,q,r) \in \mathbf{i}_4^n} \left| H_{1234}^{(\theta)} - H_{1234}^{(\theta')} \right| \le 4\nu_k L_l \rho_\Gamma + 4\nu_l L_k \rho_\Omega.$$

Now, we study the random error function $\Delta := \hat{\eta} - \eta$. Note that $\mathbb{E}\Delta = 0$ since $\hat{\eta}$ is unbiased, and $|H_{ijqr}| \le 4\nu_k\nu_l$ via Assumption (A'). This $\hat{\eta}$, and hence $\Delta$, satisfies bounded differences. Let $F$ denote the kernel tensor $H$ but with sample $(X_\ell, Y_\ell)$ replaced by $(X'_\ell, Y'_\ell)$ so that $F$ agrees with $H$ except at indicies $\ell$, and let $\hat{\eta}' = \frac{1}{(n)_4} \sum_{(i,j,q,r) \in \mathbf{i}_4^n} F_{ijqr}$ be it's HSIC estimator.

For convenience, we denote $(i,j,q,r) \in \mathbf{i}_4^n$ simply as $(i,j,q,r)$, and $(i,j,q)\backslash k$ to be the set of 3-tuples drawn without replacement from $\mathbf{i}_3^n$ that exclude the number $k$. We can compute the maximal bounded difference $|\Delta - \Delta'| = |\hat{\eta} - \hat{\eta}'|$ as

$$|\hat{\eta} - \hat{\eta}'| = \left| \frac{1}{(n)_4} \sum_{(i,j,q,r)} H_{ijqr} - F_{ijqr} \right| \le \frac{1}{(n)_4} \sum_{(i,j,q,r)} |H_{ijqr} - F_{ijqr}| \tag{15}$$

$$= \frac{1}{(n)_4} \left( \sum_{(j,q,r)\backslash\ell} \underbrace{|H_{\ell jqr} - F_{\ell jqr}|}_{\le 8\nu_k\nu_l} + \sum_{(i,q,r)\backslash\ell} |H_{i\ell qr} - F_{i\ell qr}| + \sum_{(i,j,r)\backslash\ell} |H_{ij\ell r} - F_{ij\ell r}| + \sum_{(i,j,q)\backslash\ell} |H_{ijq\ell} - F_{ijq\ell}| \right)$$

$$= \frac{1}{(n)_4} \left( (n-1)_3 \cdot 8\nu_k\nu_l \cdot 4 \right) = \frac{32\nu_k\nu_l}{n}.$$

Then, applying McDiarmid's inequality on $\Delta := \hat{\eta} - \eta$ followed by a union bound over the $T_\Omega T_\Gamma$ center pairs gives us, with probability at least $1 - \delta$, that

$$\max_{\substack{\omega' \in \{\omega_1, \ldots, \omega_{T_\Omega}\} \\ \gamma' \in \{\gamma_1, \ldots, \gamma_{T_\Gamma}\}}} |\hat{\eta}_{\theta'} - \eta_{\theta'}| \le 32\nu_k\nu_l \sqrt{\frac{1}{2n} \log \frac{2T_\Omega T_\Gamma}{\delta}}$$

$$= \frac{16\nu_k\nu_l}{\sqrt{n}} \sqrt{2\log\frac{2}{\delta} + 2\log T_\Omega + 2\log T_\Gamma}$$

$$= \frac{16\nu_k\nu_l}{\sqrt{n}} \sqrt{2\log\frac{2}{\delta} + 2D_\Omega \log \frac{4R_\Omega}{\rho_\Omega} + 2D_\Gamma \log \frac{4R_\Gamma}{\rho_\Gamma}}.$$

Finally, we combine these results to get our uniform convergence bound:

$$\sup_{\theta \in \Theta} |\hat{\eta}_\theta - \eta_\theta| \le 8\nu_k L_l \rho_\Gamma + 8\nu_l L_k \rho_\Omega + \frac{16\nu_k\nu_l}{\sqrt{n}} \sqrt{2\log\frac{2}{\delta} + 2D_\Omega \log \frac{4R_\Omega}{\rho_\Omega} + 2D_\Gamma \log \frac{4R_\Gamma}{\rho_\Gamma}}$$

$$= 8\nu_k\nu_l \left( \frac{L_k}{\nu_k}\rho_\Omega + \frac{L_l}{\nu_l}\rho_\Gamma + \frac{2}{\sqrt{n}} \sqrt{2\log\frac{2}{\delta} + 2D_\Omega \log \frac{4R_\Omega}{\rho_\Omega} + 2D_\Gamma \log \frac{4R_\Gamma}{\rho_\Gamma}} \right).$$

Setting $\rho_\Omega = \rho_\Gamma = 1/\sqrt{n}$ yields the desired result. $\qquad\square$

**Proposition D.4** (Uniform convergence of $\hat{\sigma}^2_{\omega,\gamma}$). *Under Assumptions (A') to (C'), we have that with probability at least $1 - \delta$,*

$$\sup_{\theta \in \Theta} |\hat{\sigma}^2_\theta - \sigma^2_\theta| \le \frac{2048\nu_k^2\nu_l^2}{\sqrt{n}} \left( \frac{L_k}{\nu_k} + \frac{L_l}{\nu_l} + 3\sqrt{2\log\frac{2}{\delta} + 2D_\Omega \log(4R_\Omega\sqrt{n}) + 2D_\Gamma \log(4R_\Gamma\sqrt{n})} + \frac{9}{4\sqrt{n}} \right).$$

*Proof.* We use an $\epsilon$-net argument on both spaces $\Omega$ and $\Gamma$. Using the same construction as in Proposition D.3, we once again decompose our convergence bound:

$$\sup_{\theta \in \Theta} |\hat{\sigma}^2_\theta - \sigma^2_\theta| \le \sup_\theta |\hat{\sigma}^2_\theta - \hat{\sigma}^2_{\theta'}| + \max_{\substack{\omega' \in \{\omega_1, \ldots, \omega_{T_\Omega}\} \\ \gamma' \in \{\gamma_1, \ldots, \gamma_{T_\Gamma}\}}} |\hat{\sigma}^2_{\theta'} - \sigma^2_{\theta'}| + \sup_\theta |\sigma^2_{\theta'} - \sigma^2_\theta|.$$

First, let us analyze $|\sigma^2_\theta - \sigma^2_{\theta'}|$ for any $\theta, \theta' \in \Theta$. Recall that $\sigma^2 = 16\left(\mathbb{E}[H_{1234}H_{1567}] - \eta^2\right)$. It follows that

$$|\sigma^2_\theta - \sigma^2_{\theta'}| = 16\left| \mathbb{E}[H^{(\theta)}_{1234}H^{(\theta)}_{1567} - H^{(\theta')}_{1234}H^{(\theta')}_{1567}] - \mathbb{E}[H^{(\theta)}_{1234}H^{(\theta)}_{5678}] + \mathbb{E}[H^{(\theta')}_{1234}H^{(\theta')}_{5678}]] \right|$$

$$\le 16\,\mathbb{E}\left| H^{(\theta)}_{1234}H^{(\theta)}_{1567} - H^{(\theta')}_{1234}H^{(\theta')}_{1567} \right| + 16\,\mathbb{E}\left| H^{(\theta)}_{1234}H^{(\theta)}_{5678} - H^{(\theta')}_{1234}H^{(\theta')}_{5678} \right|.$$

Under Assumptions (A') and (C') we know that $|H_{1234}| \le 4\nu_k\nu_l$ and $|H^{(\theta)}_{1234} - H^{(\theta')}_{1234}| \le 4\nu_k L_l \rho_\Gamma + 4\nu_l L_k \rho_\Omega$, and so

$$|H^{(\theta)}_{1234}H^{(\theta)}_{1567} - H^{(\theta')}_{1234}H^{(\theta')}_{1567}| \le |H^{(\theta)}_{1234}H^{(\theta)}_{1567} - H^{(\theta)}_{1234}H^{(\theta')}_{1567}| + |H^{(\theta)}_{1234}H^{(\theta')}_{1567} - H^{(\theta')}_{1234}H^{(\theta')}_{1567}|$$

$$= |H^{(\theta)}_{1234}||H^{(\theta)}_{1567} - H^{(\theta')}_{1567}| + |H^{(\theta')}_{1567}||H^{(\theta)}_{1234} - H^{(\theta')}_{1234}|$$

$$\le 32\nu_k\nu_l(\nu_k L_l \rho_\Gamma + \nu_l L_k \rho_\Omega).$$

This expression is true for both components of $|\sigma^2_\theta - \sigma^2_{\theta'}|$ and so it follows that

$$|\sigma^2_\theta - \sigma^2_{\theta'}| \le 1024\nu_k\nu_l(\nu_k L_l \rho_\Gamma + \nu_l L_k \rho_\Omega). \tag{16}$$

Similarly, replacing the expectations $\mathbb{E}[H_{1234}H_{1567}]$ and $\mathbb{E}[H_{1234}H_{5678}]$ with the respective estimators $\frac{1}{(n)_4(n-1)_3} \sum_{(ijqr),(bcd)\setminus i} H_{ijqr}H_{ibcd}$ and $\frac{1}{(n)_4^2} \sum_{(ijqr),(abcd)} H_{ijqr}H_{abcd}$ give us the same bound

$$|\hat{\sigma}^2_\theta - \hat{\sigma}^2_{\theta'}| \le 1024\nu_k\nu_l(\nu_k L_l \rho_\Gamma + \nu_l L_k \rho_\Omega). \tag{17}$$

Next, using Lemma D.1 and Lemma D.2 followed by a union bound over the $T_\Omega T_\Gamma$ center combinations gives us, with probability at least $1 - \delta$,

$$\max_{\substack{\omega' \in \{\omega_1, \ldots, \omega_{T_\Omega}\} \\ \gamma' \in \{\gamma_1, \ldots, \gamma_{T_\Gamma}\}}} |\hat{\sigma}_{\theta'}^2 - \sigma_{\theta'}^2| \leq 6144 \nu_k^2 \nu_l^2 \sqrt{\frac{2}{n} \log \frac{2T_\Omega T_\Gamma}{\delta}} + \frac{4608 \nu_k^2 \nu_l^2}{n} \tag{18}$$

$$\leq \frac{2048 \nu_k^2 \nu_l^2}{\sqrt{n}} \left( 3\sqrt{2 \log \frac{2}{\delta} + 2D_\Omega \log \frac{4R_\Omega}{\rho_\Omega} + 2D_\Gamma \log \frac{4R_\Gamma}{\rho_\Gamma}} + \frac{9}{4\sqrt{n}} \right).$$

Finally, we combine Equations (16) to (18) to get

$$\sup_{\theta \in \Theta} |\hat{\sigma}_\theta^2 - \sigma_\theta^2| \leq \frac{2048 \nu_k^2 \nu_l^2}{\sqrt{n}} \left( 3\sqrt{2 \log \frac{2}{\delta} + 2D_\Omega \log \frac{4R_\Omega}{\rho_\Omega} + 2D_\Gamma \log \frac{4R_\Gamma}{\rho_\Gamma}} + \frac{9}{4\sqrt{n}} + \sqrt{n} \left( \frac{L_k}{\nu_k} \rho_\Omega + \frac{L_l}{\nu_l} \rho_\Gamma \right) \right).$$

Setting $\rho_\Omega = \rho_\Gamma = 1/\sqrt{n}$ gives us our desired uniform convergence bound.

$\square$

## D.5 Main Results

**Theorem D.5** (Uniform convergence of $\hat{J}_{\text{HSIC}}^\lambda$). *Under Assumptions (A') to (C'), let $\Theta \subseteq \Omega \times \Gamma$ be the set of kernel parameters $\theta \in \Theta$ for which $\sigma_\theta^2 \geq s^2$, and take $\lambda = n^{-1/3}$. Assume $\nu_k, \nu_l \geq 1$. Then, with probability at least $1 - \delta$,*

$$\sup_{\theta \in \Theta} \left| \frac{\hat{\eta}_\theta}{\hat{\sigma}_{\theta,\lambda}} - \frac{\eta_\theta}{\sigma_\theta} \right| \leq \frac{2\nu_k \nu_l}{s^2 n^{1/3}} \left[ \frac{1}{s} + \frac{9216 \nu_k^2 \nu_l^2}{\sqrt{n}} \right.$$

$$\left. + \left( 12288 \nu_k^2 \nu_l^2 + \frac{8s}{n^{1/6}} \right) \left( \frac{L_k}{\nu_k} + \frac{L_l}{\nu_l} + \sqrt{2 \log \frac{4}{\delta} + 2D_\Omega \log(4R_\Omega \sqrt{n}) + 2D_\Gamma \log(4R_\Gamma \sqrt{n})} \right) \right],$$

*and thus, treating $\nu_k, \nu_l$ as constants,*

$$\sup_{\theta \in \Theta} \left| \frac{\hat{\eta}_\theta}{\hat{\sigma}_{\theta,\lambda}} - \frac{\eta_\theta}{\sigma_\theta} \right| = \tilde{\mathcal{O}}_P \left( \frac{1}{s^2 n^{1/3}} \left[ \frac{1}{s} + L_k + L_l + \sqrt{D_\Omega} + \sqrt{D_\Gamma} \right] \right).$$

*Proof.* Let $\hat{\sigma}_{\theta,\lambda}^2 := \hat{\sigma}_\theta^2 + \lambda$ be our regularized variance estimator from which we can assume is positive. We start by decomposing

$$\sup_{\theta \in \Theta} \left| \frac{\hat{\eta}_\theta}{\hat{\sigma}_{\theta,\lambda}} - \frac{\eta_\theta}{\sigma_\theta} \right| \leq \sup_{\theta \in \Theta} \left| \frac{\hat{\eta}_\theta}{\hat{\sigma}_{\theta,\lambda}} - \frac{\hat{\eta}_\theta}{\sigma_{\theta,\lambda}} \right| + \sup_{\theta \in \Theta} \left| \frac{\hat{\eta}_\theta}{\sigma_{\theta,\lambda}} - \frac{\hat{\eta}_\theta}{\sigma_\theta} \right| + \sup_{\theta \in \Theta} \left| \frac{\hat{\eta}_\theta}{\sigma_\theta} - \frac{\eta_\theta}{\sigma_\theta} \right|$$

$$= \sup_\theta \frac{|\hat{\eta}_\theta|}{\hat{\sigma}_{\theta,\lambda} \cdot \sigma_{\theta,\lambda}} \frac{|\hat{\sigma}_{\theta,\lambda}^2 - \sigma_{\theta,\lambda}^2|}{\hat{\sigma}_{\theta,\lambda} + \sigma_{\theta,\lambda}} + \sup_\theta \frac{|\hat{\eta}_\theta|}{\sigma_{\theta,\lambda} \cdot \sigma_\theta} \frac{|\sigma_{\theta,\lambda}^2 - \sigma_\theta^2|}{\sigma_{\theta,\lambda} + \sigma_\theta} + \sup_\theta \frac{1}{\sigma_\theta} |\hat{\eta}_\theta - \eta_\theta|$$

$$\leq \frac{4\nu_k \nu_l}{s\sqrt{\lambda}(s + \sqrt{\lambda})} \sup_\theta |\hat{\sigma}_\theta^2 - \sigma_\theta^2| + \frac{4\nu_k \nu_l \lambda}{s\sqrt{s^2 + \lambda}(s + \sqrt{s^2 + \lambda})} + \frac{1}{s} \sup_\theta |\hat{\eta}_\theta - \eta_\theta|$$

$$\leq \frac{4\nu_k \nu_l}{s^2 \sqrt{\lambda}} \sup_\theta |\hat{\sigma}_\theta^2 - \sigma_\theta^2| + \frac{1}{s} \sup_\theta |\hat{\eta}_\theta - \eta_\theta| + \frac{2\nu_k \nu_l \lambda}{s^3}.$$

Proposition D.3 and Proposition D.4 show the uniform convergence of $\hat{\eta}_\theta$ and $\hat{\sigma}_\theta$, from which we get that with probability at least $1 - \delta$, the error is at most

$$\sup_{\theta \in \Theta} \left| \frac{\hat{\eta}_\theta}{\hat{\sigma}_{\theta,\lambda}} - \frac{\eta_\theta}{\sigma_\theta} \right| \leq \frac{2\nu_k \nu_l \lambda}{s^3} + \frac{18432 \nu_k^3 \nu_l^3}{s^2 n \sqrt{\lambda}} + \left[ \frac{8192 \nu_k^3 \nu_l^3}{s^2 \sqrt{\lambda n}} + \frac{8\nu_k \nu_l}{s\sqrt{n}} \right] \left( \frac{L_k}{\nu_k} + \frac{L_l}{\nu_l} \right)$$

$$+ \left[ \frac{24576 \nu_k^3 \nu_l^3}{s^2 \sqrt{\lambda n}} + \frac{16 \nu_k \nu_l}{s\sqrt{n}} \right] \sqrt{2 \log \frac{4}{\delta} + 2D_\Omega \log(4R_\Omega \sqrt{n}) + 2D_\Gamma \log(4R_\Gamma \sqrt{n})}.$$

Taking $\lambda = n^{-1/3}$ gives

$$\sup_{\theta \in \Theta} \left| \frac{\hat{\eta}_\theta}{\hat{\sigma}_{\theta,\lambda}} - \frac{\eta_\theta}{\sigma_\theta} \right| \leq \frac{2\nu_k \nu_l}{s^3 n^{1/3}} + \frac{18432 \nu_k^3 \nu_l^3}{s^2 n^{5/6}} + \left[ \frac{8192 \nu_k^3 \nu_l^3}{s^2 n^{1/3}} + \frac{8\nu_k \nu_l}{s\sqrt{n}} \right] \left( \frac{L_k}{\nu_k} + \frac{L_l}{\nu_l} \right)$$
$$+ \left[ \frac{24576 \nu_k^3 \nu_l^3}{s^2 n^{1/3}} + \frac{16 \nu_k \nu_l}{s\sqrt{n}} \right] \sqrt{2 \log \frac{4}{\delta} + 2D_\Omega \log(4R_\Omega \sqrt{n}) + 2D_\Gamma \log(4R_\Gamma \sqrt{n})}.$$

Using $\nu_k, \nu_l \geq 1$ we can slightly simplify our bound to

$$\sup_{\theta \in \Theta} \left| \frac{\hat{\eta}_\theta}{\hat{\sigma}_{\theta,\lambda}} - \frac{\eta_\theta}{\sigma_\theta} \right| \leq \left[ \frac{24576 \nu_k^3 \nu_l^3}{s^2 n^{1/3}} + \frac{16 \nu_k \nu_l}{s\sqrt{n}} \right] \left( \frac{L_k}{\nu_k} + \frac{L_l}{\nu_l} + \sqrt{2 \log \frac{4}{\delta} + 2D_\Omega \log(4R_\Omega \sqrt{n}) + 2D_\Gamma \log(4R_\Gamma \sqrt{n})} \right)$$
$$+ \frac{2\nu_k \nu_l}{s^3 n^{1/3}} + \frac{18432 \nu_k^3 \nu_l^3}{s^2 n^{5/6}}.$$

$\square$

# E    Experimental Details

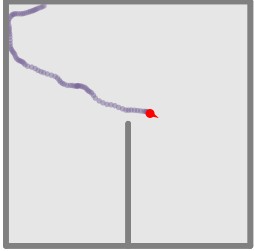

Figure 7: RatInABox simulation environment. The red dot is the current position of the rat and the purple circles indicate the past trajectory over 5 seconds. The box is designed to have only a single protruding wall.

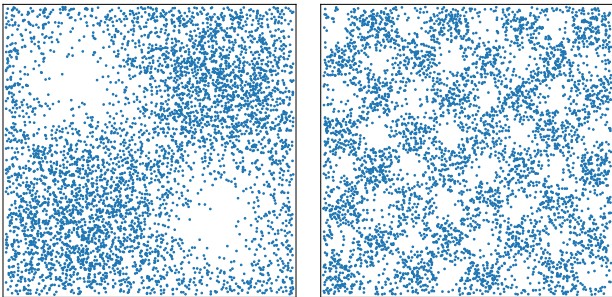

Figure 8: Samples drawn from the sinusoidal problem with frequency $\ell = 1$ (left) and $\ell = 4$ (right). We consider the latter frequency in our experiments.

## E.1    Training & Test Details

We design the featurizers $\phi_\omega$ and $\phi_\kappa$ of our deep kernels $k_\omega$ and $l_\kappa$ to be neural networks with ReLU activations. We avoid using normalization as it may affect test power. Moreover, we make the Gaussian bandwidth of both $k_\omega$ and $l_\kappa$ a learnable parameter, as well as the smoothing rate $\epsilon$. To make comparisons as fair as possible, we use similar neural network architectures for each deep learning based method. In general, we let the featurizer of HSIC-D and MMD-D be identical up to a concatenation layer which concatenates $X$ and $Y$ to frame the problem as a two-sample test. We construct the C2ST-S/L classifier as the MMD-D featurizer plus a linear layer classification head with scalar output, and we let C2ST-S/L, InfoNCE, NWJ, and NDS all use identical architectures for the classifier and critic. Detailed descriptions of each architecture are demonstrated in the following subsections.

All optimization-based methods (HSIC-D/Dx/O, NDS/InfoNCE/NWJ, MMD-D, C2ST-S/L) are first trained on an identical split of the data, and then tested on the remaining split. In contrast, HSIC-M selects the median bandwidth based on the entire dataset, and is evaluated on the test set. We train our models using the AdamW optimizer with a learning rate of 1e-4 over 1,000 epochs for HDGM and RatInABox, and 10,000 epochs for Sinusoid and Wine. We use a batch size of 512. All methods are implemented in PyTorch and trained on a NVIDIA A100SXM4 GPU.

Regarding the power vs. test size experiments, we use a training size of 10,000 for HDGM $\leq$ 30, 100,000 for HDGM > 30, and 2000 validation samples for all dimensions. For the Sinusoid problem, we train on 5,000 samples and use 1,000 for validation. RatInABox uses a training size of 4,000 samples without validation, and Wine uses 1,200 training samples without validation.

Once learned, each methods' empirical power is evaluated on 100 test sets $(S_Z^{te_1}, ..., S_Z^{te_{100}})$. Each test set contains $m$ test samples $S_Z^{te_i} = (Z_1^{te_i}, ..., Z_N^{te_i})$, which are then used to compute the average rejection rate under the null via a permutation test. We use 500 permutations for each test and with a predetermined type-I error rate of 0.05.

## E.2 Validation

Although validation sets were only used for early stopping in our experiments, it's certainly possible to use them for hyperparameter selection. The validity of permutation tests still hold as long as this validation set is separate from the test set. As for the validation criterion, a natural approach is to evaluate the same approximation of the asymptotic power used for our training objective. This avoids the need for relatively-expensive permutations, while providing a reasonable estimate of performance. Alternatively, we could also directly estimate the test statistic in cases where there isn't a simple expression for asymptotic power, though this approach may be less reliable since the statistic is not necessarily correlated with test power. It is also possible to select hyperparameters through cross-validation, where the training data is split into folds: one fold for evaluating hyperparameters and the remaining folds for training the model. Again, as long as this data is separate from the test set, any permutation test would be well-controlled. The drawback to these validation-based approaches is that effective power is reduced since we need to construct this validation set from either the training or test data.

Ideally, we would like to perform hyperparameter selection while avoiding sample splitting, but this is a challenging problem. One potential approach is to perform separate tests corresponding to each hyperparameter, and then combine these tests using a multiple test correction procedure like MMDAgg (Schrab et al., 2023) to ensure valid level control on the test data. However, this is potentially expensive if hyperparameters need to be optimized. It's also unclear if this would perform better than simply using a validation set due to the conservatism of the multiple test correction procedure. A related approach is MMDFuse (Biggs et al., 2023), which constructs an aggregate statistic by combining MMD estimates at varying bandwidths. This approach could be naturally extended to consider estimates based on different training sets and would naturally fit into our optimization framework; though doing so in a way that does not explode compute nor reduce critic quality is another challenge.

## E.3 Architectures

In all experiments we consider deep kernels with Gaussian feature and smoothing kernels $\kappa$ and $q$, where each bandwidth is a trainable parameter randomly initialized around 1.0. We let the smoothing weight $\epsilon$ also be a learnable parameter initialized to 0.01. No batch normalization is used and all hidden layers use ReLU activations. Dataset-specific designs are elaborated below.

**High-dimensional Gaussian mixture**. We use a feed-forward network for our deep kernel featurizer with latent dimensions $2d, 3d,$ and $2d$. Details of each model is given in Table 1.

**Sinusoid**. The deep kernel featurizer is taken to be a feed-forward network with widths 1x8x12x8. C2st, infoNCE, and NWJ use a similar architecture –one with widths 2x8x12x8x1– which includes an additional scalar output layer.

**RatInABox**. We use a feed-forward featurizer with details given in Table 2. Unlike the previous two problems, the sample spaces $\mathcal{X}$ and $\mathcal{Y}$ are not equivalent, and so the deep featurizers for $k$ and $l$ have different architectures.

| dataset | model | input | featurizer |
|---------|-------|-------|------------|
| HDGM-4 | HSIC-D | X or Y | $[\ 2 \to 4 \to 6 \to 4\ ]$ |
|  | MMD-D | [X, Y] | $[\ 4 \to 8 \to 12 \to 8\ ]$ |
|  | C2ST-S/L | [X, Y] | $[\ 4 \to 8 \to 12 \to 8 \to 1\ ]$ |
| HDGM-8 | HSIC-D | X or Y | $[\ 4 \to 8 \to 12 \to 8\ ]$ |
|  | MMD-D | [X, Y] | $[\ 8 \to 16 \to 24 \to 16\ ]$ |
|  | C2ST-S/L | [X, Y] | $[\ 8 \to 16 \to 24 \to 16 \to 1\ ]$ |
| HDGM-10 | HSIC-D | X or Y | $[\ 5 \to 10 \to 15 \to 10\ ]$ |
|  | MMD-D | [X, Y] | $[\ 10 \to 20 \to 30 \to 20\ ]$ |
|  | C2ST-S/L | [X, Y] | $[\ 10 \to 20 \to 30 \to 20 \to 1\ ]$ |
| HDGM-20 | HSIC-D | X or Y | $[\ 10 \to 20 \to 30 \to 20\ ]$ |
|  | MMD-D | [X, Y] | $[\ 20 \to 40 \to 60 \to 40\ ]$ |
|  | C2ST-S/L | [X, Y] | $[\ 20 \to 40 \to 60 \to 40 \to 1\ ]$ |

| dataset | model | input | featurizer |
|---------|-------|-------|------------|
| HDGM-30 | HSIC-D | X or Y | $[\ 15 \to 30 \to 45 \to 30\ ]$ |
|  | MMD-D | [X, Y] | $[\ 30 \to 60 \to 90 \to 60\ ]$ |
|  | C2ST-S/L | [X, Y] | $[\ 30 \to 60 \to 90 \to 60 \to 1\ ]$ |
| HDGM-40 | HSIC-D | X or Y | $[\ 20 \to 40 \to 60 \to 40\ ]$ |
|  | MMD-D | [X, Y] | $[\ 40 \to 80 \to 120 \to 80\ ]$ |
|  | C2ST-S/L | [X, Y] | $[\ 40 \to 80 \to 120 \to 80 \to 1\ ]$ |
| HDGM-50 | HSIC-D | X or Y | $[\ 25 \to 50 \to 75 \to 50\ ]$ |
|  | MMD-D | [X, Y] | $[\ 50 \to 100 \to 150 \to 100\ ]$ |
|  | C2ST-S/L | [X, Y] | $[\ 50 \to 100 \to 150 \to 100 \to 1\ ]$ |

Table 1: Featurizer architectures used in deep kernels for HSIC-D, MMD-D, and classifier architecture used for C2ST-S/L on the HDGM problem. Brackets denote a sequence of linear layers with corresponding input and output features.

| method | input | network |
|--------|-------|---------|
| HSIC-D | X | $[\ 8 \to 32 \to 64 \to 32\ ]$ |
|  | Y | $[\ 2 \to 4 \to 8 \to 4\ ]$ |
| MMD-D | [X, Y] | $[\ 10 \to 32 \to 64 \to 32\ ]$ |
| C2ST-S/L | [X, Y] | $[\ 10 \to 32 \to 64 \to 32 \to 1\ ]$ |

Table 2: Featurizer architectures used in deep kernels for HSIC-D, MMD-D, and classifier architecture used for C2ST-S/L on the RatInABox problem. Brackets denote a sequence of linear layers with corresponding input and output features.

# F  Additional Experiments

## F.1  High-Dimensional Gaussian Mixture

We provide comprehensive power versus test size results for the HDGM problem at all dimensions $d = \{2, 4, 5, 10, 15, 20\}$ in Figure 9. Overall, our method HSIC-D/Dx achieves highest power at the smallest test sizes.

Additionally, we demonstrate the effectiveness of our method at various dimensions $d/2$ by examining the empirical test power at HDGM-$d$ for $d \in \{4, 8, 10, 20, 30, 40, 50\}$ with fixed test sizes $m$. Results are shown in Figure 10. Again, HSIC-D exhibits the highest test power across all dimensions. When using a small number of test samples (e.g. $m = 100$), the performance of HSIC-D slightly degrades with increasing dimension, whereas at larger test sample sizes it consistently has near-perfect power.

## F.2  Type-I Error

Table 3 shows that the type-I error rates for our optimization-based tests are well-controlled.

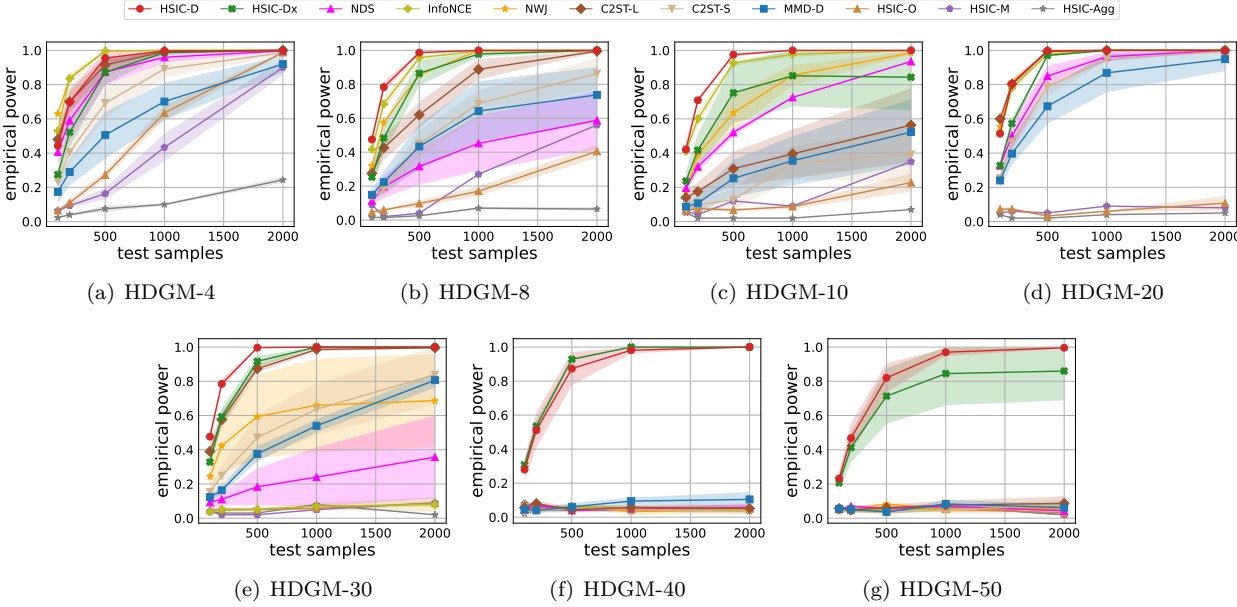

Figure 9: Power vs test size $m$ for the HDGM problem at dimensions $d = \{2, 4, 5, 10, 15, 20\}$. The average test power is computed over 5 training runs, where the empirical power is determined over 100 permutation tests. The shaded region covers one standard error from the mean

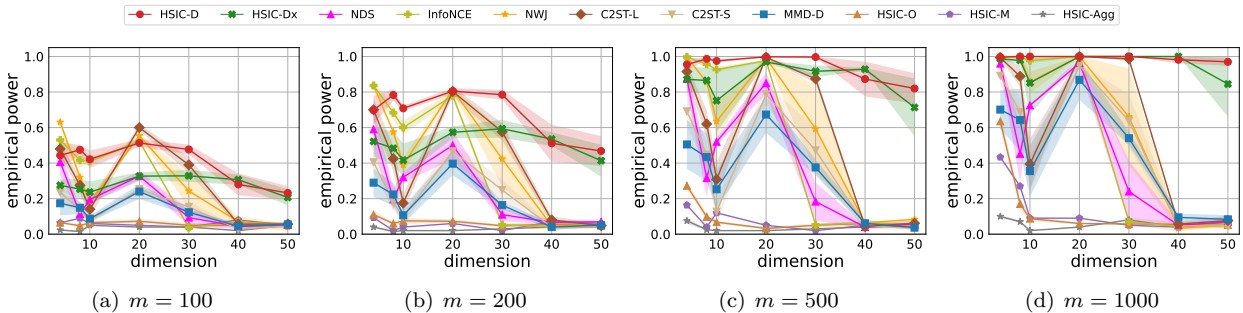

Figure 10: Empirical power vs dimension across various test sample sizes $m = \{100, 200, 500, 1000\}$ for HDGM. The shaded region covers one standard error over 5 training runs.

### F.3 Maximizing HSIC vs. its SNR

We examine the trade-off between directly optimizing HSIC versus its SNR objective $J$. The results on power versus test size are shown in Figure 11. Optimizing our proposed objective $J$ significantly outperforms optimizing HSIC for all problems.

### F.4 Independence Testing with MMD

Let $\mathbb{Z} = \{(X_1, Y_1), ..., (X_m, Y_m)\}$ be a test set and $\mathcal{S}_m$ be the permutation group of $[m]$ with elements $\sigma \in \mathcal{S}_m$. Suppose we take $p$ permutations $\boldsymbol{\sigma} = \{\sigma_1, ..., \sigma_p\}$, with each permutation sampled uniformly from $\mathcal{G}_m$. We define the action of $\boldsymbol{\sigma}$ on test samples $\mathbb{Z}$ as $\boldsymbol{\sigma}\mathbb{Z} = \{(X_i, Y_{\sigma_\ell(i)})\}_{i \in [m], \ell \in [p]}$, and the action of $\boldsymbol{\sigma}$ on the empirical distribution $\hat{\mathbb{P}}_{XY}$ as the empirical distribution of the permuted samples, i.e. $\boldsymbol{\sigma}\hat{\mathbb{P}}_{XY} = \frac{1}{np} \sum_{i=1}^{n} \sum_{\ell=1}^{p} \delta_{X_i \times Y_{\sigma_\ell(i)}}$.

| Method | HDGM-4 | HDGM-8 | HDGM-10 | HDGM-20 | HDGM-30 | HDGM-40 | HDGM-50 | Sinusoid | RatInABox |
|--------|--------|--------|---------|---------|---------|---------|---------|----------|-----------|
| HSIC-D | 0.043 | 0.043 | 0.050 | 0.050 | 0.062 | 0.057 | 0.052 | 0.050 | 0.048 |
| MMD-D | 0.048 | 0.055 | 0.040 | 0.053 | 0.048 | 0.048 | 0.055 | 0.054 | 0.050 |
| C2ST-L | 0.060 | 0.030 | 0.053 | 0.048 | 0.053 | 0.058 | 0.045 | 0.046 | 0.048 |
| InfoNCE | 0.046 | 0.046 | 0.046 | 0.054 | 0.044 | 0.050 | 0.048 | 0.048 | 0.045 |
| NWJ | 0.050 | 0.054 | 0.058 | 0.052 | 0.044 | 0.064 | 0.054 | 0.052 | 0.042 |

Table 3: Average type-I error rates under the null distribution over 400 tests. We use $m = 512$ samples.

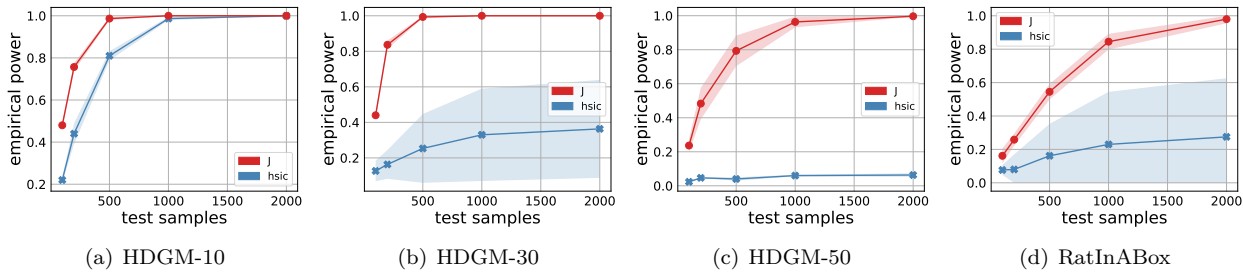

(a) HDGM-10      (b) HDGM-30      (c) HDGM-50      (d) RatInABox

Figure 11: Test power using deep kernels optimized for the approximate asymptotic test power $J$ (red) versus optimizing just the test statistic HSIC (blue).

One way we can construct an independence criterion is by taking the MMD between $\mathbb{P}_{XY}$ and $\mathbb{P}_X \times \mathbb{P}_Y$. We do this for MMD-D by using the empirical distributions $\hat{\mathbb{P}}_{XY}$ and $\boldsymbol{\sigma}\hat{\mathbb{P}}_{XY}$ respectively, which yields

$$\mathrm{MMD}_{k \times l}^2(\hat{\mathbb{P}}_{XY}, \boldsymbol{\sigma}\hat{\mathbb{P}}_{XY}) = \frac{1}{m^2} \sum_{i,j \in [m]} k_{i,j} l_{i,j} - \frac{2}{m^2 p} \sum_{\substack{i,j \in [m] \\ q \in [p]}} k_{i,j} l_{i,\sigma_q(j)} + \frac{1}{m^2 p^2} \sum_{\substack{i,j \in [m] \\ q,r \in [p]}} k_{i,j} l_{\sigma_q(i),\sigma_r(j)}.$$

We first show that even with one permutation, this yields a consistent estimator.

**Proposition 3.2.** *Suppose $h$ satisfies $\sup_{x \in \mathcal{X}, y \in \mathcal{Y}} h((x,y),(x,y)) \leq \nu^2$. The estimator $\widehat{\mathrm{MMD}}_b^2(\mathbb{Z}, \sigma\mathbb{Z})$, where $\mathbb{Z}$ is independent of a uniformly random $\sigma$, satisfies with probability at least $1 - \delta$ that*

$$|\widehat{\mathrm{MMD}}_b^2(\mathbb{Z}, \sigma\mathbb{Z}) - \mathrm{MMD}^2(\mathbb{P}_{xy}, \mathbb{P}_x \times \mathbb{P}_y)| \leq \frac{17\nu^2}{\sqrt{m}} \left[ \sqrt{2 \log \frac{8}{\delta}} + \frac{1}{\sqrt{m}} \right].$$

*Proof.* First notice that

$$\mathrm{MMD}^2(\mathbb{P}_{xy}, \mathbb{P}_x \times P_y) = \underbrace{\mathbb{E}\, h((X,Y),(X',Y'))}_{\mu_1} + \underbrace{\mathbb{E}\, h((X,Y'),(X'',Y'''))}_{\mu_2} - 2\underbrace{\mathbb{E}\, h((X,Y),(X',Y''))}_{\mu_3},$$

while the estimator $\widehat{\mathrm{MMD}}_b^2(\mathbb{Z}, \sigma\mathbb{Z})$ is given by

$$\underbrace{\frac{1}{m^2} \sum_{i=1}^m \sum_{j=1}^m h((x_i,y_i),(x_j,y_j))}_{T_1} + \underbrace{\frac{1}{m^2} \sum_{i=1}^m \sum_{j=1}^m h((x_i,y_{\sigma_i}),(x_j,y_{\sigma_j}))}_{T_2} - 2\underbrace{\frac{1}{m^2} \sum_{i=1}^m \sum_{j=1}^m h((x_i,y_i),(x_j,y_{\sigma_j}))}_{T_3}.$$

The first term $T_1$ is a typical $V$-statistic:

$$\frac{1}{m^2} \sum_{i \neq j} h((x_i,y_i),(x_j,y_j)) + \frac{1}{m^2} \sum_{i=1}^m h((x_i,y_i),(x_i,y_i)) = \frac{m-1}{m} U_1 + \frac{1}{m} R_1,$$

where we have defined a $U$-statistic $U_1 = \frac{1}{m(m-1)} \sum_{i \neq j} h((x_i, y_i), (x_j, y_j))$ and a remainder term $R_1 = \frac{1}{m} \sum_i h((x_i, y_i), (x_i, y_i))$. For $U_1$, we have immediately that $\mathbb{E} U_1 = \mathbb{E} h((X, Y), (X', Y')) =: \mu_1$. Noting that both $R_1$ and $\mu_1$ are necessarily in $[0, \nu^2]$, the overall error from this term is therefore

$$|T_1 - \mu_1| = \left| \left(1 - \frac{1}{m}\right)(U_1 - \mu_1) + \frac{1}{m}(R_1 - \mu_1) \right| \leq \left(1 - \frac{1}{m}\right)|U_1 - \mu_1| + \frac{1}{m}\nu^2.$$

Changing a single $(x_i, y_i)$ pair changes $U_1$ by at most $\frac{1}{m(m-1)} \cdot 2(m-1) \cdot \nu^2 = \frac{2\nu^2}{m}$, and so McDiarmid's inequality gives that with probability at least $1 - \delta_1$,

$$|U_1 - \mu_1| \leq \frac{2\nu^2}{m} \sqrt{\frac{m}{2} \log \frac{2}{\delta_1}} = \nu^2 \sqrt{\frac{2}{m} \log \frac{2}{\delta_1}},$$

and so with probability at least $1 - \delta_1$,

$$|T_1 - \mu_1| \leq \left(1 - \frac{1}{m}\right)\nu^2 \sqrt{\frac{2}{m} \log \frac{2}{\delta_1}} + \frac{1}{m}\nu^2 \leq \frac{\nu^2}{\sqrt{m}}\left[\sqrt{2 \log \frac{2}{\delta_1}} + \frac{1}{\sqrt{m}}\right]. \tag{19}$$

Turning to $T_3$ next, we can use a similar approach if we also take into account the random $\sigma$. We can write $T_3$ as

$$\frac{1}{m^2} \sum_{i,j:|\{i,j,\sigma_j\}|=3} h((x_i, y_i), (x_j, y_{\sigma_j})) + \frac{1}{m^2} \sum_{i,j:|\{i,j,\sigma_j\}|<3} h((x_i, y_i), (x_j, y_{\sigma_j})) = \frac{N_3^{(\sigma)}}{m^2} U_3 + \left(1 - \frac{N_3^{(\sigma)}}{m^2}\right) R_3,$$

where we let $N_3^{(\sigma)}$ be the (random) number of $(i, j)$ pairs for which $i, j, \sigma_j$ are all distinct, $U_3$ the mean of $h((x_i, y_i), (X_j, y_{\sigma_j}))$ for which these indices are distinct, and $R_3$ the mean for which they are not. Note that $|\mu_3|, |R_3| \leq \nu^2$, regardless of the choice of $\sigma$ and $\mathbb{Z}$, and so

$$|T_3 - \mu_3| = \left| \frac{N_3^{(\sigma)}}{m^2}(U_3 - \mu_3) + \left(1 - \frac{N_3^{(\sigma)}}{m^2}\right)(R_3 - \mu_3) \right| \leq \frac{N_3^{(\sigma)}}{m^2}|U_3 - \mu_3| + \left(1 - \frac{N_3^{(\sigma)}}{m^2}\right)2\nu^2.$$

Fix the choice of $\sigma$, but let $\mathbb{Z}$ be random. Then $\mathbb{E} U_3 = \mu_3$, and changing a single $(x_i, y_i)$ pair changes the value of $U_3$ by at most $\frac{1}{N_3^{(\sigma)}} \cdot 3(m-1) \cdot \nu^2$. Thus, applying McDiarmid's inequality conditionally on the choice of $\sigma$, with probability at least $\delta_3'$

$$|U_3 - \mu_3| \leq \frac{3\nu^2}{\sqrt{2}} \frac{\sqrt{m}(m-1)}{N_3^{(\sigma)}} \sqrt{\log \frac{2}{\delta_3'}},$$

obtaining that

$$|T_3 - \mu_3| \leq \frac{3\nu^2}{\sqrt{2}} \frac{m-1}{m\sqrt{m}} \sqrt{\log \frac{2}{\delta_3'}} + \left(1 - \frac{N_3^{(\sigma)}}{m^2}\right)2\nu^2.$$

We will also need to show that $N_3^{(\sigma)}/m^2$ is nearly 1. We have that

$$\mathbb{E} N_3^{(\sigma)} = \mathbb{E} \sum_{i=1}^m \sum_{j=1}^m \mathbb{1}(|\{i, j, \sigma_j\}| = 3) = \sum_{i \neq j} \Pr(\sigma_j \notin \{i, j\}) = m(m-1) \cdot \frac{m-2}{m} = (m-1)(m-2),$$

so that $\mathbb{E}\left(1 - N_3^{(\sigma)}/m^2\right) = \frac{3}{m} - \frac{2}{m^2}$. Moreover, if $\sigma$ and $\sigma'$ almost agree except that $\sigma_k' = \sigma_l$ and $\sigma_l' = \sigma_k$, then $|N_3^{(\sigma)} - N_3^{(\sigma')}| \leq 2m$: the only $(i, j)$ which are potentially affected are those of the form $(\cdot, k)$ or $(\cdot, l)$. Using a version of McDiarmid's inequality for uniform distributions of permutations (Lemma F.1) then gives us that with probability at least $1 - \delta_3'$ over the choice of $\sigma$,

$$N_3^{(\sigma)} \geq (m-1)(m-2) - m\sqrt{8m \log \frac{1}{\delta_3''}}$$

and so

$$1 - \frac{N_3^{(\sigma)}}{m^2} \leq \frac{3}{m} - \frac{2}{m^2} + \sqrt{\frac{8}{m} \log \frac{1}{\delta_3''}};$$

thus with probability at least $\delta_3' + \delta_3''$, we have that

$$\begin{aligned}
|T_3 - \mu_3| &\leq \frac{\nu^2}{\sqrt{m}} \left[ \frac{3}{\sqrt{2}} \frac{m-1}{m} \sqrt{\log \frac{2}{\delta_3'}} + \frac{6}{\sqrt{m}} - \frac{4}{m\sqrt{m}} + 4\sqrt{2 \log \frac{1}{\delta_3''}} \right] \\
&\leq \frac{\nu^2}{\sqrt{m}} \left[ \frac{3}{2} \sqrt{2 \log \frac{2}{\delta_3'}} + 4\sqrt{2 \log \frac{1}{\delta_3''}} + \frac{6}{\sqrt{m}} \right].
\end{aligned}$$

To simplify this a little further, let $\delta_3' = \frac{2}{3}\delta_3$ and $\delta_3'' = \frac{1}{3}\delta_3$; then it holds with probability at least $1 - \delta_3$ that

$$|T_3 - \mu_3| \leq \frac{6\nu^2}{\sqrt{m}} \left[ \sqrt{2 \log \frac{3}{\delta_3}} + \frac{1}{\sqrt{m}} \right]. \tag{20}$$

It remains to handle $T_2$, which is similar to $T_3$. Letting $N_2^{(\sigma)}$ be the number of $(i,j)$ pairs for which $i, j, \sigma_i, \sigma_j$ are all distinct, we can similarly define $U_2$ with mean $\mu_2$ and $R_2$ with $|R_2| \leq \nu^2$ so that

$$|T_2 - \mu_2| = \left| \frac{N_2^{(\sigma)}}{m^2}(U_2 - \mu_2) + \left(1 - \frac{N_2^{(\sigma)}}{m^2}\right)(R_2 - \mu_2) \right| \leq \frac{N_2^{(\sigma)}}{m^2}|U_2 - \mu_2| + \left(1 - \frac{N_3^{(\sigma)}}{m^2}\right) 2\nu^2.$$

For a fixed $\sigma$, changing a single $(x_i, y_i)$ pair changes the value of $U_2$ by at most $\frac{1}{N_2^{(\sigma)}} \cdot 4(m-1) \cdot \nu^2$, and we obtain like before that with probability at least $1 - \delta_2'$,

$$|T_2 - \mu_2| \leq \frac{4\nu^2}{\sqrt{2}} \frac{m-1}{m\sqrt{m}} \sqrt{\log \frac{2}{\delta_2'}} + \left(1 - \frac{N_2^{(\sigma)}}{m^2}\right) 2\nu^2.$$

Since when $i \neq j$ any permutation satisfies that $\sigma_i \neq \sigma_j$, we have that

$$\mathbb{E}\, N_2^{(\sigma)} = \sum_{i \neq j} \Pr(\sigma_i, \sigma_j \notin \{i, j\}) = \sum_{i \neq j} \frac{m-2}{m} \cdot \frac{m-3}{m-1} = (m-2)(m-3).$$

A single transposition changes changes $N_2^{(\sigma)}$ by no more than $4m$, and so by Lemma F.1 we obtain that with probability at least $1 - \delta_2''$,

$$1 - \frac{N_2^{(\sigma}}{m^2} \leq \frac{5}{m} - \frac{6}{m^2} + \frac{4}{\sqrt{m}} \sqrt{2 \log \frac{1}{\delta_2''}}.$$

Letting $\delta_2' = \frac{2}{3}\delta_2$ and $\delta_2'' = \frac{1}{3}\delta_2$, it thus holds with probability at least $1 - \delta_2$ that

$$\begin{aligned}
|T_2 - \mu_2| &\leq \frac{2\nu^2}{\sqrt{m}} \left[ \frac{m-1}{m} \sqrt{2 \log \frac{3}{\delta_2}} + \frac{5}{\sqrt{m}} - \frac{6}{m\sqrt{m}} + 4\sqrt{2 \log \frac{3}{\delta_2}} \right] \\
&\leq \frac{10\nu^2}{\sqrt{m}} \left[ \sqrt{2 \log \frac{3}{\delta_2}} + \frac{1}{\sqrt{m}} \right]. \tag{21}
\end{aligned}$$

To combine (19), (20), and (21), it will be convenient to use $\delta_1 = \frac{1}{4}\delta$ and $\delta_2 = \delta_3 = \frac{3}{8}\delta$, since then $\delta_1/2 = \delta/8$, $\delta_2/3 = \delta/8$ and $\delta_1 + \delta_2 + \delta_3 = \delta$. Then we obtain that with probability at least $1 - \delta$,

$$|T_1 + T_2 - 2T_3 - (\mu_1 + \mu_2 - 2\mu_3)| \leq \frac{17\nu^2}{\sqrt{m}} \left[ \sqrt{2 \log \frac{8}{\delta}} + \frac{1}{\sqrt{m}} \right]. \qquad \square$$

**Lemma F.1** (McDiarmid's inequality for uniform permutations). *Let $f : \mathcal{S}_m \to \mathbb{R}$, where $\mathcal{S}_m$ is the symmetric group of permutations on $[m]$, satisfy that for every $\sigma, \sigma' \in \mathcal{S}_m$, $|f(\sigma) - f(\sigma')| \le c\,|\{i : \sigma_i \ne \sigma'_i\}|$. Let $S$ be a random variable which is uniform on $\mathcal{S}_m$. Then it holds with probability at least $1 - \delta$ that $f(S) - \mathbb{E}_S\, f(S) \le c\sqrt{8m \log \frac{1}{\delta}}$.*

*Proof (based on Błasiok, 2025).* For $i \in \{0, \ldots, m\}$ and any permutation $\sigma$, define the Doob martingale

$$X_i = \mathbb{E}_S[f(S) \mid S_1 = \sigma_1, \ldots, S_i = \sigma_i],$$

so that $X_0 = \mathbb{E}_S\, f(S)$ and $X_m = f(\sigma)$.

We now show that $|X_i - X_{i+1}| \le 2c$. No matter the known values $(\sigma_1, \ldots, \sigma_i)$, for any $(\sigma_{i+1}, \ldots, \sigma_n)$, we can identify a uniquely paired $(\sigma'_{i+1}, \ldots, \sigma'_n)$ differing in at most two positions, which implies that $|f(\sigma) - f(\sigma')| \le 2c$. Because these pairs are equiprobable under the uniform distribution of $S$, this implies that any $\mathbb{E}[X_{i+1} \mid S_{i+1} = j]$ differs from any $\mathbb{E}[X_{i+1} \mid S_{i+1} = j']$ by at most $2c$, and so the same holds on average over $j$.

The Azuma-Hoeffding inequality then gives $\Pr(X_m - X_0 \ge \varepsilon) \le 2\exp\left(-\varepsilon^2/(2m(2c)^2)\right)$, and the desired result follows by solving for $\varepsilon$. $\square$

When considering more than one permutation, this looks reminiscent of the biased HSIC estimator. Indeed, when we consider the set of $n$ circular shifts $\boldsymbol{\sigma}_\circ$ (i.e. permutations where the order of the $Y$s are rotations of one another), then the two are equal with

$$\mathrm{MMD}^2_{k \times l}(\hat{\mathbb{P}}_{XY}, \boldsymbol{\sigma}_\circ \hat{\mathbb{P}}_{XY}) = \mathrm{MMD}^2_{k \times l}(\hat{\mathbb{P}}_{XY}, \hat{\mathbb{P}}_X \otimes \hat{\mathbb{P}}_Y) = \mathrm{HSIC}_{k,l}(\hat{\mathbb{P}}_{XY}).$$

In practice, MMD-D underperforms against HSIC-D. We believe this might be because the variance of the permuted MMD estimator is often higher than that of HSIC. The following proposition proves this for the biased estimators.

**Proposition F.2.** *Suppose we have samples $\mathbb{Z} = \{(X_1, Y_1), ..., (X_m, Y_m)\}$ drawn iid from $\mathbb{P}_{XY}$, and let $\boldsymbol{\sigma} = \{\sigma_1, ..., \sigma_p\}$ be a set of $p$ permutations sampled uniformly from $\mathcal{S}_m$, the permutation group of $[m]$. We define the action of $\boldsymbol{\sigma}$ on samples $\mathbb{Z}$ as $\boldsymbol{\sigma}\mathbb{Z} = \{(X_i, Y_{\sigma_\ell(i)})\}_{i \in [m], \ell \in [p]}$. Then for the biased HSIC and MMD estimators we have*

$$\mathrm{Var}[\widehat{\mathrm{HSIC}}_{k,l}(\mathbb{Z})] \le \mathrm{Var}[\widehat{\mathrm{MMD}^2}_{k \times l}(\mathbb{Z}, \boldsymbol{\sigma}\mathbb{Z})].$$

*Proof.* First we note that

$$\widehat{\mathrm{MMD}^2}(\mathbb{Z}, \boldsymbol{\sigma}\mathbb{Z}) = \frac{1}{m^2} \sum_{i,j \in [m]} k_{i,j} l_{i,j} - \frac{2}{m^2 p} \sum_{\substack{i,j \in [m] \\ q \in [p]}} k_{i,j} l_{i,\sigma_q(j)} + \frac{1}{m^2 p^2} \sum_{\substack{i,j \in [m] \\ q,r \in [p]}} k_{i,j} l_{\sigma_q(i), \sigma_r(j)}.$$

Taking the expectation of this estimator conditioned on the samples $\mathbb{Z}$ gives us a Rao-Blackwellization. Now notice that for part of the second term

$$\mathbb{E}_{\boldsymbol{\sigma}}\left[\frac{1}{p} \sum_{q=1}^p k_{i,j} l_{i,\sigma_q(j)}\right] = k_{i,j}\left(\frac{1}{p}\sum_{q=1}^p \mathbb{E}_{\sigma_q}[l_{i,\sigma_q(j)}]\right) = \frac{1}{m}\sum_{t=1}^m k_{i,j} l_{i,t},$$

where we use the fact that $\mathbb{E}_{\sigma_q}[l_{i,\sigma_q(j)}] = \frac{1}{m}\sum_{t=1}^m l_{i,t}$ since the permutation of $j$ is equally likely to be any number in $[m]$. Similarly part of the last term yields

$$\mathbb{E}_{\boldsymbol{\sigma}}\left[\frac{1}{p^2}\sum_{q,r \in [p]} k_{i,j} l_{\sigma_q(i), \sigma_r(j)}\right] = k_{i,j}\left(\frac{1}{p}\sum_q \mathbb{E}_{\sigma_q}\left[\frac{1}{p}\sum_r \mathbb{E}_{\sigma_r}[l_{\sigma_q(i), \sigma_r(j)}]\right]\right) = \frac{1}{m^2}\sum_{t,u \in [m]} k_{i,j} l_{t,u}.$$

Therefore, the Rao-Blackwellization is just the biased HSIC estimator

$$\mathbb{E}\left[\widehat{\mathrm{MMD}}^2(\mathbb{Z}, \boldsymbol{\sigma}\mathbb{Z}) \mid \mathbb{Z}\right] = \widehat{\mathrm{HSIC}}(\mathbb{Z}). \tag{22}$$

It follows from the law of total variance that

$$\mathrm{Var}[\widehat{\mathrm{MMD}}^2(\mathbb{Z}, \boldsymbol{\sigma}\mathbb{Z})] = \mathrm{Var}[\mathbb{E}[\widehat{\mathrm{MMD}}^2(\mathbb{Z}, \boldsymbol{\sigma}\mathbb{Z}) \mid \mathbb{Z}]] + \mathbb{E}[\mathrm{Var}[\widehat{\mathrm{MMD}}^2(\mathbb{Z}, \boldsymbol{\sigma}\mathbb{Z}) \mid \mathbb{Z}]] \geq \mathrm{Var}[\widehat{\mathrm{HSIC}}(\mathbb{Z})].$$

The result follows. □

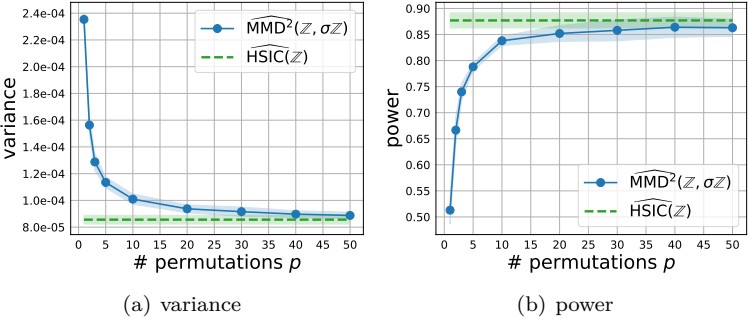

(a) variance         (b) power

Figure 12: (a) compares the sample variance of the biased MMD estimator using $p$ shuffles of the original sample. (b) examines how the number of shuffles $p$ affects the empirical test power. All results are based on $m = 50$ samples from the Sinusoid problem with frequency parameter $\ell = 1$. Both MMD and HSIC use Gaussian kernels $k$ and $l$ with bandwidth 1.

Figure 12 shows that the higher variance of the permuted MMD estimator negatively impacts its overall test power compared to HSIC. Also, including more permutations seems to decrease the overall variance, although never lower than that of HSIC.

In practice, the kernels learned by MMD-D and HSIC-D may not correspond, and so Proposition F.2 is inapplicable. That said, we observe the same phenomenon in experiments. Figure 13 shows estimates of the asymptotic variance of MMD and HSIC along a training trajectory. For permuted MMD we consider both a single shuffling of the data (MMD-full), as well as a split shuffling (MMD-split) where we use half the data for our joint distribution sample, and the other half to permute for our product-of-marginals sample. We note that the initial variance of MMD-split is substantially higher than that of MMD-full, which is much higher than the variance of HSIC. MMD-split/full also exhibit greater final variances, particularly at larger batch sizes.

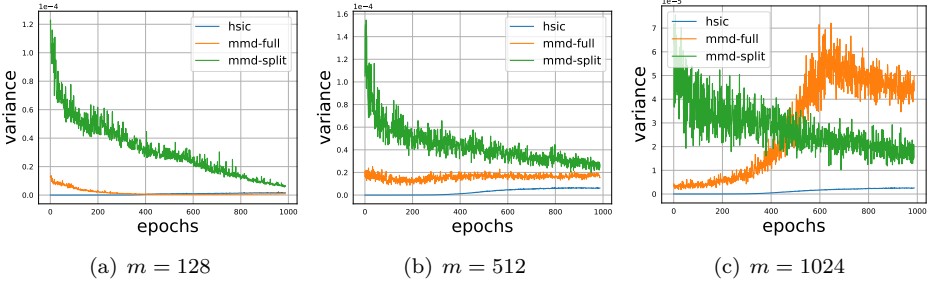

(a) $m = 128$       (b) $m = 512$       (c) $m = 1024$

Figure 13: Estimates of the asymptotic variance of HSIC (blue), MMD with a single permutation (orange), and MMD with a split permutation (green) along a training trajectory for HDGM-10 at sample sizes $m = 128$ (a), $m = 512$ (b), and $m = 1024$ (c).

### F.5   SNR Pitfall

Recent work by Ren et al. (2024) argues a so-called pitfall of the HSIC signal-to-noise ratio paradigm. They identify a corner case whereby, when the bandwidth of one of kernel $k$ or $l$ approaches 0, ignoring the threshold term causes the SNR objective $J_{w/o} = \widehat{\text{HSIC}}_b/\hat{\sigma}_{\mathfrak{H}_1}$ to differ from the true asymptotic power objective $J_{w/} = \left(\widehat{\text{HSIC}}_b - r/m\right)/\hat{\sigma}_{\mathfrak{H}_1}$ by a factor of $-(m-1)$, where $r$ denotes the asymptotic threshold. We argue that this is not a cause for concern, and in some cases, ignoring this is even preferred.

First, the behavior of the SNR objective in the bandwidth limit tells us nothing about the actual global maximum. Their argument tells us that $J_{w/o} = -(m-1)J_{w/}$ when the bandwidth $w_X \to 0$ and for a specific estimator $r = \mathbb{E}[m\widehat{\text{HSIC}}_b]$. This does not imply, however, that $J_{w/o}$ explodes as $w_X \to 0$ since the SNR objective is optimized at a fixed sample size $m$, and even in the bandwidth limit $J_{w/o}$ can not be $-\infty$. The latter observation is true because $\widehat{\text{HSIC}}_b$ is bounded and our variance estimates $\hat{\sigma}^2_{\mathfrak{H}_1}$ are also bounded away from zero. Therefore even though $J_{w/o}$ is high when $w_X \searrow 0$, this value is not necessarily the global maximum. Second, their argument seems entirely dependent on the choice of estimators. For instance, if we use $\widehat{\text{HSIC}}_u$ rather than $\widehat{\text{HSIC}}_b$, and take the asymptotic threshold to be $\mathbb{E}[m\widehat{\text{HSIC}}_u] = m\,\text{HSIC}(\mathbb{P}_x \times \mathbb{P}_y) = 0$, we get that $J_{w/o} = J_{w/}$ in this regime.

As further evidence, we plot both $J_{w/}$ and $J_{w/o}$ at varying bandwidths $w_X$ on the ISA dataset used in Ren et al. (2024). We use the same settings as they do: $m = 250$, $d = 3$, $\theta = \pi/10$, $w_Y = 1.0$. Results are shown in Figure 14. For very small bandwidths, $J_{w/o}$ does not explode and is not the global maximum. Moreover, the actual maximum agrees relatively well between $J_{w/}$ and $J_{w/o}$ at larger sample sizes $m$. We were unable to reproduce their Figure 1.

Additionally, Figure 14 (a) shows that ignoring this threshold term may actually be preferred. Consider the limiting behavior as the bandwidth $\omega_X$ goes to infinity. In this regime, the Gram and centered Gram matrices are respectively $K|_{\omega=\infty} = \mathbf{1}\mathbf{1}^T$ and $K_c|_{\omega=\infty} = \mathbf{0}$, which tells us that $\widehat{\text{HSIC}}_b = \text{Tr}[K_c L]/m = 0$. Since $m\widehat{\text{HSIC}}_b$ is degenerate its variance and $1 - \alpha$ quantile are zero, and so both $J_{w/}$ and $J_{w/o}$ are also 0 in this bandwidth limit.[13] Now consider a difficult problem where the power at initialization is less than 0.5, meaning $J_{w/}$ is less than 0. Then the maximum of $J_{w/}$ is erroneously this bandwidth limit.

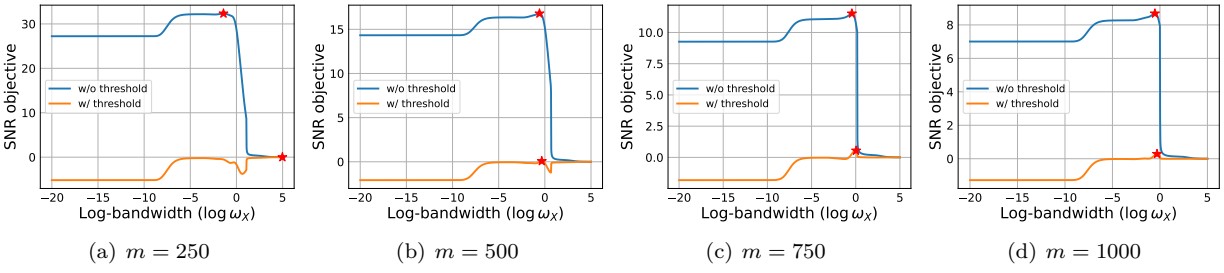

(a) $m = 250$          (b) $m = 500$          (c) $m = 750$          (d) $m = 1000$

Figure 14: Plots of $J_{w/o}$ (blue) and $J_{w/}$ (orange) at very small bandwidths $\omega_X$ with sample size $m$. For the threshold estimate, we use the .95-quantile of the Gamma approximation. The objective maximum is indicated by stars.

### F.6   SNR with Threshold

Although maximizing the HSIC signal-to-noise objective $J_{\text{HSIC}}$ is sufficient for learning powerful tests, it may also be beneficial to estimate the threshold-dependent term in the asymptotic power expression of Equation (6). Adopting the same notation as before, we define the SNR with threshold expression as

$$\tilde{J}_{\text{HSIC}}(X, Y; k, l) = \frac{\text{HSIC}(X, Y; k, l)}{\sigma_{\mathfrak{H}_1}(X, Y; k, l)} - \frac{\Psi^{-1}(1 - \alpha)}{m\sigma_{\mathfrak{H}_1}(X, Y; k, l)}$$

---

[13]Recall that our variance estimate in $J$ adds a small positive constant for stability, preventing $J$ from being undefined.

where $\Psi$ is the problem-dependent cdf described in Proposition A.2. Estimating $\tilde{J}_{\text{HSIC}}$ proves troublesome, largely due to $\Psi^{-1}$ depending on both the variables $X, Y$ as well as kernels $k, l$. Additionally, it needs to be differentiated with respect to those kernel parameters. We opt to approximate the null distribution $\Psi$ with a Gamma distribution $F_{\nu,\theta}$ with shape and scale parameters

$$\nu = \frac{\mathbb{E}[\widehat{\text{HSIC}}_b]^2}{\text{Var}[\widehat{\text{HSIC}}_b]} \qquad \text{and} \qquad \theta = \frac{m\text{Var}[\widehat{\text{HSIC}}_b]}{\mathbb{E}[\widehat{\text{HSIC}}_b]},$$

as suggested by Gretton et al. (2007). When performing gradient updates, we need to compute the gradients $\nabla_\nu F_{\nu,\theta}^{-1}(1-\alpha)$ and $\nabla_\theta F_{\nu,\theta}^{-1}(1-\alpha)$. This is typically not possible with auto-differentiation, as the inverse cdf is not tractable. Instead, we propose a solution based on implicit differentiation: suppose $f_{\nu,\theta}$ is the Gamma pdf and $r = F_{\nu,\theta}^{-1}(1-\alpha)$. Then

$$\nabla_\nu F_{\nu,\theta}^{-1}(1-\alpha) = -\frac{\nabla_\nu F_{\nu,\theta}(r)}{f_{\nu,\theta}(r)} \qquad \text{and} \qquad \nabla_\theta F_{\nu,\theta}^{-1}(1-\alpha) = -\frac{\nabla_\theta F_{\nu,\theta}(r)}{f_{\nu,\theta}(r)}.$$

The gradient of the Gamma cdf with respect to the scale $\theta$ yields the simplified expression

$$\nabla_\theta F_{\nu,\theta}(r) = -\frac{1}{\theta \cdot \Gamma(\nu)} \left(\frac{r}{\theta}\right)^\nu e^{-r/\theta}.$$

The gradient $\nabla_\nu F_{\nu,\theta}(r)$ is more troublesome. We resort to approximating this gradient via series expansions, detailed in Moore (1982).

We compare the performance of HSIC and NDS tests maximized with and without the threshold-dependent term in Figure 15. For HSIC, including this term seems to hurt the overall test power for harder problems. This is expected, since the null distribution is difficult to accurately estimate. We use two approximations here: one being the Gamma approximation, and two being the series expansion used to estimate the derivative of the Gamma cdf with respect to the shape parameter. On the other hand, NDS with the threshold term yields higher test power. Again, this is not too surprising as, unlike for HSIC, the null distribution here follows the CLT and thus is easily approximated given enough samples.

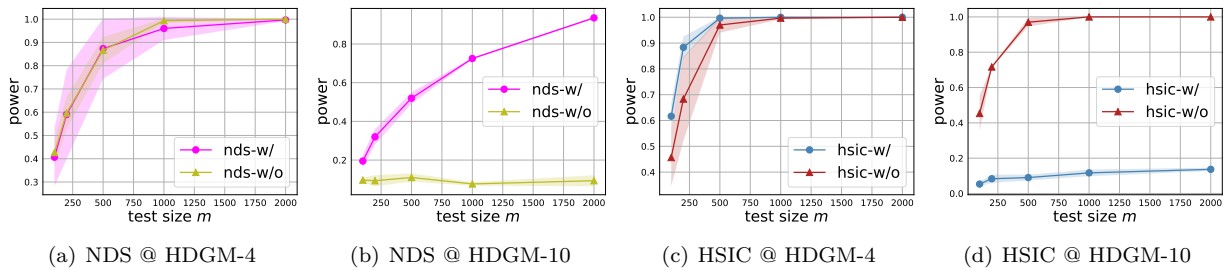

Figure 15: Power vs test size based on NDS and HSIC signal-to-noise objectives, both with (blue) and without (red) estimating the threshold-dependent term. Our proposed methods NDS (12) and HSIC-D (13) correspond to nds-w/ and hsic-w/o, respectively. We consider both HDGM-8 and HDGM-10, and use the same training configuration as specified in Section 6.

### F.7 Why do Permutation Tests deviate from Asymptotics?

As Figure 6 suggests, the asymptotics (i.e. tests based on the asymptotic null and alternative distributions) may not explain permutation tests well. Figure 16 illustrates why this might be: we observe strong dependency between the sample statistic $\hat{T}(\mathbb{X}, \mathbb{Y})$ and its rejection threshold, estimated via permutation.

For the InfoNCE critic this dependency is substantial, with a correlation between the two of 98%. While permutation tests still ensure the appropriate level, this coupling between the test statistic and permutation threshold is not accounted for in our asymptotic analysis. If these tests do follow their asymptotics, then we

expect this correlation to be relatively small, since the threshold should always be a reasonable estimate of the $1 - \alpha$ quantile. This is certainly not true for InfoNCE, and explains why its permutation test power is so much higher than what its asymptotics suggest. As for why or how InfoNCE critics exhibit such strong coupling, we do not yet know; this is a very interesting area for future work.

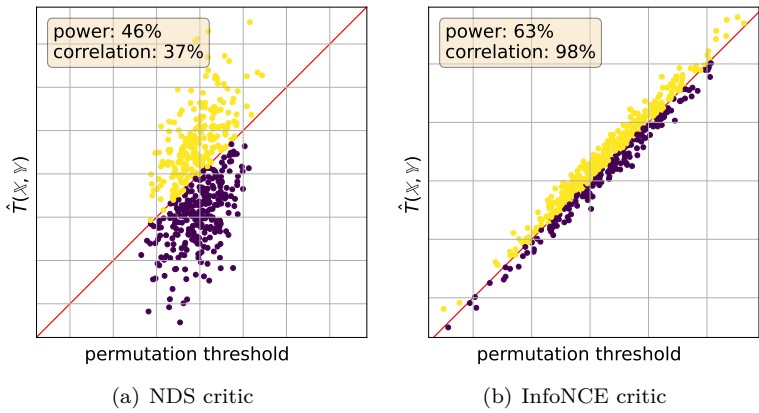

(a) NDS critic  (b) InfoNCE critic

Figure 16: Plot of the NDS statistic $\hat{T}$ evaluated on a sample $(\mathbb{X}, \mathbb{Y})$ from RatInABox, versus its rejection threshold estimated with 200 permutations. Each point corresponds to a separate test set $(\mathbb{X}, \mathbb{Y})$ of size $m = 1000$, but is evaluated with the same critic function $f$. The results in (a) use a critic that maximizes the NDS test power, as in Section 5. The critic in (b) directly maximizes InfoNCE. The red line is $y = x$, which is the rejection boundary; points above this line (colored yellow) are tests that reject the null hypothesis, while points below (purple) do not.

