# OpenReview forum: "Learning Representations for Independence Testing"
_TMLR — Accepted by TMLR_

### Review · Reviewer_iko6 · 2025-11-13

**Summary Of Contributions:**

This work studies representation learning for independence testing using variational MI estimators and HSIC-based kernel methods. It analyzes their connections, proposes a framework that optimizes an estimate of asymptotic test power through the signal-to-noise ratio, and demonstrates its effectiveness through theoretical results and experiments.

**Audience:**

Yes

**Claims And Evidence:**

Yes

**Requested Changes:**

I do not separate major and minor concerns in this review, and all revision suggestions are grouped together in the Requested Changes section.

---
### References

[1] Kim, Ilmun, and Aaditya Ramdas. “Dimension-agnostic inference using cross U-statistics.” Bernoulli, 2024.

[2] Shekhar, Shubhanshu, et al. “A permutation-free kernel two-sample test.” Advances in Neural Information Processing Systems, 2022.

[3] Gretton, Arthur, et al. “A kernel two-sample test.” Journal of Machine Learning Research, 2012.

[4] Cai, Zhanrui, et al. “Asymptotic distribution-free independence test for high-dimension data.” Journal of the American Statistical Association, 2024.

---
## Requested Changes
1. On page 4, the expression
$$
\frac{1}{\tau_{\mathfrak{H}1}} \sqrt{m}\big(\hat{T}-T_{\mathfrak{H}_1}\big) \xrightarrow{d} \mathcal{N}(0,1)
$$

requires that the function $f$ used in $\hat{T}$ does not depend on the evaluation samples $(Z_1, \ldots, Z_m)$. A brief explanation regarding the need for sample splitting, or at least an explicit acknowledgment of this issue, would improve clarity.

Emphasizing, as discussed in [1], that sample splitting is needed to remove the dependency between $f$ and the evaluation samples would help clarify why the stated asymptotic normality holds.

2. $$
\frac{\sqrt{m} \hat{T}}{\tau_{\Im_1}}>r_m, \quad m \widehat{\mathrm{HSIC}}_u>r_m
$$
would align better with conventional forms of test statistics based on CLTs.

3. The text should explicitly clarify that the quantities $a=a_m$ and $b=b_m$ depend on $m$, and that the relation $a \sim b$ refers to the limit as $m \to \infty$. Since this passage is intended to define the notation $a \sim b$, it would be helpful to state these points directly.

4. Clarify why CLT-based approximations justify ratio convergence of tail probabilities. The current argument does not fully justify this; further explanation is needed, especially in the context of Equation (3).

5. Clarify why the CLT is needed for discussing testing power. It would be helpful for the paper to explain why asymptotic normality is required to characterize power consistency, rather than directly showing that power converges to one as $(m \to \infty)$. The Proof of Theorem 8 in [2] may help illustrate how asymptotic normality contributes to such arguments.

6. When constructing samples from $P_X P_Y$ via permutation, it is unclear whether this yields i.i.d. samples from $P_X P_Y$. Please clarify how this differs from standard permutation HSIC tests and under what assumptions the MMD-based test is valid.

7. $\widehat{\operatorname{MMD}}^2_b$ first appears in Proposition 3.2, but its definition is provided only later in the text. Introducing the definition before this proposition would improve readability.

8. Clarifying with respect to which distribution each expectation is taken in the definition of $\mathrm{MMD}^2$  introduced above equation (8) would make the presentation clearer.

9. Explain the role of the concentration bound in constructing valid tests. The paper provides a bound but does not tie it directly to hypothesis testing. Additional explanation, possibly similar to Corollary 9 in [3] would help the reader understand how to form a rejection region.

10. Clarify the role of the parameter space $\Theta$ in the theoretical analysis. It is unclear why this specific space must be considered and why uniform convergence over this space is required. Also, the relation between parameter-space constraints and conditions such as $\lambda=\Theta(n^{-1 / 3})$ should be discussed.

11. Discuss the realism of the assumption, appearing in Theorem 5.1 and Theorem 5.2, that the population quantity governing the variance of the statistic is uniformly bounded away from zero. This condition may be too strong in practice. Commenting on possible relaxations or alternative assumptions would strengthen the theoretical contribution.

12. Expand the empirical evaluation to more challenging representation-learning settings. Since the method is motivated by the flexibility of deep networks, evaluating it on higher-dimensional modalities (such as images or text) would strengthen the empirical contribution. In addition, incorporating comparisons in high-dimensional tabular settings, as explored in [4], would provide a useful benchmark for understanding the method’s performance in more classical data regimes.

13. Clarify whether dimensions of $X$ and $Y$ must match. It would help to explicitly state whether the method requires equal-dimensional inputs or whether the framework supports mismatched dimensionalities.

**Strengths And Weaknesses:**

## Strengths
- The focus on optimizing the signal-to-noise ratio rather than the raw statistic is well motivated and leads to more meaningful gains in test power.
- The connection between MI-based statistics and HSIC is clearly articulated and helps unify two commonly used approaches in independence testing.
- The empirical evaluation is thorough and shows that representation learning can significantly improve performance in high-dimensional settings.
- The uniform convergence results provide useful theoretical support for the proposed objectives.

## Weaknesses
Several of the points discussed below are connected to specific suggestions in the Requested Changes section.
- The practical impact of the MI–HSIC connection on test power remains unclear, as the bounds used offer limited interpretability.
- Some assumptions in the theoretical analysis, including the structure of the parameter space and the choice of $\lambda=\Theta(n^{-1/3})$, are not fully justified.
- Several arguments rely on CLTs without sufficiently explaining how they support the tail-probability approximations used in testing.
- The assumption that certain population quantities are uniformly bounded away from zero may be restrictive in practice.
- Experiments focus mostly on low-dimensional tabular data, leaving open how well the method scales to more complex modalities where deep architectures are most beneficial.

---

> ### Author Response · Authors · 2026-01-26
>
> Thank you for your work in reviewing our paper! We have uploaded a new version addressing your comments as well as the other reviewers', and also respond to them below.
>
> #### Weaknesses
>
> > The practical impact of the MI–HSIC connection [...]
>
> We do not intend to use these bounds directly in practical settings; rather, we establish a connection between tests based on MI and HSIC, thus suggesting that they may share similar statistical properties. In 4.2, we make this connection explicit by showing that for kernels $k(x,x')=f(x)f(x')$ and $l(y,y')=g(y)g(y')$, permutation tests based on HSIC are roughly equivalent to those based on NCE with a separable critic $(x,y)\mapsto f(x)g(y)$.
>
> Other weaknesses: see requested changes.
>
> > Experiments focus mostly on low-dimensional tabular data [...]
>
> HDGM-40/50, while tabular, are high-dimensional relative to typical toy independence testing problems. Moreover, RatInABox, while synthetic, is a fairly accurate simulator for real-life neurological data – proposed as a replacement for neuroscientists to study without requiring physical animal experiments. In both these experiments we found that neural network-based approaches significantly outperform classical baselines.
>
> One challenge with using accessible real-world datasets is that they are typically designed for classification or regression tasks. When these problems are translated to independence testing, they tend to become too simple: for example, on CIFAR-10 a linear classifier achieves ~30% accuracy, clearly implying dependence between images and their labels. On this same problem, the median heuristic HSIC test achieves perfect power with only a few hundred samples. Our optimization-based methods, on the other hand, shine in problems with much subtler dependencies. We believe that such questions do arise in real scientific and engineering tasks, but lacking the tools to address them, people usually don't think to ask or publicize such problems.
>
> #### Requested Changes
>
> For points without a response, we have made the appropriate change in the updated paper.
>
> > 4) Clarify why CLT-based approximations justify ratio convergence of tail probabilities
>
> Sorry, our notation here perhaps implied a stronger claim than we meant: we've clarified to avoid ~ in favor of additive $o(1)$ error terms.
>
> > 5) Clarify why the CLT is needed for discussing testing power
>
> Although consistency guarantees that the test power eventually goes to one, it doesn't tell us anything about the rate of convergence; as long as its kernel is characteristic, e.g. a Gaussian kernel with any fixed bandwidth, the HSIC test is consistent for fixed alternatives. To maximize power, we need a finer-grained understanding than only that it is consistent.
>
> The CLT gives us a convenient way to characterize the asymptotic power, and hence a good objective to optimize. Our new Appendix B also justifies the same objective from a non-asymptotic Bernstein-type inequality, helping solidify the motivation.
>
>
> > 6) When constructing samples [...]
>
> Thanks for emphasizing this question, which made us realize that what we did previously was not quite correct (see the new footnote 3 on page 8); we have corrected it, obtaining slightly lower power for the independence-as-two-sample tests. We can, though, obtain exact validity with a slight tweak; we've re-written 3.2 to describe.
>
> > 9) Explain the role of the concentration bound [...]
>
> If you're referring to Theorems 5.1 and 5.2 these bounds are not used to construct tests, but rather show consistency of our optimization objective. One could construct a non-asymptotic test based on a confidence interval from a concentration inequality, but these tend to be extremely conservative (and hence much lower-power); e.g. see Gretton et al. (2012a, Section 4).
>
> > 10) Clarify the role of the parameter space [...]
>
> Uniform convergence over Banach balls is a standard approach in statistical learning theory, applying to many practical problems. The assumption that the kernel is uniformly Lipschitz and bounded is also standard and is satisfied for a broad class of kernels, including the Gaussian kernel. For deep kernels this is true when the featurizer network is uniformly Lipschitz, as is the case for typical MLP architectures on a bounded parameter space.
>
> $\lambda = \Theta(n^{-1/3})$ is the regularizer giving the best asymptotic bound. Letting $\lambda$ decay as we see more samples is sensible since the variance estimate should also improve, allowing weaker regularization. We have added discussion to our theoretical analysis section.
>
> > 11) Discuss the realism [...]
>
> The variance being bounded away from zero is admittedly not elegant, but as long as the statistic is non-degenerate there is some constant satisfying this. Relaxing this assumption is tricky; Gretton et al. (2012b), Liu et al. (2020), and Ren et al. (2024) all found it unavoidable. This challenge stems from a $1/\sigma$ in the bound.

---

### Review · Reviewer_Hm98 · 2025-12-30

**Summary Of Contributions:**

The paper makes a number of contributions in the area of independence testing via variational mutual information (MI) bounds and Hilbert-Schmidt Independence Criterion (HSIC) bounds. The focus is mainly on statistical tests of independence for high-dimensional continuous variables where classical tests like chi-square or parametric assumptions are not applicable. The kernel based methods might not perform well due to poor representation of the data. The authors propose a method to optimize the representation of the data to make the dependency more explicit.  In short the key contributions are:

1. The authors introduce a new testing technique based on variational MI estimators (e.g., InfoNCE, NWJ). Previously, only estimators based on variational bounds were of main focus.
2. Establishes a connection between variational MI-based tests and the HSIC, showing that HSIC serves as a lower bound on MI.
3. Proposes an algorithmic framework to learn optimal data representations by maximizing the Signal-to-Noise Ratio (SNR) of the test statistic, which directly increases the power of the test.
4. Provides uniform convergence proofs (Theorems 5.1 and 5.2) for the SNR estimators, guaranteeing that representations optimized on finite samples reliably generalize to population-level test power.
5. Corrections to Literature such as identifying and correcting mathematical errors in prior work (Ren et al., 2024) and expands the use of deep kernels in HSIC optimization.
6. Theoretical results are supported by experiments on a range of synthetic and real world datasets. Empirical results suggest that HSIC-D dominates MI-based tests.

**Audience:**

Yes

**Claims And Evidence:**

Yes

**Requested Changes:**

See weaknesses 1 (critical) and 4 (minor).

**Strengths And Weaknesses:**

Strengths:
1. Apart from the contributions mentioned above, the authors also provide a clear analysis of the results and comparison+background  with existing methods.
2. The paper is well written and the contributions are clear.
3. The authors highlight the difference between maximizing a statistic value vs maximizing its statistical power esp for MI based bounds.
4. HSIC is shown to be a lower bound on mutual information.
5. The authors justify the ``learning representations'' approach through Uniform Convergence arguments. They show that (under regularity assumptions)  the representations optimized by Algorithm 1 are maximizes a population SNR objective within a restricted class of kernels.


Weaknesses:
1. The main idea of optimizing deep kernels by maximizing the test power (SNR) rather than the statistic value has been studied by several papers including Liu et al. (2020) and others. Although this has been acknowledged in the paper, the main proof sections (Appendix B and C) are also acknowledged to be based on Liu et al. (2020). It would be more helpful for readers if the authors could provide a more detailed comparison of proof techniques with Liu et al. (2020) and may be with other related papers.
2. In section 3.1, the optimization objective is based on maximizing the SNR (HSIC/sigma), which is derived from a CLT-based Gaussian approximation. However, this approximation is least accurate when the signal is weak. I am curious if maximizing a Gaussian-based SNR is "right thing to do" when the true distribution of the statistic is closer to a weighted sum of Chi-squares (as mentioned in Prop. A.2). In this regime the learned representations are even more valuable.
3. (Minor) The tests in this paper are based on data splitting between training and test sets (which might reduce the power for a fixed set of samples). Although I believe the analysis will become more complex if we try to remove the data splitting.
4. (Minor) All learned methods are evaluated using sample splitting between representation learning and evaluating the test statistic. While this is acceptable from theoretical point of view, it might be informative to include a diagnostic experiment showing what happens if the same data are used for both training and testing, to better understand the practical impact of sample splitting and potential overfitting effects (such as seeing the percentage of times the outputs in the two cases agree).

---

> ### Author Response · Authors · 2026-01-26
>
> Thank you for your work in reviewing our paper! We have uploaded a new version addressing your comments as well as the other reviewers', and also respond to them below.
>
> #### Weaknesses / Requested Changes
>
> > 1) The main idea of optimizing deep kernels by maximizing the test power (SNR) rather than the statistic value has been studied by several papers [...]
>
> Our approach for proving uniform convergence involves decomposing the SNR objective into differences between estimators and their corresponding population value (e.g. $|\widehat{\text{HSIC}}_u - \text{HSIC}_u|$, $|\hat{\sigma}^2 - \sigma^2|$, etc...), and then separately proving uniform convergence for each of these estimates with a covering number based proof. The main theorem then follows via a union bound. This is the same general approach as taken by Liu et al. (2020), but our proof is slightly more involved, for a few reasons. 1) Uniform convergence of HSIC and its variance estimators now include two kernels, $k$ and $l$, instead of one, and a somewhat different estimator form. 2) For NDS we also need to show convergence of a null-to-alternative variance ratio, a term which is not present in the MMD or HSIC objectives.
>
> That said, the convergence results are not the main contribution of our paper: they are indeed extensions of the approach of Liu et al. Our results are provided to show that optimizing the proposed power objective (with or without the threshold term) is a justified approach to maximizing the true asymptotic power.
>
>
>
> > 2) In section 3.1, the optimization objective is based on maximizing the SNR (HSIC/sigma), which is derived from a CLT-based Gaussian approximation [...]
>
> Indeed, at smaller test sizes the CLT may not yet kick in, and so there is no guarantee that our learned critic actually maximizes the test power at these smaller sample sizes. To clarify our view of the situation slightly: at training time, no matter the size of our training set, we attempt to maximize the power of a test with a large test size. When we apply that test to whatever actual test set we will be using, we must hope that the large-sample powerful test will still have good power on the finite sample. In some sense, this is an unavoidable consequence of building on asymptotic analyses, and typical to a huge portion of work in statistics. Our experiments seem to support this assumption: the power expression is a good approximation (Figure 3), and HSIC-O is consistently better than the median heuristic even when using <100 samples (Figure 4).
>
> For U-statistics, we can also derive a power bound that doesn't depend on the CLT, instead using a Bernstein-type inequality that holds non-asymptotically. In that case, maximizing an exact lower bound on the power also motivates essentially maximizing the SNR. We have added this derivation as a new Appendix B.
>
>
> > 3) (Minor) The tests in this paper are based on data splitting between training and test sets (which might reduce the power for a fixed set of samples). Although I believe the analysis will become more complex if we try to remove the data splitting.
>
>
> Indeed, data splitting results in a smaller test set which reduces the power. It would be ideal to both train and test on the same data while maintaining correct level control, but this is quite challenging; previous attempts on this selective inference/post-selection inference problem have only worked on very restricted algorithmic settings ([Kübler et al. 2020](https://arxiv.org/abs/2006.02286))
>
>
>
> > 4) (Minor) All learned methods are evaluated using sample splitting between representation learning and evaluating the test statistic. While this is acceptable from theoretical point of view, it might be informative to include a diagnostic experiment showing what happens if the same data are used for both training and testing, to better understand the practical impact of sample splitting and potential overfitting effects (such as seeing the percentage of times the outputs in the two cases agree).
>
> If the same samples are used for both training and testing then several things would break. First, our asymptotics would no longer apply, as the CLT requires i.i.d. evaluation samples. This means objectives (10) and (11) are no longer consistent estimates of the asymptotic test power and maximizing them becomes questionable. Secondly, any permutation test on this training data cannot guarantee correct type-I error, since the test statistic under permutations are not identially distributed (Hemerik & Goeman, 2018). This means that, even though the power could be higher, the test would likely not be valid. To see this most clearly, consider a critic/kernel that drastically overfits to the particular training set.

---

### Review · Reviewer_7pdE · 2026-01-02

**Summary Of Contributions:**

This paper explores the problem of learning a witness function to test for dependence between two random variables. The paper aims to learn a function that provides the most powerful permutation test for dependence between two variables. In order to learn this function, they optimise asymptotic power expressions in the training set, which used as an optimisable proxy for the power of the final test. They use two forms of this test, one where the test statistic is the mean of the learned function (they term this the neural dependency statistic) and one using HSIC with deep kernels. They experimentally demonstrate strong performance of their method compared to alternatives.

**Audience:**

Yes

**Claims And Evidence:**

Yes

**Requested Changes:**

Based on the above, I would ask for the following:
- Can the authors include a discussion or provide their thoughts on hyperparameter optimisation in their setting?
- Can the authors add discussion on sample splitting or K-fold validation.

**Strengths And Weaknesses:**

Strengths:
- The method certainly appears to me correct, is well motivated, and well explained.
- The switch to optimising the asymptotic power statistics is an interesting idea and it appears to lead to good results.
- The experimental results are extensive and convincing
- The paper includes a thorough theoretical analysis of their methods.

Weaknesses:
- There is little discussion of hyperparmeter optimisation for the learning algorithm which seems like it would require some thought for the best way to incorporate into the framework. For example, this could be done by using a validation set but then there is the question of if this should use a permutation test of the asymptotic statistic for validation. Alternatively some option of doing a hyperparameter sweep, permuting on the test set, and then doing multiple test corrections could be used. This is just to say the question of how to do hyperparameter optimisation seems an interesting and important one in this case and to the best of my knowledge it is not discussed.
- Based on the above, the authors do discuss the use of the validation set but it is not totally clear how it is used. I would appreciate more discussion on this.
- Not a weakness per say, but I think the method could potentially benefit from some form of sample splitting of the type more common in doubly robust estimation of causal effects, or k fold validation, and it would be nice to see some discussion of this. For example instead of having to split into one train/test set, the authors could use k different folds where the each fold consists of a different train/test split to train a model on and the final test statistic is calculated by evaluating each of the k functions on their test set. This would have the advantage of improving effective sample size.

---

> ### Author Response · Authors · 2026-01-26
>
> Thank you for your work in reviewing our paper! We have uploaded a new version addressing your comments as well as the other reviewers', and also respond to them below.
>
> #### Weaknesses
>
> > There is little discussion of hyperparmeter optimisation for the learning algorithm which seems like it would require some thought for the best way to incorporate into the framework. For example, this could be done by using a validation set but then there is the question of if this should use a permutation test of the asymptotic statistic for validation. Alternatively some option of doing a hyperparameter sweep, permuting on the test set, and then doing multiple test corrections could be used. This is just to say the question of how to do hyperparameter optimisation seems an interesting and important one in this case and to the best of my knowledge it is not discussed.
>
>
> As you point out, we can perform hyperparameter selection on a validation set; as long as this is separate from the test set, the permutation test still guarantees correct level control. As for the validation criterion, one natural approach is to evaluate the same approximation of the asymptotic power used for our training objective. (This is what we do in practice.) This avoids the need for relatively-expensive permutations, while providing a reasonable estimate of performance. Alternatively, we could also directly estimate the test statistic in cases where there isn't a simple expression for asymptotic power, though this approach may be less reliable since the statistic is not necessarily correlated with test power.
>
> The suggestion you proposed involving hyperparameter sweeping and multiple test corrections is very closely connected to the kernel aggregation procedure called MMDAgg (Schrab et al., 2023). This is certainly possible to do, though potentially expensive if hyperparameters need to be optimized. It's also unclear if this would perform better than simply using a validation set due to the conservatism of the multiple test correction procedure, though this is certainly an interesting idea to explore in future work.
>
>
> > Based on the above, the authors do discuss the use of the validation set but it is not totally clear how it is used. I would appreciate more discussion on this.
>
> The validation sets in the experiments are used for early stopping of the training process, not for explicit hyperparameter selection, based on the training objective. We have clarified this in the experiments section and the new Appendix A.1.
>
>
>
> > Not a weakness per say, but I think the method could potentially benefit from some form of sample splitting of the type more common in doubly robust estimation of causal effects, or k fold validation, and it would be nice to see some discussion of this. For example instead of having to split into one train/test set, the authors could use k different folds where the each fold consists of a different train/test split to train a model on and the final test statistic is calculated by evaluating each of the k functions on their test set. This would have the advantage of improving effective sample size.
>
> There are certainly many ideas in this general area which could be interesting to explore further. A related approach is MMDFuse (Biggs et al., 2023), which constructs an aggregate statistic by combining MMD estimates at varying bandwidths. This approach could be naturally extended to consider estimates based on different training sets, and would naturally fit into our optimization framework. Doing so in a way that doesn't significantly reduce the critic quality (and hence test power), while also neither breaking the validity of the permutation test nor exploding the amount of required computation, would require care and careful evaluation beyond the scope of a minor addition to this paper; we have added a mention of this as a potential direction for future work.
>
>
> #### Requested Changes
>
> > 1) Can the authors include a discussion or provide their thoughts on hyperparameter optimisation in their setting?
>
> > 2) Can the authors add discussion on sample splitting or K-fold validation.
>
> As discussed above, we have added Appendix E.2 to our paper to include more discussion of both of these interesting issues.

---

### Decision · Action_Editor_mvvU · 2026-02-22

**Recommendation:** Accept with minor revision

**Additional Comments:**

Please substantiate or revise the claim regarding Ren et al. (2024)'s proof, as discussed above.

**Audience:**

Yes

**Audience Explanation:**

The paper addresses independence testing in high dimensions, connecting MI-based and kernel-based approaches in a unified framework. While the audience is specialized, the topic is well-represented within the TMLR readership, and the findings — particularly the connection between the two test families and the empirical observation about permutation test power — provide useful insights for both practitioners and theorists.

**Claims And Evidence:**

Yes

**Claims Explanation:**

Three reviewers with appropriate expertise in kernel methods, information theory, and nonparametric testing evaluated this submission. All found the paper technically sound, well-written, and supported by thorough theoretical analysis and experiments. The review process was productive — detailed questioning by Reviewer iko6 led the authors to discover and correct an error in their MMD-based test construction, and reviewer feedback prompted useful additions including a non-asymptotic justification for the SNR objective (Appendix B).

**Minor concern: Discussion of related work by Ren et al.** The paper makes a claim about prior work by [Ren et al.] that the AE is not able to substantiate based on the exposition. At the end of the introduction the authors write:

> We also note that Ren et al. (2024) recently proposed a closely related scheme for the HSIC power optimization setting; unfortunately, however, we show that several of their main algorithmic suggestions are misguided and the proof of their main theorem is incorrect.

The algorithmic disagreement (about whether to include a threshold term in the HSIC optimization objective) is argued in Appendix F.5 with analysis and supporting plots. However, the claim about the incorrect proof is supported only by footnote 10, which states:

> Ren et al. (2024) stated a similar result to Theorem 5.2, but with an incorrect proof; although they claim uniform convergence over their threshold estimate, their proof makes no attempt at showing uniformity, instead showing pointwise convergence.

Based on this short footnote alone, the AE is cannot easily assess whether there is indeed a problem with the proof in the Ren et al. paper, or what the extent of the problem is. The AE would therefore like to ask the authors to either expand on this statement, or to revise it. The AE may ask reviewers to comment on this particular point before approving the camera ready.

---

> ### Author Response · Authors · 2026-02-27
>
> Thank you! We have uploaded a camera-ready revision expanding the details in that footnote (as well as doing the other camera-ready things).

---

> > ### Comment · Action_Editor_mvvU · 2026-03-15
> > **Thank you for the revision –– Could you take one last look at wording in the introduction?**
> >
> > Thank you for the revision. I think your updated text in the footnote communicates the apparent issue with the proof more clearly.
> >
> > I do notice that you now hedge a bit more in the footnote (you now write "seemingly incorrect proof"), but the text in the introduction remains as before:
> >
> > > We also note that Ren et al. (2024) recently proposed a closely related scheme for the HSIC power optimization setting; unfortunately, however, we show that several of their main algorithmic suggestions are misguided and the proof of their main theorem is incorrect.
> >
> > I would recommend updating this particular sentence to more align with the wording later on, e.g., something like
> >
> > > We also note that Ren et al. (2024) recently proposed a closely related scheme for the HSIC power optimization setting; however, we argue that several of their main algorithmic suggestions are not well-supported (Appendix F.5) and identify an apparent gap in the proof of their main theorem (footnote 10).
> >
> > This is ultimately up to your judgement as authors; If you wish to keep the text as is, then the AE will accept.

---

> > > ### Author Response · Authors · 2026-03-19
> > >
> > > Thanks – we've reworded according to your suggestion, and also forgot that we hadn't previously added a repository link. This version should be good!